# Neural Lagrangian Schrödinger Bridge: Diffusion Modeling for Population Dynamics

**Takeshi Koshizuka, Issei Sato**
The University of Tokyo
{koshizuka-takeshi938444, sato}@g.ecc.u-tokyo.ac.jp

## Abstract

Population dynamics is the study of temporal and spatial variation in the size of populations of organisms and is a major part of population ecology. One of the main difficulties in analyzing population dynamics is that we can only obtain observation data with coarse time intervals from fixed-point observations due to experimental costs or measurement constraints. Recently, modeling population dynamics by using continuous normalizing flows (CNFs) and dynamic optimal transport has been proposed to infer the sample trajectories from a fixed-point observed population. While the sample behavior in CNFs is deterministic, the actual sample in biological systems moves in an essentially random yet directional manner. Moreover, when a sample moves from point A to point B in dynamical systems, its trajectory typically follows *the principle of least action* in which the corresponding action has the smallest possible value. To satisfy these requirements of the sample trajectories, we formulate the Lagrangian Schrödinger bridge (LSB) problem and propose to solve it approximately by modeling the advection-diffusion process with regularized neural SDE. We also develop a model architecture that enables faster computation of the loss function. Experimental results show that the proposed method can efficiently approximate the population-level dynamics even for high-dimensional data and that using the prior knowledge introduced by the Lagrangian enables us to estimate the sample-level dynamics with stochastic behavior.

## 1 Introduction

The population dynamics of time-evolving individuals appears in various scientific fields, such as cell population in biology (Schiebinger et al., 2019; Yang & Uhler, 2018), air in meteorology (Fisher et al., 2009), and healthcare statistics (Manton et al., 2008) in medicine. However, tracking individuals over a long period is often difficult due to experimental costs. Furthermore, it can sometimes be impossible to track the time evolution. For example, since single-cell RNA sequencing (scRNA-seq) destroys all measured cells, we cannot analyze the behavior of individual cells over time in cell transcriptome measurements. Instead, we only obtain individual samples from cross-sectional populations without alignment across time steps at a few distinct time points. Under these constraints on data measurements, our goal is to better understand the time evolution of samples in the populations.

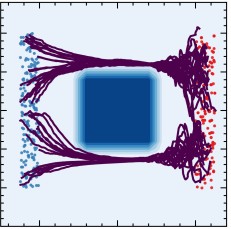

Figure 1: Example of trajectories by NLSB.

Existing methods attempt to estimate population-level dynamics following the Wasserstein gradient flow using a recurrent neural network (RNN) (Hashimoto et al., 2016) or the Jordan-Kinderlehrer-Otto (JKO) flow (Bunne et al., 2021). Recent studies have attempted to interpolate the trajectories of individual samples between cross-sectional populations at multiple time points by using optimal transport (OT) (Schiebinger et al., 2019; Yang & Uhler, 2018), or CNF (Tong et al., 2020). Using a CNF generates continuous-time non-linear sample trajectories from multiple time points. In addition, Tong et al. (2020) proposed a regularization for CNF that encourages a straight trajectory on the basis of the OT theory. Since the probability distribution transformation based on ordinary differential equations (ODEs) is used in CNF, the behavior of each sample is described by its initial condition in a

completely deterministic manner. However, samples in population are known to move stochastically and diffuse in nature, *e.g.*, biological system (Kolomgorov et al., 1937).

To handle the stochastic and complex behavior of individual samples, we propose to model the advection-diffusion processes by using SDEs to describe the time evolution of the sample. Furthermore, on the basis of *the principle of least action*, we estimate the sample trajectories that minimize *action*, defined by the time integral of the Lagrangian determined from the prior knowledge. We formulate this problem as the Lagrangian Schrödinger bridge (LSB) problem, which is a special case of the stochastic optimal transport (SOT) problem, and propose an approximate solution method neural Lagrangian Schrödinger bridge (NLSB). In NLSB, we train regularized neural SDE (Li et al., 2020; Tzen & Raginsky, 2019a;b) by minimizing the Wasserstein loss between the ground-truth and the predicted population. The Lagrangian design defining regularization allows the sample-level dynamics to reflect various prior knowledge such as OT, manifold geometry, and local velocity arrows proposed by Tong et al. (2020). In addition, we parameterize a potential function instead of the drift function. Adopting the model architecture of the potential function from OT-Flow (Onken et al., 2021) will speed up the computation of the potential function's gradient and the regularization term. As a result, we capture the population-level dynamics as well as or better than conventional methods, and can more accurately predict the sample trajectories.

In short, our contributions are summarized as follows.

1. We formulate the LSB problem to estimate the stochastic sample trajectory according to *the principle of least action*.
2. We propose NLSB to approximate the LSB problem practically by modeling the advection-diffusion process with regularized neural SDE on the basis of the prior knowledge introduced by the Lagrangian.
3. We adopt the model architecture of the potential function from OT-Flow(Onken et al., 2021) to speed up the computation of the regularization term to minimize HJB-PDE loss.

## 2 BACKGROUND

In Section 2.1, we introduce the method of combining CNF and the OT theory, the basis of our method. Then, we explain dynamics modeling techniques: the diffusion modeling using neural SDE (Section 2.2) and the SOT theory (Section 2.3).

### 2.1 FLOWS REGULARIZED BY OPTIMAL TRANSPORT

CNFs (Chen et al., 2018) are a method for learning the continuous transformation between two distributions $p$ and $q$ by modeling the ordinary differential equation (ODE):

$$\frac{\mathrm{d}\mathbf{x}(t)}{\mathrm{d}t} = \mathbf{f}_\theta(\mathbf{x}, t), \qquad \text{subject to} \quad \mathbf{x}(t_0) \sim p, \ \mathbf{x}(t_1) \sim q,$$

where $\mathbf{f}_\theta$ is the velocity model with learnable parameters $\theta$.

Several regularizations of CNFs leading to straight trajectories have been proposed on the basis of the OT theory. The likelihood maximization problem of regularized CNF is derived by replacing the terminal constraint of the Brenier-Benamou formulation (Benamou & Brenier, 2000) with Kullback–Leibler (KL) divergence. RNODE (Finlay et al., 2020), OT-Flow (Onken et al., 2021), and TrajectoryNet (Tong et al., 2020) introduced a regularization $\tilde{\mathcal{R}}_e$ in Eq. (1). Potential Flow (Yang & Karniadakis, 2020) and OT-Flow modeled the potential function $\Phi$ that satisfies $\mathbf{f} = -\nabla\Phi$ instead of modeling the velocity function $\mathbf{f}$. They also proposed an additional OT-based regularization $\tilde{\mathcal{R}}_h$ derived from the Hamilton–Jacobi–Bellman (HJB) equation (Evans, 1983) satisfied by the potential function shown in Eq. (2). These OT-based regularizations have resulted in faster CNFs (Finlay et al., 2020; Onken et al., 2021) and improved the modeling of cellular dynamics (Tong et al., 2020).

$$\tilde{\mathcal{R}}_e = \int_{t_0}^{t_1} \int_{\mathbb{R}^d} \frac{1}{2} \|\mathbf{f}_\theta(\mathbf{x}, t)\|^2 \, \mathrm{d}\rho_t(\mathbf{x}) \, \mathrm{d}t, \tag{1}$$

$$\tilde{\mathcal{R}}_h = \int_{t_0}^{t_1} \int_{\mathbb{R}^d} \left| \partial_t \Phi_\theta(\mathbf{x}, t) - \frac{1}{2} \|\nabla_\mathbf{x} \Phi_\theta(\mathbf{x}, t)\|^2 \right| \, \mathrm{d}\rho_t(\mathbf{x}) \, \mathrm{d}t, \tag{2}$$

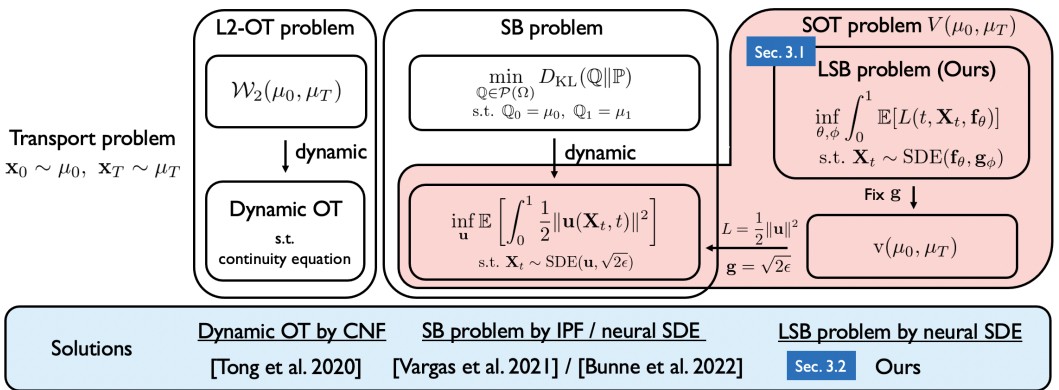

Figure 2: Overview

where $\rho_t$ is the law of the sample $\mathbf{x}$ at the time $t$.

## 2.2 NEURAL SDE AND APPLICATIONS TO DIFFUSION MODELING

SDE has been used to model real-world random phenomena in a wide range of areas, such as chemistry, biology, mechanics, economics and finance (Higham, 2001). As an extension of neural ODEs, neural SDE has been proposed to model drift and diffusion functions with neural networks (NNs), as follows.

$$d\mathbf{X}_t = \mathbf{f}_\theta(\mathbf{X}_t, t)\, dt + \mathbf{g}_\phi(\mathbf{X}_t, t)\, d\mathbf{W}_t, \tag{3}$$

where $\{\mathbf{X}_t\}_{t \in [t_0, t_{K-1}]}$ is a continuous $\mathbb{R}^d$-valued stochastic process, $\mathbf{f}_\theta : \mathbb{R}^d \times [t_0, t_{K-1}] \mapsto \mathbb{R}^d$ is a drift function, $\mathbf{g}_\phi : \mathbb{R}^d \times [t_0, t_{K-1}] \mapsto \mathbb{R}^{d \times m}$ is a diffusion function and $\{\mathbf{W}_t\}_{t \in [t_0, t_{K-1}]}$ is an $m$-dimensional Wiener process. Score-based generative models (SGM) (Song et al., 2020), which use score matching to learn the reverse diffusion process of generating images from noise as SDE, have demonstrated the ability to produce high-quality data. Other recent studies on diffusion modeling using SDE (De Bortoli et al., 2021; Wang et al., 2021; Vargas et al., 2021; Chen et al., 2021a; Zhang & Chen, 2022) have proposed learning SDE solutions to the SB problem. Vargas et al. (2021) and De Bortoli et al. (2021) solved the SB problem by combining iterative proportional fitting (IPF) with mean-matching regression of the SDE drift function using Gaussian process (GP) and NN, respectively.

## 2.3 STOCHASTIC OPTIMAL TRANSPORT

Mikami (2008) generalized the OT problem and defined the SOT problem as a random mechanics problem determined by *the principle of least action*. The SOT problem with the endpoint marginals fixed to $\mu_0$ and $\mu_1$ is represented as

$$V(\mu_0, \mu_1) := \inf_{\mathbf{X} \in \mathscr{A}} \left\{ \mathbb{E}\left[ \int_{t_0}^{t_1} L\left(t, \mathbf{X}_t; \mathbf{f_X}(\mathbf{X}_t, t)\right)\, dt \right] \,\middle|\, \mathbf{X}_{t_0} \sim \mu_0,\ \mathbf{X}_{t_1} \sim \mu_1 \right\}, \tag{4}$$

where $L(t, \mathbf{x}, \mathbf{u})$ is continuous and convex in $\mathbf{u}$ and $\mathscr{A}$ is the set of all $\mathbb{R}^d$-valued, continuous semimartingales $\{\mathbf{X}_t\}_{t_0 \le t \le t_1}$ on a complete filtered probability space such that there exists a Borel measurable drift function $\mathbf{f_X}(\mathbf{X}_t, t)$ for which satisfies several conditions (see Appendix A.3). The function $L$, called the Lagrangian, is the transport cost defined on the space-time of the system and allows us to describe phenomena consistently regardless of the choice of coordinate system.

Mikami (2008) also introduced another variational version of the SOT problem for a flow of marginal distributions which satisfies the Fokker–Planck (FP) equation, which is a advection-diffusion equation describing the time evolution of the probability density function.

$$v(\mu_0, \mu_1) := \inf_{\mathbf{f}} \left\{ \int_{t_0}^{t_1} \int_{\mathbb{R}^d} L\left(t, \mathbf{x}; \mathbf{f}(\mathbf{x}, t)\right)\, d\rho_t(\mathbf{x})dt \,\middle|\, \rho_{t_k} = \mu_k, (\mathbf{f}, p_t) \text{ satisfies the FP eq} \right\}, \tag{5}$$

where $\rho_t$ and $p_t$ are the law and density of the random variable $\mathbf{X}_t$, respectively.

When $L(t, \mathbf{x}, \mathbf{u}) = \frac{1}{2}||\mathbf{u}||^2$, the SOT problem is regarded as the special case of Schrödinger bridge (SB) (Jamison, 1975; Chen et al., 2021b; Dai Pra, 1991; Léonard, 2013; Mikami, 1990) problem. See Appendix A.3 for details on the SOT theory.

## 3    PROPOSED METHOD

In this section, we first introduce Lagrangian-Schrödinger bridge (LSB) problem. We next present an approximate solution for the LSB problem: neural Lagrangian Schrödinger bridge (NLSB). Finally, we give specific examples of our method for specific use cases. Figure 2 illustrates the position of the LSB problem in the transport problem and the relationship between the NLSB and existing methods.

### 3.1    LAGRANGIAN SCHRÖDINGER BRIDGE

We consider the problem of estimating dynamics at both the population and individual sample levels by using probability distributions at two known end points. We make two realistic assumptions about the target system model.

1. The stochastic behavior of individual samples yields the population diffusion phenomenon.
2. Individual samples are encouraged to move according to *the principle of least action*.

Samples in populations, such as a biological system (Kolomgorov et al., 1937), are known to move stochastically and diffuse. The principle of least action is known as a fundamental principle in dynamical systems, which states that when a sample moves from point A to point B, its trajectory is the one that has least *action*. We formulate this dynamics estimation problem on the target system as a special case of the SOT problem and call it *the Lagrangian-Schrödinger Bridge (LSB) problem*. The position of the LSB problem in the SOT problem is clarified in Appendices A.3 and B.1.

**Definition 3.1** (LSB problem).

$$
\begin{aligned}
\underset{\mathbf{f}, \mathbf{g}}{\text{minimize}} \quad & \int_{t_0}^{t_1} \int_{\mathbb{R}^d} L(t, \mathbf{x}, \mathbf{f}(\mathbf{x}, t)) \, \mathrm{d}\rho_t(\mathbf{x}; \mathbf{f}, \mathbf{g}) \, \mathrm{d}t, \\
\text{subject to} \quad & \mathrm{d}\mathbf{X}_t = \mathbf{f}(\mathbf{X}_t, t) \, \mathrm{d}t + \mathbf{g}(\mathbf{X}_t, t) \, \mathrm{d}\mathbf{W}_t, \\
& \mathbf{X}_0 \sim \rho_{t_0} = \mu_0, \ \mathbf{X}_1 \sim \rho_{t_1} = \mu_1,
\end{aligned}
\tag{6}
$$

where $\rho_t$ is the law of the random variable $\mathbf{X}_t$, depending on the functions $\mathbf{f}$ and $\mathbf{g}$.

The LSB problem is the problem of exploring the sample paths that minimize *action* defined by the time integral of the Lagrangian, given the distributions at the two endpoints. The stochastic movement of samples and diffusion phenomena are explicitly modeled by using SDEs. The diffusion coefficient $\mathbf{g}$ is also optimized together with the drift $\mathbf{f}$ to reveal the effect of noise in real environments.

### 3.2    NEURAL LAGRANGIAN SCHRÖDINGER BRIDGE

In the setting of our paper, it is difficult to solve the LSB problem because the exact endpoint constraints $\mu_0$ and $\mu_1$ are unknown and only samples from them are available. Therefore, we propose a practical solution method for the LSB problem using neural SDEs (Eq. (3)) with regularization. We propose to approximate the LSB problem by learning neural SDEs with the gradients of the loss (See Appendix B.2 for theoretical justification and connections to existing works.):

$$
\underset{\theta, \phi}{\text{minimize}} \quad \mathbb{D}(\mu_1, \rho_{t_1}) + \mathcal{R}_e(\theta; t_0, t_1) + \mathcal{R}_h(\theta, \phi; t_0, t_1),
\tag{7}
$$

$$
\mathcal{R}_e(\theta; t_0, t_1) = \int_{t_0}^{t_1} \int_{\mathbb{R}^d} L(t, \mathbf{x}, \mathbf{f}_\theta(\mathbf{x}, t)) \, \mathrm{d}\rho_t(\mathbf{x}) \mathrm{d}t,
\tag{8}
$$

$$
\mathcal{R}_h(\theta, \phi; t_0, t_1) = \int_{t_0}^{t_1} \int_{\mathbb{R}^d} \left| \partial_t \Phi_\theta(\mathbf{x}, t) + \sum_{i,j=1}^{d} D_{i,j}(\mathbf{x}, t; \phi) \left[ \nabla_\mathbf{x}^2 \Phi_\theta \right]_{i,j} - H_\theta^*(\mathbf{x}, t) \right| \, \mathrm{d}\rho_t(\mathbf{x}) \mathrm{d}t,
$$

$$
H_\theta^*(\mathbf{x}, t) := H(t, \mathbf{x}, -\nabla_\mathbf{x} \Phi_\theta(\mathbf{x}, t)) = \langle -\nabla_\mathbf{x} \Phi_\theta(\mathbf{x}, t), \mathbf{f}_\theta(\mathbf{x}, t) \rangle - L(t, \mathbf{x}, \mathbf{f}_\theta(\mathbf{x}, t)),
\tag{9}
$$

where $\mathbb{D}$ is the distribution discrepancy measure, $D_{i,j}(\cdot; \phi)$ is the entry in the $i$-th row and $j$-th column of the diffusion coefficient matrix $\mathbf{D}_\phi = \frac{1}{2}\mathbf{g}_\phi \mathbf{g}_\phi^\top$, $H$ is the Hamiltonian defined by $H(t, \mathbf{x}, \mathbf{z}) := \langle \mathbf{z}, \mathbf{f}_\theta \rangle - L(t, \mathbf{x}, \mathbf{f}_\theta)$, and $\Phi_\theta$ is the potential function satisfying $\mathbf{f}_\theta = \nabla_\mathbf{z} H(t, \mathbf{x}, -\nabla_\mathbf{x}\Phi_\theta(\mathbf{x}, t))$.

We briefly explain the action cost $\mathcal{R}_e$ and the HJB regularization $\mathcal{R}_h$. Equation 7 is actually computed by Eq. (11). First, we relax the constraint at time $t_1$, *i.e.* $\mathcal{R}_e + \mathbb{D}(\mu_1, \rho_{t_1})$, by using Wasserstein distance (Bunne et al., 2021; Hashimoto et al., 2016), KL-divergence (Finlay et al., 2020; Onken et al., 2021; Tong et al., 2020), or a combination of these (Lavenant et al., 2021) for the discrepancy measure $\mathbb{D}$. This objective function can be reinterpreted with $\mathbb{D}(\mu_1, \rho_{t_1})$ as the data-fitting term and the action cost $\mathcal{R}_e$ as the regularization. Second, we exploit SOT theory by incorporating further structure into the modeling. Similar to OT-Flow (Onken et al., 2021), we model the drift functions by using the potential function $\mathbf{f}_\theta = \nabla_\mathbf{z} H(t, \mathbf{x}, -\nabla_\mathbf{x}\Phi_\theta(\mathbf{x}, t))$ and encourage the potential function $\Phi_\theta$ to satisfy the (stochastic) HJB equation (Yong & Zhou, 1999) by minimizing the PDE loss $\mathcal{R}_h$. The relation between the drift and the potential function is an analogue of Hamilton's equations of motion. The HJB equation represents Bellman's principle of optimality in continuous-time optimization.

### 3.3 Examples of Lagrangian in Neural Lagrangian Schrödinger Bridge

In this section, we provide the three Lagrangian examples of the NLSB and their use cases. The examples demonstrate that the Lagrangian design allows a variety of prior knowledge to be reflected in the sample trajectories. See Appendix E.5 for more examples.

**Potential-free system.** Without a specific external force on the individual sample, *the principle of least action* typically indicates that when a sample moves from point A to point B, it tries to minimize energy by reducing the travel distance. This means that the drift is encouraged to be straight and the stochastic movement to be small, *i.e.*, the Lagrangian is formulated by $L(t, \mathbf{x}, \mathbf{u}) = \frac{1}{2}||\mathbf{u}||^2$, and the drift function is given as $\mathbf{f}_\theta = -\nabla_\mathbf{x}\Phi_\theta(\mathbf{x}, t)$.

**Cellular system.** Tong et al. (2020) proposed to introduce the prior knowledge of manifold geometry and local velocity arrows such as RNA-velocity for modeling cellular systems. In the NLSB, these prior knowledge can be handled consistently by designing the Lagrangian. First, we introduce a density-based penalty to constrain the sample trajectories on the data manifold. We estimate the density function $U(\mathbf{x}, t)$ from the data, *e.g.*, by using Gaussian mixture models (GMM) and add it to the Lagrangian. Next, we redefine the velocity regularization, which is formulated as cosine similarity maximization by Tong et al. (2020), as a squared error minimization and add it to the Lagrangian as well. Therefore, the Lagrangian for the cellular system is defined by

$$L(t, \mathbf{x}, \mathbf{u}) = \underbrace{\frac{1}{2}||\mathbf{u}(\mathbf{x}, t)||^2}_{\text{Energy}} - \underbrace{U(\mathbf{x}, t)}_{\text{Density}} + \underbrace{\frac{1}{2}||\mathbf{u}(\mathbf{x}, t) - \mathbf{v}(\mathbf{x}, t)||^2}_{\text{Velocity}}, \tag{10}$$

where $\mathbf{v}(\mathbf{x}, t)$ is the reference velocity of the cell at the position $\mathbf{x}$ and the time $t$. The drift function is obtained by $\mathbf{f}_\theta = \frac{1}{2}(\mathbf{v}(\mathbf{x}, t) - \nabla_\mathbf{x}\Phi_\theta(\mathbf{x}, t))$.

**Random dynamical system.** Let $U(\mathbf{x})$ be the potential energy of the system, $\mathbf{R}$ be the mass matrix, which is symmetric and $L(t, \mathbf{x}, \mathbf{u}) = \frac{1}{2}\mathbf{u}^\top \mathbf{R}\mathbf{u} - U(\mathbf{x})$ be the Lagrangian in the random dynamical system. The drift function is given by $\mathbf{f}_\theta = -\mathbf{R}^{-1}\nabla_\mathbf{x}\Phi_\theta(\mathbf{x}, t)$. Then, the individual samples are encouraged to follow a stochastic analogue of *the equations of motion* of the Newtonian mechanics. In practice, the potential function can be used to roughly incorporate information such as obstacles and regions where samples are not likely to exist. Practical examples are shown in Figs. 1, 12 and 13.

## 4 Implementation of Neural Lagrangian Schrödinger Bridge

### 4.1 Training for Neural Lagrangian Schrödinger Bridge

In this section, we describe a practical learning method for neural SDEs by minimizing loss in Eq. (7) on the training data, as described in Algorithm 1. First, for the data, only individual samples from a cross-sectional population with no alignment across time steps at $K$ separate time points are available. Let $T = \{t_0, \ldots, t_{K-1}\}$ be a set of time points and denote the data set at time $t_i$ as $\mathcal{X}_{t_i}$. Next, we describe how we calculate the loss in Eq. (7) using the training data $\{\mathcal{X}_{t_i}\}_{t_i \in T}$. We compute

the distribution discrepancy using the L2-Sinkhorn divergence $\overline{\mathcal{W}}_\epsilon$ between the observed data and the predicted sample at all observation points in $T$ except the initial point at time $t_0$. The Sinkhorn divergence can efficiently approximate the Wasserstein distance and solve the entropic bias problem when using the Sinkhorn algorithm, *i.e.* $\mu = \nu \Leftrightarrow \overline{\mathcal{W}}_\epsilon(\mu, \nu) = 0$. The prediction sample is obtained by numerically simulating neural SDE from the training data at one previous time point by a standard SDE solver such as the Euler-Maruyama method. The second and third terms, $\mathcal{R}_e$ and $\mathcal{R}_h$, are also approximated on the sample paths obtained from the numerical simulations of SDE. Therefore, the computational cost of empirical $\mathcal{R}_e$ and $\mathcal{R}_h$ is not high and can be further accelerated by the model architecture described in the next section. To summarize, we simultaneously solve $K - 1$ approximated LSB problems by minimizing the subsequent loss for the model parameters $\theta$, $\phi$.

$$\ell(\theta, \phi) = \sum_{t_k \in T \backslash t_0} \overline{\mathcal{W}}_\epsilon(\mu_k, \rho_{t_k}^{\theta, \phi}) + \lambda_e(t_{k-1}, t_k)\hat{\mathcal{R}}_e^\theta(t_{k-1}, t_k) + \lambda_h(t_{k-1}, t_k)\hat{\mathcal{R}}_h^{\theta, \phi}(t_{k-1}, t_k), \quad (11)$$

where $\mu_k$, $\rho_{t_k}^{\theta, \phi}$ are the ground-truth and predicted probability measures at time $t_k$ expressed by data samples, respectively. $\hat{\mathcal{R}}_e^\theta(t_{k-1}, t_k)$ and $\hat{\mathcal{R}}_h^{\theta, \phi}(t_{k-1}, t_k)$ are empirical quantities computed on the simulated sample paths from $t_{k-1}$ to $t_k$ with data $\forall \mathbf{x}(t_{k-1}) \in \mathcal{X}_{t_{k-1}}$ as the initial value. The weight coefficients $\lambda_e(t_{k-1}, t_k)$ and $\lambda_h(t_{k-1}, t_k)$ are tuned for each interval $[t_{k-1}, t_k]$ respectively.

## 4.2 Model Architecture Selection for Speedup

We adopt the model of the potential function $\Phi_\theta$ proposed by OT-Flow (Onken et al., 2021) because it has two advantages in our framework. First, it can compute the gradient $\nabla_\mathbf{x}\Phi$ explicitly, which enables us to calculate the drift function easily. Second, the model is designed for the fast and exact computation of the diagonal component of the potential function's Hessian. Moreover, when we assume that the diffusion model's output is a diagonal matrix, *i.e.* $\mathbf{g}_\phi(\mathbf{x}, t) \in \mathbb{R}^{d \times d}$ and $\mathbf{g}_{i,j} = 0$ ($i \neq j$), the function $\mathbf{D}_\phi$ is also a diagonal matrix, and the $\sum_{i,j}$ in the second term of $\mathcal{R}_h$ shown in Eq. (9) turns into the sum of the diagonal components only. Combining these two tricks enables speeding up the computation of $\mathcal{R}_h$, which requires the expensive computation of the Hessian of the potential function $\Phi$. While this model architecture trick was originally used for speeding up the computation of the Jacobian term in neural ODE maximum likelihood training, we propose for the first time to use it as a technique to speed up the computation of $\mathcal{R}_h$ in neural SDE. We use a two-layer fully connected NN for the diffusion function. We also adopt the device used by Kidger et al. (2021) in which the activation function $\tanh$ is used after the final layer to prevent the output of the diffusion function from becoming excessively large. See Appendix C for details on the model of $\Phi$.

## 5 Experiments

We evaluated our methods on two datasets. First, we used artificial synthetic data generated from one-dimensional SDEs, where the predicted trajectory and uncertainty can be compared with the ground-truth and easily evaluated by visualization. We set the Lagrangian for the potential-free system, *i.e.* $L(t, \mathbf{x}, \mathbf{u}) = \frac{1}{2}||\mathbf{u}||^2$. Second, we used the evolution of single-cell populations obtained from a developing human embryo system. We used the Lagrangian for the cellular system and compared several combinations of the regularization terms. In Tables 1 and 2, "E" is the energy term, "D" is the density term, and "V" is the velocity term. The density term $U(\mathbf{x}, t)$ is the log-likelihood function of the data estimated by GMM. We compared our methods against standard neural SDE, OT-Flow (Onken et al., 2021), TrajectoryNet (Tong et al., 2020), IPF with GP (Vargas et al., 2021) and NN (De Bortoli et al., 2021), and SB-FBSDE (Chen et al., 2021a). We trained the standard neural SDE using only the Sinkhorn divergence. The velocity model of TrajectoryNet includes the concatsquash layers used in Grathwohl et al. (2018). The base models of OT-Flow and TrajectoryNet were trained with the standard neural ODEs scheme. +OT represents a model trained with the OT-based regularization defined by Eqs. (1) and (2). We used only $\tilde{\mathcal{R}}_e$ for TrajectoryNet; we used both $\tilde{\mathcal{R}}_e$ and $\tilde{\mathcal{R}}_h$ for OT-Flow. We set the interval-dependent coefficients $\tilde{\lambda}_e$, $\tilde{\lambda}_h$ for the OT-based regularization as well as Eq. (11). The drift model of IPF (GP) was changed to sparse GP from vanilla GP (Vargas et al., 2021) to save computation cost. The drift model of IPF (NN) and SB-FBSDE are the same networks as the NLSB for a fair comparison. The diffusion coefficients of IPF and SB-FBSDE were tuned as hyperparameters. See Appendix E for more details on hyperparameters.

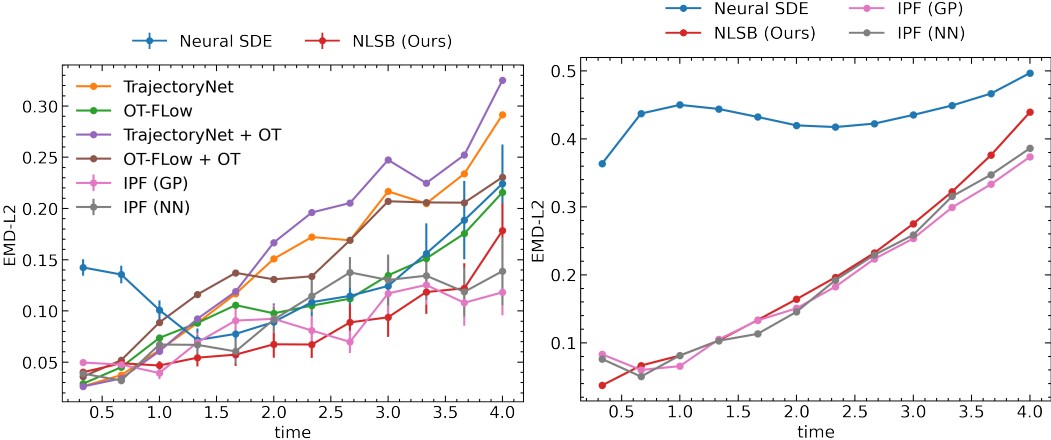

(a) MDD at 12 time points on synthetic OU process data.  (b) CDD at 12 time points on synthetic OU process data.

Figure 3: Numerical evaluation on synthetic OU process data. All MDD and CDD values were computed between the ground-truth and the estimated samples within generated trajectories all-step ahead from initial samples $\mathbf{x}(t_0)$.

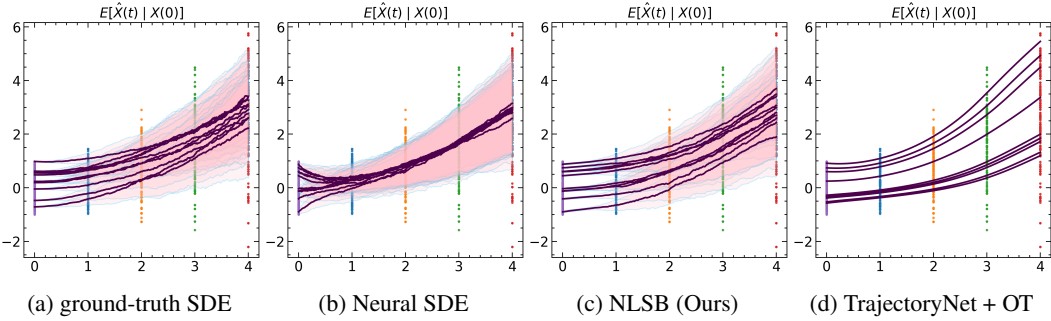

(a) ground-truth SDE     (b) Neural SDE     (c) NLSB (Ours)     (d) TrajectoryNet + OT

Figure 4: 1D OU process data and predictions. The five colored point clouds in the background are the ground-truth data given at each time point. The pink area and the light blue line are the one-sigma empirical confidence intervals and their boundaries for each trajectory, respectively. All trajectories were generated by all-step prediction from the initial samples at the time $t = 0$.

## 5.1 SYNTHETIC POPULATION DYNAMICS: ORNSTEIN–UHLENBECK PROCESS

**Data**. We used a time-dependent one-dimensional Ornstein–Uhlenbeck (OU) process defined by:

$$\mathrm{d}X_t = (\mu t - \theta X_t)\,\mathrm{d}t + \left(\frac{2t\sigma}{t_{K-1}}\right)\,\mathrm{d}W_t,$$

where $\mu = 0.4$, $\theta = 0.1$, $\sigma = 0.8$, and $t_{K-1} = 4$. First, we simulated the several trajectories from $t = 0$ to $4$ and then extracted only the data at the time of $T = \{0, 1, 2, 3, 4\}$ as snapshots for training. We generated 2048 and 512 samples for each time point as training and validation data, respectively.

**Performance metrics**. We evaluated the estimation performance of the dynamics in the time interval $[t_0, t_{K-1}]$ by using two metrics on test data: marginal distribution discrepancy (MDD) and conditional distribution discrepancy (CDD).

A smaller MDD between $\mu_t$ and $\rho_t$ calculated with the Wasserstein-2 distance indicates better prediction of population-level dynamics at time $t$. We calculated MDD at 12 equally spaced time points and the square root of the earth mover's distance with $L^2$ cost (EMD-L2) between 1000 samples generated from the ground-truth SDE and predicted by the model. When evaluating the SDE-based method, we ran 100 simulations from the same initial values, computing the MDD value each time and computing their mean and variance. A smaller CDD between $\mu_{\mathbf{x}(t)|\mathbf{x}(t_0)}$ and $\rho_{\mathbf{x}(t)|\mathbf{x}(t_0)}$

using the Wasserstein-2 distance indicates better prediction of the time evolution of the initial sample $\mathbf{x}(t_0)$. In short, it is a metric for evaluating the time evolution of at the individual sample level. In the actual CDD calculation, we first prepared 1000 samples $\mathbf{x}(t_0)$ at the initial time point. We then generated 100 trajectories from each initial sample by the trained model and the ground-truth SDE, and calculated the EMD-L2 for the samples from $\mu_{\mathbf{x}(t)|\mathbf{x}(t_0)}$ and $\rho_{\mathbf{x}(t)|\mathbf{x}(t_0)}$ at 12 equally spaced time points.

**Results**. The evaluation results are shown in Fig. 3, and the visualization of trajectories is shown in Fig. 4. Figure 3 shows that NLSB and IPF outperform neural SDE and is comparable to other ODE-based methods in estimating populations with small variance. In contrast, the SDE-based methods outperform ODE-based methods when estimating populations with a large variance. That indicates that NLSB and IPF can estimate population-level dynamics even when the population variance is large or small. Furthermore, NLSB and IPF have a smaller CDD value than neural SDE. Figure 4b shows that the average behavior of samples $\mathbb{E}[X(t)|X(0)]$ estimated by neural SDE is different from that of the ground-truth SDE (see Fig. 4a), especially in the interval $[0, 1]$. In contrast, the predictions by NLSB and IPF in Figs. 6c to 6e are much closer to the ground-truth. These results show that the prior knowledge of the potential-free system helps to estimate the sample-level dynamics. See Appendix E.2 for further results and analysis.

## 5.2 SINGLE-CELL POPULATION DYNAMICS

**Data**. We evaluated on embryoid body scRNA-seq data (Moon et al., 2019). This data shows the differentiation of human embryonic stem cells from embryoid bodies into diverse cell lineages, including mesoderm, endoderm, neuroectoderm, and neural crest, over 27 days. During this period, cells were collected at five different snapshots ($t_0$: day 0 to 3, $t_1$: day 6 to 9, $t_2$: day 12 to 15, $t_3$: day 18 to 21, $t_4$: day 24 to 27). The collected cells were then measured by scRNAseq, filtered at the quality control stage, and mapped to a low-dimensional feature space using a principal component analysis (PCA). For details, see Appendix E.2 in (Tong et al., 2020). We split the dataset into train, validation($\sim 8.5\%$) and test data ($\sim 15\%$).

**Performance metrics**. Unlike the experiment described in Section 5.1, there are no ground-truth trajectories in the real data. Thus, MDD can be calculated only at the time of observation, and CDD between the ground truth and predicted trajectories cannot be calculated. To evaluate the sample-level dynamics, the model was trained on the full data and the data without only one intermediate snapshot, respectively. We then calculated CDD between the predicted trajectories by those models. Let $\mathcal{D}_{-t_k}$ be the training data without a snapshot at time $t_k$. Larger CDD between them indicates the prediction of the sample-level dynamics is not robust to the exclusion of intermediate snapshots, representing poorer performance in interpolating the sample-level dynamics. When evaluating the SDE-based methods, we calculated the mean and standard deviation of 100 MDD scores. All performance metrics are calculated on the test data.

**Results**. Table 1 and Figure 5 show that the NLSB can predict population-level dynamics with better performance and can be trained in a shorter time against all existing ODE-based methods as the data become higher dimensional. In particular, the SDE-based methods significantly outperform the ODE-based ones in predicting the transitions where the samples from $t_1$ to $t_2$ and from $t_3$ to $t_4$ are highly diffuse, indicating that the explicit modeling of diffusion is effective. (see Fig. 10 in Appendix E). Overall, the standard deviation of the MDD values is smaller for NLSB than for neural SDE, demonstrating less variation in the approximation accuracy of the marginal distribution. Table 2 shows that NLSB estimates the sample-level dynamics robustly with and without population at the intermediate time point than neural SDE with some exceptions. Especially, energy regularization is the most stable and effective. This result suggests that the LSB-based regularization helps interpolate the sample-level dynamics. See Appendix E.3 for further results and analysis.

## 6 DISCUSSION AND CONCLUSION

In this work, we proposed a novel framework for estimating population dynamics that reflect prior knowledge about the target system. Unlike existing methods with OT (Schiebinger et al., 2019; Yang & Uhler, 2018), or CNF (Tong et al., 2020) for a biological system, we explicitly modeled the diffusion phenomena by using SDEs for the samples with stochastic behavior. This allowed us to

Table 1: Evaluation results for population-level dynamics on five-dimensional (5D) PCA space at time of observation for scRNA-seq data. The MDD value at $t_k$ is computed between the ground-truth and the samples predicted from the previous ground-truth samples at $t_{k-1}$ for each $k = 1, 2, 3$ and $4$.

| MDD (EMD-L2) $\downarrow$ | $t_1$ | $t_2$ | $t_3$ | $t_4$ |
|---|---|---|---|---|
| NLSB (E) | $0.71 \pm 0.020$ | $0.86 \pm 0.027$ | $0.83 \pm 0.016$ | $\mathbf{0.79} \pm 0.012$ |
| NLSB (D) | $0.67 \pm 0.017$ | $0.90 \pm 0.029$ | $0.87 \pm 0.018$ | $\mathbf{0.79} \pm 0.016$ |
| NLSB (V) | $0.70 \pm 0.023$ | $0.89 \pm 0.030$ | $0.83 \pm 0.022$ | $0.81 \pm 0.019$ |
| NLSB (E+D+V) | $0.68 \pm 0.016$ | $0.84 \pm 0.030$ | $\mathbf{0.81} \pm 0.018$ | $\mathbf{0.79} \pm 0.017$ |
| Neural SDE | $0.69 \pm 0.020$ | $0.91 \pm 0.029$ | $0.85 \pm 0.025$ | $0.81 \pm 0.017$ |
| OT-Flow | $0.83$ | $1.10$ | $1.07$ | $1.05$ |
| OT-Flow + OT | $0.85$ | $1.05$ | $1.09$ | $1.00$ |
| TrajectoryNet | $0.73$ | $1.06$ | $0.90$ | $1.01$ |
| TrajectoryNet + OT | $0.76$ | $1.05$ | $0.88$ | $1.10$ |
| IPF (GP) | $0.70 \pm 0.015$ | $1.04 \pm 0.041$ | $0.94 \pm 0.029$ | $0.98 \pm 0.033$ |
| IPF (NN) | $0.73 \pm 0.019$ | $0.89 \pm 0.030$ | $0.84 \pm 0.019$ | $0.83 \pm 0.020$ |
| SB-FBSDE | $\mathbf{0.56} \pm 0.010$ | $\mathbf{0.80} \pm 0.017$ | $1.00 \pm 0.019$ | $1.00 \pm 0.010$ |

Table 2: Mean value of CDD on 5D PCA space evaluated at 7 equally spaced time points within time period $[t_{k-1}, t_{k+1}]$ around excluded time point $t_k$ for each $k = 1, 2$ and $3$. The CDD value at the time $t \in [t_{k-1}, t_{k+1}]$ was computed between the two groups of predicted samples generated from the samples at $t_{k-1}$ using the model trained on all data and the data $\mathcal{D}_{-t_k}$.

| Mean CDD $\downarrow$ | $[t_0, t_2]$ | $[t_1, t_3]$ | $[t_2, t_4]$ |
|---|---|---|---|
| NLSB (E) | $0.88$ | $\mathbf{0.72}$ | $0.79$ |
| NLSB (D) | $1.64$ | $0.76$ | $\mathbf{0.77}$ |
| NLSB (V) | $1.15$ | $0.82$ | $0.86$ |
| NLSB (E+D+V) | $0.96$ | $0.83$ | $0.84$ |
| Neural SDE | $1.36$ | $0.85$ | $0.87$ |
| IPF (GP) | $0.97$ | $1.03$ | $1.06$ |
| IPF (NN) | $0.90$ | $0.95$ | $0.97$ |
| SB-FBSDE | $\mathbf{0.84}$ | $1.06$ | $1.49$ |

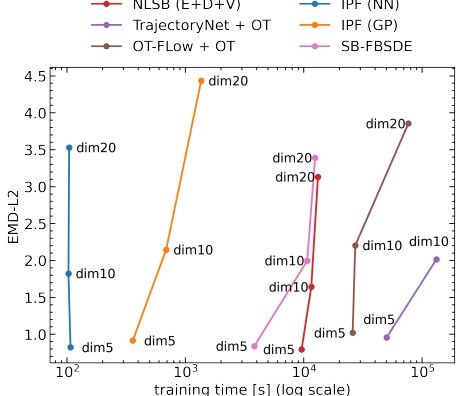

Figure 5: Relationship between data dimension, performance, and training time. The x-axis is the learning time until sufficient convergence. The y-axis is the mean value of MDD over time points $t_1$ to $t_4$, representing population-level performance.

handle the uncertainty of the trajectory (Fig. 9) and to successfully capture the diffuse transitions (Table 1 and Fig. 10). In contrast, Vargas et al. (2021) and Bunne et al. (2022) estimated the sample trajectories of biological systems using the SDE solution to the SB problem. Vargas et al. (2021) proposed GP-based IPF to solve the SB problem. Bunne et al. (2022) proposed GSB-Flow, in which two SB problems are solved sequentially. Compared with these methods, our method handled a wider class of SDEs, and the diffusion function of the SDE was learned from the data using a backpropagation. In addition, designing the Lagrangian enables us to flexibly incorporate prior knowledge about the target system into the model. Our Lagrangian-based regularization also treated the biological constraints proposed by Tong et al. (2020) and Maoutsa & Opper (2021) in a unified manner. We demonstrated that NLSB can efficiently estimate the population-level dynamics with better performance than existing methods even for high-dimensional data and that the prior knowledge introduced by the Lagrangian is useful to estimate the sample-level dynamics. Our method is limited in that it cannot model reaction phenomena such as cell birth and death, and restriction to the diagonal diffusion matrix (Section 4.2) may cause the model to be less expressive. Future work includes developing methods that can handle reaction phenomena with advection and diffusion.

ACKNOWLEDGMENTS

We thank the lab members for their discussions and inspiration for our research.

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

---

**Algorithm 1** Training of NLSB

---

**Input:** Dataset $\{\mathcal{X}_t\}_{t \in T}$, potential function $\mathbf{\Phi}_\theta$, diffusion function $\mathbf{g}_\phi$, Lagrangian $L$, regularization coefficients $\lambda_e$, $\lambda_h$.

   **while** $\theta$, $\phi$ have not converged **do**

      Set loss $\mathcal{L} = 0$.

     **for** $k \leftarrow 0$ to $K - 2$ **do**

        Sample mini-batch $\mathbf{x}^{(k)}$ from $\mathcal{X}_{t_k}$. Set $\mathbf{y}_{t_k} = \mathbf{x}^{(k)}$.

        Numerically solve augmented SDE $d\mathbf{s}_t = \tilde{\mathbf{f}}_\theta(\mathbf{s}_t, t)\,dt + \tilde{\mathbf{g}}_\phi(\mathbf{s}_t, t)\,d\mathbf{W}_t$ from $t_k$ to $t_{k+1}$.

$$
\tilde{\mathbf{f}}_\theta(\mathbf{s}_t, t) = \begin{bmatrix} \nabla_{\mathbf{z}} H(t, \mathbf{y}_t, -\nabla\Phi_\theta(\mathbf{y}_t, t)) \\ L(t, \mathbf{y}_t, \nabla_{\mathbf{z}} H(t, \mathbf{y}_t, -\nabla\Phi_\theta(\mathbf{y}_t, t))) \\ \left| \partial_t \Phi_\theta(\mathbf{y}_t, t) + \sum_{i,j} D_{i,j}(\mathbf{y}_t, t)\nabla^2\Phi_\theta(\mathbf{y}_t, t) + H_\theta^*(\mathbf{y}_t, t) \right| \end{bmatrix},
$$

$$
\tilde{\mathbf{g}}_\phi(\mathbf{s}_t, t) = \begin{bmatrix} \mathbf{g}_\phi(\mathbf{y}_t, t) & & \\ & 0 & \\ & & 0 \end{bmatrix}, \quad \mathbf{s}_t = \begin{bmatrix} \mathbf{y}_t \\ \hat{\mathcal{R}}_e(t_k, t) \\ \hat{\mathcal{R}}_h(t_k, t) \end{bmatrix}, \quad \mathbf{s}_{t_k} = \begin{bmatrix} \mathbf{y}_{t_k} \\ 0 \\ 0 \end{bmatrix}.
$$

        Sample mini-batch $\mathbf{x}^{(k+1)}$ from $\mathcal{X}_{t_{k+1}}$ and calculate the loss function $\ell(t_k, t_{k+1})$.

$$
\ell(t_k, t_{k+1}) = \overline{\mathcal{W}}_\epsilon(\mathbf{y}_{t_{k+1}}, \mathbf{x}^{(k+1)}) + \lambda_e(t_k, t_{k+1})\hat{\mathcal{R}}_e(t_k, t_{k+1}) + \lambda_h(t_k, t_{k+1})\hat{\mathcal{R}}_h(t_k, t_{k+1}),
$$

$$
\mathbf{s}_{t_{k+1}} = \begin{bmatrix} \mathbf{y}_{t_{k+1}} \\ \hat{\mathcal{R}}_e^\theta(t_k, t_{k+1}) \\ \hat{\mathcal{R}}_h^{\theta,\phi}(t_k, t_{k+1}) \end{bmatrix}.
$$

        Accumulate loss $\mathcal{L} \leftarrow \mathcal{L} + \ell(t_k, t_{k+1})$.

     Update $\theta$, $\phi$ with gradient $\nabla_\theta \mathcal{L}$ and $\nabla_\phi \mathcal{L}$, respectively.

**Output:** $\Phi_\theta, \mathbf{g}_\phi$

---

# A  THEORY OF STOCHASTIC OPTIMAL TRANSPORT

The SOT problem is the problem of finding a stochastic process that minimizes the expected cost under fixed marginal distributions at several time points. The SOT problem is a stochastic analog of the OT problem, especially related to the dynamic formulation of the OT problem introduced by Benamou & Brenier (2000). It is also considered as a stochastic optimal control (SOC) problem with additional terminal constraint. Furthermore, the classical Schrödinger Bridge (SB) problem is a special case of the SOT problem and is related to Nelson's stochastic mechanics, a reformulation of quantum mechanics using diffusion processes. An overview of the relationships among these problems is illustrated in Fig. 2.

We first explain the dynamic formulation of the OT problem in Appendix A.1, the SB problem in Appendix A.2 and then describe the SOT problem given two fixed marginal distributions as a generalization of the SB problem in Appendix A.3. Next, we discuss the LSB problem, which is the main focus of our paper among the SOT problems in Appendix B.1, and finally, we discuss the relationship between NLSB and OT-Flow in Appendix B.2.

## A.1  DYNAMIC FORMULATION OF OPTIMAL TRANSPORT PROBLEM

Benamou & Brenier (2000) redefined the OT problem with a quadratic cost in a continuum mechanics framework. They introduced a time parameter $t \in [0, 1]$ and considered the transport process from $\mu$ to $\nu$ as the advection in the time interval $[0, 1]$. The OT problem was then reformulated as a minimization problem with respect to the time-varying velocity vector field $v_t$ and the time evolution of the distribution $p_t$ advected by $v_t$.

**Theorem A.1** (Brenier-Benamou formulation (Eularian formalism); (Benamou & Brenier, 2000))**.**

$$\mathcal{W}_2(\mu, \nu)^2 = \inf_{(\mathbf{v}, p_t)} \int_0^1 \int_{\mathbb{R}^d} \|\mathbf{v}(\mathbf{x}, t)\|^2 p_t(\mathbf{x}) \, \mathrm{d}\mathbf{x} \mathrm{d}t, \tag{12}$$

$$\text{subject to} \quad \partial_t p_t = -\operatorname{div}(p_t \mathbf{v}), \ p_0 = p, \ p_1 = q, \tag{13}$$

*where $p$ and $q$ are the densities of probability measures $\mu$ and $\nu$, respectively.*

The first condition in Eq. (13) is known as the continuity equation and represents the conservation of mass in time evolution. We can consider an equivalent formulation of Eq. (12) by introducing Lagrangian coordinates $\mathbf{X}(t, \mathbf{x})$. Lagrangian coordinates $\mathbf{X}(t, \mathbf{x})$ represent the position at time $t$ of the particle whose initial position is $\mathbf{x}$, *i.e.* $\mathbf{X}(0, \mathbf{x}) = \mathbf{x}$.

**Theorem A.2** (Brenier-Benamou formulation (Lagrangian formalism); (Benamou & Brenier, 2000))**.**

$$\mathcal{W}_2(\mu, \nu)^2 = \inf_{\mathbf{v}} \int_0^1 \int_{\mathbb{R}^d} \|\mathbf{v}(\mathbf{X}(t, \mathbf{x}), t)\|^2 p_0(\mathbf{x}) \, \mathrm{d}\mathbf{x} \mathrm{d}t, \tag{14}$$

$$\text{subject to} \quad \frac{\mathrm{d}\mathbf{X}(t, \cdot)}{\mathrm{d}t} = \mathbf{v}(\mathbf{X}(t, \cdot), t), \ p_0 = p, \ p_1 = q.$$

The right-hand side of Eq. (14) can be viewed as a problem of finding the shortest path (a.k.a. geodesic) for each particle in the sense of Euclidean space between probability distributions specified at times $t = 0, 1$. In continuum mechanics, Lagrangian formalism (Eq. (14)) describes the motion of each individual particle, while Eulerian formalism (Eq. (12)) focuses on the global property of all particles.

Benamou & Brenier (2000) also showed the optimality conditions of the dynamic formulation. They introduced the Lagrangian multiplier of the constraint of Eq. (13) and obtained the saddle point conditions using the variational method.

**Theorem A.3** (Optimality conditions for the dynamic formulation; (Benamou & Brenier, 2000))**.** *There exists a space-time dependent potential function $\Phi \colon \mathbb{R}^d \times [0, 1] \mapsto \mathbb{R}$ which satisfies:*

$$\mathbf{v}^*(\mathbf{X}(t, \cdot), t) = -\nabla_{\mathbf{x}} \Phi(\mathbf{X}(t, \cdot), t), \tag{15}$$

$$\partial_t \Phi(\mathbf{x}, t) - \left\{ \langle -\nabla_{\mathbf{x}} \Phi(\mathbf{x}, t), \mathbf{v}^*(\mathbf{x}, t) \rangle - \frac{1}{2} \|\mathbf{v}^*(\mathbf{x}, t)\|^2 \right\} = 0, \tag{16}$$

$$\Phi_0(\mathbf{x}) = f^*(\mathbf{x}), \ \Phi_1(\mathbf{y}) = -g^*(\mathbf{y}), \tag{17}$$

*where $f^*$ and $g^*$ are Kantorovich potentials. Equation 15 is Hamilton's equation of motion with the Hamiltonian defined by $H(\mathbf{p}, \mathbf{x}) := \sup_{\mathbf{v}} \left\{ \langle \mathbf{p}, \mathbf{v}(\mathbf{x}, t) \rangle - \frac{1}{2} \|\mathbf{v}(\mathbf{x}, t)\|^2 \right\}$ and the momentum as $\mathbf{p} := -\nabla_{\mathbf{x}} \Phi$ and equation 16 is Hamilton-Jacob-Bellman (HJB) equation.*

Equations 15 and 17 indicate that the particle, whose initial position is $\mathbf{x}_0 \in \mathcal{X}$, moves straight ahead at the constant velocity $\mathbf{v}_0 = -\nabla_{\mathbf{x}} \Phi_0(\mathbf{x}_0) = -\nabla_{\mathbf{x}} f^*(\mathbf{x}_0)$ during $t \in [0, 1]$. Even without using the variational method, Hamilton's equation of motion (Eq. (15)) and the HJB equation (Eq. (16)) can be also derived from the Bellman's principle of optimality. The existence of a potential function satisfying the equation 15 is guaranteed from the Pontryagin Maximum Principle (Evans, 1983; 2010).

## A.2  SCHRÖDINGER BRIDGE

**Definition A.4** (SB problem; (Jamison, 1975))**.** Let $\Omega = C([0, 1], \mathbb{R}^d)$ be the space of $\mathbb{R}^d$-valued continuous functions on time interval $[0, 1]$. Denote by $\mathcal{P}(\Omega)$ the probability measures space on the path pace $\Omega$. The SB problem is defined by

$$\min_{\mathbb{Q} \in \mathcal{P}(\Omega)} D_{\mathrm{KL}}(\mathbb{Q} \| \mathbb{P}), \qquad \text{subject to} \quad \mathbb{Q}_0 \sim \mu_0, \ \mathbb{Q}_1 \sim \mu_1, \tag{18}$$

where $\mu_0, \nu_1 \in \mathcal{P}(\mathbb{R}^d)$ are the probability measures at the time 0 and 1, respectively, and the relative entropy $D_{\mathrm{KL}} = \int \log \left( \frac{\mathrm{d}\mathbb{Q}}{\mathrm{d}\mathbb{P}} \right) \mathrm{d}\mathbb{Q}$ if $\mathbb{Q} \ll \mathbb{P}$, and $D_{\mathrm{KL}} = \infty$ otherwise.

The case of no prior dynamics is considered classically, *i.e.*, the reference path measure $\mathbb{P}$ is the Brownian diffusion SDE $d\mathbf{X}_t = \sqrt{2\epsilon}\, d\mathbf{W}_t$, where $\mathbf{W}_t$ is the standard Wiener process. We refer to this setting as the classical SB problem, following the reference (Caluya & Halder, 2021). Chen et al. (2021b) derived the optimality condition for problem Eq. (18), which is characterized by the forward and backward time-harmonic equations.

**Theorem A.5.** *Let $\Psi(\mathbf{x}, t)$ and $\hat{\Psi}(\mathbf{x}, t)$ be the solutions to the following PDEs:*

$$\begin{cases} \frac{\partial \Psi}{\partial t} = -\epsilon \Delta \Psi \\ \frac{\partial \widehat{\Psi}}{\partial t} = \epsilon \Delta \widehat{\Psi} \end{cases} \quad s.t. \ \ \Psi(\cdot, 0)\widehat{\Psi}(\cdot, 0) = p_0, \ \Psi(\cdot, 1)\widehat{\Psi}(\cdot, 1) = p_1 \ , \tag{19}$$

*where $p_0$ and $p_T$ are the probability density of $\mu_0$ and $\mu_T$, respectively. Then, the solution to the SB problem (Eq. (18)) can be described by the following forward or backward SDE:*

$$\begin{aligned} d\mathbf{X}_t &= 2\epsilon \nabla_{\boldsymbol{x}} \log \Psi\left(\mathbf{X}_t, t\right)\, dt + \sqrt{2\epsilon}\, d\mathbf{W}_t, \quad \mathbf{X}_0 \sim \mu_0, \\ d\mathbf{X}_t &= -2\epsilon \nabla_{\boldsymbol{x}} \log \widehat{\Psi}\left(\mathbf{X}_t, t\right)\, dt + \sqrt{2\epsilon}\, d\mathbf{W}_t, \quad \mathbf{X}_1 \sim \mu_1. \end{aligned} \tag{20}$$

Dai Pra (1991) considered a dynamic formulation of the classic SB problem (Eq. (18)) by interpreting it as a SOC problem with the additional terminal constraint.

**Theorem A.6** (Dynamic formulation; (Dai Pra, 1991)). *Let $\mathbf{f}^* = 2\epsilon \nabla_{\boldsymbol{x}} \log \Psi$, where $\Psi$ satisfies the SB optimality (Eq. (19)). Then, $\mathbf{f}^*$ is the minimizer of the following optimization problem:*

$$V_{S,\epsilon}(\mu_0, \mu_1) := \inf_{\mathbf{f}} \mathbb{E}\left[\int_0^1 \frac{1}{2} \|\mathbf{f}\left(\mathbf{X}_t, t\right)\|^2\, dt\right], \tag{21}$$

$$\text{subject to} \quad d\mathbf{X}_t = \mathbf{f}\left(\mathbf{X}_t, t\right)\, dt + \sqrt{2\epsilon}\, d\mathbf{W}_t, \ \mathbf{X}_0 \sim \mu_0, \ \mathbf{X}_1 \sim \mu_1.$$

Léonard (2013) formulated the SB problem in Eq. (18) into a variational SOC problem equivalent to the above problem (Eq. (21)).

**Theorem A.7** (Dynamic formulation (Eularian formalism); (Léonard, 2013)). *Let $\mathbf{f}^*(\mathbf{x}, t) = \nabla_{\boldsymbol{x}} \log \mathbb{E}\left[\Psi(\mathbf{X}_T, T) \mid \mathbf{X}_t = \mathbf{x}\right]$, where $\Psi$ satisfies the SB optimality (Eq. (19)). Then, $\mathbf{f}^*$ is the minimizer of the following optimization problem:*

$$\mathrm{v}_{S,\epsilon}(\mu_0, \mu_1) := \inf_{(\mathbf{f}, \rho_t)} \int_0^1 \int_{\mathbb{R}^d} \frac{1}{2} \|\mathbf{f}\left(\mathbf{x}, t\right)\|^2\, d\rho_t(\mathbf{x}) dt, \tag{22}$$

$$\text{subject to} \quad \partial_t p_t = -\operatorname{div}(p_t \mathbf{f}) + \epsilon \Delta p_t, \ \rho_0 = \mu_0, \ \rho_1 = \mu_1, \tag{23}$$

*where $p_t$ is the probability density of the probability measure $\rho_t$.*

The first condition in Eq. (23) is Fokker-Planck (FP) equation. The two versions of the dynamic formulation (Eq. (21) and Eq. (22)) are equivalent, *i.e.* $V_{S,\epsilon}(\mu_0, \mu_1) = \mathrm{v}_{S,\epsilon}(\mu_0, \mu_1)$. Furthermore, the dynamic solution of the OT problem (Eq. (15)) is obtained as the zero-noise limit (Mikami, 2004; Léonard, 2012) of the classical SB problem.

**Theorem A.8** (Zero-noise limit of the classical SB problem; (Mikami, 2004)). *Let $X^\epsilon(t)$ be the solution of the SB problem (Eq. (20)). Suppose that $\mu_0, \mu_1 \in \mathcal{P}(\mathbb{R}^d)$ have finite second moments and the density function $p_0(x) := \mu_0(dx)/dx$ exists. Then, the following holds*

$$\lim_{\epsilon \to 0} \epsilon V_{S,\epsilon}(\mu_0, \mu_1) = \mathcal{W}_2(\mu_0, \mu_1)^2,$$

*and there exists a convex function $\varphi$ satisfying*

$$\lim_{\epsilon \to 0} \mathbb{E}\left[\sup_{0 \le t \le 1} |\mathbf{X}^\epsilon(t) - (\mathbf{X}_0 + t(\nabla_{\mathbf{x}}\varphi(\mathbf{X}_0) - \mathbf{X}_0))|^2\right] = 0.$$

*The map $\nabla_{\mathbf{x}}\varphi$ is the optimal transport map of the OT problem with a quadratic cost.*

The connection between the classical SB problem and the dynamic formulation of the OT problem with Fisher information regularization (Chen et al., 2016) is also well-known. Finally, we introduce a simpler variant of the SB problem for which closed-form solution exists.

**Theorem A.9** (Gaussian SB problem; (Bunne et al., 2022)). *Let $\mathbb{Q}^*$ be the solution of the Gaussian SB problem defined by*

$$\min_{\mathbb{Q} \in \mathcal{P}(\Omega)} D_{\mathrm{KL}}(\mathbb{Q}\|\mathbb{P}), \qquad subject\ to \quad \mathbb{Q}(0) \sim \mathcal{N}_0,\ \mathbb{Q}(1) \sim \mathcal{N}_1,$$

*where $\mathcal{N}_0 = \mathcal{N}(\mu_0, \Sigma_0)$ and $\mathcal{N}_1 = \mathcal{N}(\mu_1, \Sigma_1)$ are Gaussian distributions. The reference path measure $\mathbb{P}$ is described by the linear SDE as follows.*

$$\mathrm{d}\mathbf{X}_t = (c(t)\mathbf{X}_t + \mathbf{f}(t))\ \mathrm{d}t + g(t)\ \mathrm{d}\mathbf{W}_t, \qquad \mathbf{X}_0 \sim \mathcal{N}_0,$$

*where $c\colon [0,1] \mapsto \mathbb{R}$, $\mathbf{f}\colon [0,1] \mapsto \mathbb{R}^d$, $g\colon [0,1] \mapsto \mathbb{R}_+$ are smooth functions. We define the following notation from (Bunne et al., 2022):*

$$\tau_t := \exp\left(\int_0^t c(s)\ \mathrm{d}s\right)$$

$$D_\sigma := \left(4\Sigma_0^{\frac{1}{2}}\Sigma_T\Sigma_0^{\frac{1}{2}} + \sigma^4 I\right)^{\frac{1}{2}}, \quad C_\sigma := \frac{1}{2}\left(\Sigma_0^{\frac{1}{2}} D_\sigma \Sigma_0^{-\frac{1}{2}} - \sigma^2 I\right)$$

$$r_t := \frac{\kappa(t,T)}{\kappa(T,T)}, \quad \bar{r}_t := \tau_t - r_t\tau_T, \quad \sigma_\star := \sqrt{\tau_T^{-1}\kappa(T,T)}$$

$$\zeta(t) := \tau_t \int_0^t \tau_s^{-1}\alpha(s)\mathrm{d}s, \quad \rho_t := \frac{\int_0^t \tau_s^{-2}g^2(s)\mathrm{d}s}{\int_0^T \tau_s^{-2}g^2(s)\mathrm{d}s}$$

$$P_t := \dot{r}_t\left(r_t\Sigma_T + \bar{r}_t C_{\sigma_\star}\right), \quad Q_t := -\dot{\bar{r}}_t\left(\bar{r}_t\Sigma_0 + r_t C_{\sigma_\star}\right)$$

$$S_t := P_t - Q_t^\top + \left[c(t)\kappa(t,t)\left(1 - \rho_t\right) - g^2(t)\rho_t\right] I$$

$$\mu_t^* := \bar{r}_t\mu_0 + r_t\mu_T + \zeta(t) - r_t\zeta(T)$$

$$\Sigma_t^* := \bar{r}_t^2\Sigma_0 + r_t^2\Sigma_T + r_t\bar{r}_t\left(C_{\sigma_\star} + C_{\sigma_\star}^\top\right) + \kappa(t,t)\left(1 - \rho_t\right) I$$

*Then, the solution $\mathbb{Q}^*$ is a Markov Gaussian process where the marginal $\mathbf{X}_t^* \sim \mathcal{N}(\mu_t^\star, \Sigma_t^\star)$, and follows the following SDE:*

$$\mathrm{d}\mathbf{X}_t^* = S_t^\top \Sigma_t^{*-1}(\mathbf{X}_t^* - \mu_t^*)\ \mathrm{d}t + g(t)\ \mathrm{d}\mathbf{W}_t.$$

### A.3 STOCHASTIC OPTIMAL TRANSPORT WITH TWO ENDPOINT MARGINALS

Mikami (2008) generalized the OT problem and defined the SOT problem as a random mechanics problem determined by *the principle of least action*. The SOT problem with the endpoint marginals fixed to $\mu_0$ and $\mu_1$ is given by the following.

**Definition A.10** (SOT problem; (Mikami, 2021)). Let $L$ be a continuous function, the Lagrangian and let $\mathbf{u} \mapsto L(t, \mathbf{x}, \mathbf{u})$ be convex. The SOT problem with two endpoint marginals is defined by

$$V(\mu_0, \mu_1) := \inf_{\mathbf{X} \in \mathscr{A}} \mathbb{E}\left[\int_0^1 L\left(t, \mathbf{X}_t; \mathbf{f}_{\mathbf{X}}(\mathbf{X}, t)\right)\ \mathrm{d}t\right], \quad subject\ to \quad \mathbf{X}_0 \sim \mu_0,\ \mathbf{X}_1 \sim \mu_1, \quad (24)$$

where $\mathscr{A}$ is the set of all $\mathbb{R}^d$-valued, continuous semimartingales $\{\mathbf{X}_t\}_{0 \le t \le 1}$ on a complete filtered probability space such that there exists a Borel measurable drift function $\mathbf{f}_{\mathbf{X}}(\mathbf{X}, t)$ for which satisfies the following conditions:

1. $\omega \mapsto \mathbf{f}_{\mathbf{X}}(t, \omega)$ is Borel-measureable for all $t$.

2. $\mathbf{X}_t = \mathbf{X}_0 + \int_0^t \mathbf{f}_{\mathbf{X}}(\mathbf{X}, s)\ \mathrm{d}s + \int_0^t \mathbf{g}(\mathbf{X}_s, s)\ \mathrm{d}\mathbf{W}_s,\ 0 \le t \le 1$

3. $\mathbb{E}\left[\int_0^1 (|\mathbf{f}_{\mathbf{X}}(\mathbf{X}_t, s)| + |\mathbf{g}(\mathbf{X}_t, t)|^2)\ \mathrm{d}t\right] < \infty$

**Definition A.11** (SOT problem for marginal flows; (Mikami, 2008)).

$$\mathrm{v}(\mu_0, \mu_1) := \inf_{\mathbf{f} \in \mathbf{A}\left(\{\rho_t\}_{0 \le t \le 1}\right)} \int_0^1 \int_{\mathbb{R}^d} L\left(t, \mathbf{x}, \mathbf{f}(\mathbf{x}, t)\right)\ \mathrm{d}\rho_t(\mathbf{x})\ \mathrm{d}t \tag{25}$$

$$subject\ to \quad \rho_0 = \mu_0,\ \rho_1 = \mu_1, \tag{26}$$

$$\mathbf{A}\left(\{\rho_t\}_{0 \le t \le 1}\right) := \left\{\mathbf{f}(\mathbf{x}, t)\ \middle|\ \partial_t p_t = -\operatorname{div}(p_t\mathbf{f}) + \sum_{i,j=1}^d \frac{\partial^2}{\partial x_i \partial x_j}\left[D_{i,j}(\mathbf{x}, t)p_t(\mathbf{x})\right]\right\}, \tag{27}$$

where $p_0$ and $p_1$ are the densities of probability measures $\rho_0$ and $\rho_1$, respectively.

The minimizer of the SOT problem for marginal flows (Eq. (25)) is obtained by $\mathbf{f}^*(\mathbf{x}, t) = \mathbb{E}\left[\mathbf{f_X}(\mathbf{X}, t) | (t, \mathbf{X}_t = \mathbf{x})\right]$. We introduce assumptions from (Mikami, 2021).

(A-1) $L \in C^1(\mathbb{R}^d \times \mathbb{R}^d; [0, \infty])$. $\mathbf{u} \mapsto L(t, \mathbf{x}, \mathbf{u})$ is strictly convex. $L(t, \mathbf{x}, \mathbf{u})/(1 + L(t, \mathbf{y}, \mathbf{u}))$ and $|\nabla_{\mathbf{x}} L(t, \mathbf{x}, \mathbf{u})|/(1 + L(t, \mathbf{x}, \mathbf{u}))$ are bounded on $t \in [0, 1]$ and $\mathbf{x}, \mathbf{y}, \mathbf{u} \in \mathbb{R}^d$. $\sup_{\mathbf{x} \in \mathbb{R}^d} |\nabla_{\mathbf{x}} L(t, \mathbf{x}, \mathbf{u})|^1$ is locally bounded. $\lim_{|\mathbf{u}| \to \infty} \inf L(t, \mathbf{x}, \mathbf{u})/|\mathbf{u}| = \infty$.

(A-2) $\nabla_{\mathbf{u}}^2 L(t, \mathbf{x}, \mathbf{u})$ is bounded uniformly nondegenerate on $[0, 1] \times \mathbb{R}^d \times \mathbb{R}^d$.

**Theorem A.12** (Trevisan's Superposition Principle; (Trevisan, 2016)). *Assume that there exists* $\mathbf{f} : \mathbb{R}^d \times [0, 1] \mapsto \mathbb{R}^d$ *and* $\{\rho_t\}_{0 \le t \le 1} \subset \mathcal{P}(\mathbb{R}^d)$ *such that* $\mathbf{f}$ *satisfies the FP equation, i.e.* $\mathbf{f} \in \mathbf{A}\left(\{\rho_t\}_{0 \le t \le 1}\right)$. *Then, there exists a semimartingale* $\{\mathbf{X}_t\}_{0 \le t \le 1}$ *for which the following holds:*

$$\mathbf{X}_t = \mathbf{X}_0 + \int_0^t \mathbf{f}(\mathbf{X}_s, s) \, \mathrm{d}s + \int_0^t \mathbf{g}(\mathbf{X}_s, s) \, \mathrm{d}\mathbf{W}_t,$$
$$\mathbf{X}_t \sim \rho_t \quad (0 \le t \le 1)$$

From Theorem A.12, it can be easily shown that $V(\mu_0, \mu_1) = \mathrm{v}(\mu_0, \mu_1)$ and there exist minimizers $\mathbf{X}^*$ of the SOT problem (Eq. (24)) for which $\mathbf{f_{X^*}}(\mathbf{X}^*, t) = \mathbf{f}^*(\mathbf{X}_t^*, t)$.

**Theorem A.13** (Duality theorem; (Mikami, 2021)). *Suppose that (A-1) holds. Then, for any* $\mu_0, \mu_1 \in \mathcal{P}(\mathbb{R}^d)$,

$$V(\mu_0, \mu_1) = \mathrm{v}(\mu_0, \mu_1) = \sup_{f \in C_b^\infty(\mathbb{R}^d)} \left\{ \int_{\mathbb{R}^d} \Phi(\mathbf{x}, 0; f) \, \mathrm{d}\mu_0(\mathbf{x}) - \int_{\mathbb{R}^d} f(\mathbf{x}) \, \mathrm{d}\mu_1(\mathbf{x}) \right\}, \quad (28)$$

*where* $C_b^\infty(\mathbb{R}^d)$ *is the set of all infinitely differentiable functions on* $\mathbb{R}^d$, *which have bounded continuous derivative and* $\Phi(\mathbf{x}, t; f)$ *is the viscosity solution to the HJB equation:*

$$\partial_t \Phi(\mathbf{x}, t; f) + \sum_{i,j=1}^d D_{i,j}(\mathbf{x}, t) \left[\nabla_{\mathbf{x}}^2 \Phi(\mathbf{x}, t; f)\right]_{i,j} - H\left(t, \mathbf{x}, -\nabla_{\mathbf{x}} \Phi(\mathbf{x}, t; f)\right) = 0,$$

$$\Phi(\mathbf{x}, 1; f) = f(\mathbf{x}).$$

*The Hamiltonian* $H$ *is defined by* $H(t, \mathbf{x}, \mathbf{z}) := \sup_{\mathbf{u}} \{\langle \mathbf{z}, \mathbf{u} \rangle - L(t, \mathbf{x}, \mathbf{u})\}$.

**Theorem A.14** ((Mikami & Thieullen, 2006)). *Suppose that (A-1) and (A-2) hold and* $V(\mu_0, \mu_1)$ *is finite. Then, there exists a minimizer* $\mathbf{X}^* \in \mathscr{A}$ *of* $V(\mu_0, \mu_1)$ *given by*

$$\mathbf{X}_t^* = \mathbf{X}_0^* + \int_0^t \mathbf{f}^*(\mathbf{X}_s^*, s) \, \mathrm{d}s + \int_0^t \mathbf{g}(\mathbf{X}_s^*, s) \, \mathrm{d}\mathbf{W}_s,$$

*For any maximizing sequence* $\{\Phi_n\}_{n \ge 1}$ *of Eq. (28), there exists a subsequence* $\{n_k\}_{k \ge 1}$ *such that*
$$\mathbf{f}^*(\mathbf{X}_s, s) = \lim_{k \to \infty} \nabla_{\mathbf{z}} H(s, \mathbf{X}_s, -\nabla_{\mathbf{x}} \Phi_{n_k}(\mathbf{X}_s, s)).$$

## B  SCHRÖDINGER BRIDGE AND GENERATIVE MODELING

The theory of the SB problem is mostly mature as shown in Appendix A.2, but scalable numerical methods for estimating SB are still actively studied. In particular, there have been many recent studies (De Bortoli et al., 2021; Wang et al., 2021; Vargas et al., 2021; Chen et al., 2021a; Bunne et al., 2022), which uses the SB as a process for generating data. These studies other than (Wang et al., 2021) proposed methods to learn SDE solutions of the SB problem between the prior distribution and the target data distribution. These methods combine the classical multi-stage optimization method called Iterative proportional fitting (IPF) (Fortet, 1940; Kullback, 1968; Ruschendorf, 1995) for solving the SB with machine learning methods for optimization of subproblems. In IPF, the following subproblems are solved alternately and iteratively. The reference path measure $\mathbb{P}$ is set to the initial measure $\mathbb{Q}_*^{(0)}$.

$$\mathbb{R}_*^{(i)} = \underset{\mathbb{P} \in \mathcal{P}(\Omega)}{\operatorname{argmin}} D_{\mathrm{KL}}(\mathbb{R} \| \mathbb{Q}_*^{(i-1)}), \qquad \text{subject to} \quad \mathbb{R}(1) \sim \mu_1, \tag{29}$$

$$\mathbb{Q}_*^{(i)} = \underset{\mathbb{Q} \in \mathcal{P}(\Omega)}{\operatorname{argmin}} D_{\mathrm{KL}}(\mathbb{Q} \| \mathbb{R}_*^{(i)}), \qquad \text{subject to} \quad \mathbb{Q}(0) \sim \mu_0, \tag{30}$$

where $\mathbb{R}(t)$, $\mathbb{Q}(t)$ are the probability measures at the time $t$ on the path measures $\mathbb{R}$ and $\mathbb{Q}$. The path measures $\mathbb{Q}_*^{(i)}$, $\mathbb{R}_*^{(i)}$ at the $i$-th step are simulated by the following forward-backward SDEs in Eqs. (31) and (32), respectively.

$$d\mathbf{X}_t = \mathbf{f}^{(i)}(\mathbf{X}_t, t)\,dt + \sqrt{2\epsilon}\,d\mathbf{W}_t, \qquad \mathbf{X}_0 \sim \mu_0, \tag{31}$$

$$d\mathbf{X}_t = \mathbf{b}^{(i)}(\mathbf{X}_t, t)\,dt + \sqrt{2\epsilon}\,d\mathbf{W}_t, \qquad \mathbf{X}_1 \sim \mu_1. \tag{32}$$

The convergence of IPF was proved in (Ruschendorf, 1995).

The sub-optimization problems (Eqs. (29) and (30)) are approached differently for each method. First, Vargas et al. (2021) and De Bortoli et al. (2021) proposed to solve them by mean-matching regression of the SDE drift function using Gaussian process (GP) and NN, respectively. They find the drift function of the SDEs that minimizes the following losses for some sampled time $t$.

$$\mathbf{b}_t^{(i)} = \underset{\mathbf{b}_t}{\operatorname{argmin}}\, \mathbb{E}_{\mathbf{X} \sim \mathbb{Q}_*^{(i-1)}} \left\| \mathbf{b}_t(\mathbf{X}_t) - \left(\mathbf{X}_t + \mathbf{f}_{t-\Delta t}^{(i-1)}(\mathbf{X}_{t-\Delta t}) - \mathbf{f}_{t-\Delta t}^{(i-1)}(\mathbf{X}_t)\right) \right\|,$$

$$\mathbf{f}_t^{(i)} = \underset{\mathbf{f}_t}{\operatorname{argmin}}\, \mathbb{E}_{\mathbf{X} \sim \mathbb{R}_*^{(i)}} \left\| \mathbf{f}_t(\mathbf{X}_t) - \left(\mathbf{X}_t + \mathbf{b}_{t+\Delta t}^{(i)}(\mathbf{X}_{t+\Delta t}) - \mathbf{b}_{t+\Delta t}^{(i)}(\mathbf{X}_t)\right) \right\|.$$

In contrast, Chen et al. (2021a) proposed to use the divergence-based losses as shown in Eq. (33). The divergence-based losses are a modified version of the approximate likelihood-maximization training of SGM for use in alternating optimization schemes.

$$\mathbf{v}^{(i)} = \underset{\mathbf{v}}{\operatorname{argmax}}\, \mathbb{E}_{\mathbf{X} \sim \mathbb{Q}_*^{(i-1)}} \left[ \int_0^1 \frac{1}{2} \|\mathbf{v}(\mathbf{X}_t, t)\|^2 + g\operatorname{div}_{\mathbf{x}}(\mathbf{v}) + \mathbf{u}^{(i-1)\top}\mathbf{v}\,dt \right],$$

$$\mathbf{u}^{(i)} = \underset{\mathbf{u}}{\operatorname{argmax}}\, \mathbb{E}_{\mathbf{X} \sim \mathbb{R}_*^{(i)}} \left[ \int_0^1 \frac{1}{2} \|\mathbf{u}(\mathbf{X}_t, t)\|^2 + g\operatorname{div}_{\mathbf{x}}(\mathbf{u}) + \mathbf{v}^{(i)\top}\mathbf{u}\,dt \right],$$

where $\mathbf{u}^{(i)}, \mathbf{v}^{(i)}$ are learnable drift terms of the forward-backward SDEs that redefines Eqs. (31) and (32) with fixed prior drift $\mathbf{f}_{\mathrm{prior}}$, simulating the path measures $\mathbb{Q}_*^{(i)}, \mathbb{R}_*^{(i)}$.

$$d\mathbf{X}_t = \left(\mathbf{f}_{\mathrm{prior}}(\mathbf{X}_t, t) + g(t)\mathbf{u}^{(i)}(\mathbf{X}_t, t)\right)\,dt + \sqrt{2\epsilon}\,d\mathbf{W}_t, \qquad \mathbf{X}_0 \sim \mu_0,$$
$$d\mathbf{X}_t = \left(\mathbf{f}_{\mathrm{prior}}(\mathbf{X}_t, t) - g(t)\mathbf{v}^{(i)}(\mathbf{X}_t, t)\right)\,dt + \sqrt{2\epsilon}\,d\mathbf{W}_t, \qquad \mathbf{X}_1 \sim \mu_1. \tag{33}$$

To estimate more complex dynamics, Bunne et al. (2022) proposed to solve a general SB problem (Eq. (18)) in which the solution of the Gaussian SB problem is used as a reference measure $\mathbb{P}$. As shown in Theorem A.9, the Gaussian SB problem has a closed-form solution, and the general SB problem is solved using the alternating optimization with divergence-based losses proposed by Chen et al. (2021a).

## B.1 Lagrangian Schrödinger Bridge Problem

We consider the LSB problem constrained by the FP equation corresponding to Ito SDE (Eq. (3)) over the Euclidean space $\mathbb{R}^d$.

**Definition B.1** (LSB problem constrained by the FP equation).

$$\mathcal{V}(\mu_0, \mu_1) := \inf_{(\mathbf{f}, \rho_t) \in \mathcal{S}} \int_0^1 \int_{\mathbb{R}^d} L(t, \mathbf{x}, \mathbf{f}(\mathbf{x}, t))\,d\rho_t(\mathbf{x})\,dt, \tag{34}$$

$$\text{subject to} \quad \rho_0 = \mu_0, \ \rho_1 = \mu_1, \tag{35}$$

$$\mathcal{S} := \left\{ (\mathbf{f}, \rho_t) \ \middle| \ \partial_t p_t = -\operatorname{div}(p_t \mathbf{f}) + \sum_{i,j=1}^d \frac{\partial^2}{\partial x_i \partial x_j}\left[D_{i,j}(\mathbf{x}, t; \mathbf{f}, \rho_t)p_t(\mathbf{x})\right] \right\},$$

The LSB problem in Eq. (34) is a more general problem that does not fix $D_{i,j}$ in the SOT problem for marginal flows (Eq. (25)). Thus, a solution to the LSB problem in Eq. (34) clearly exists on the basis of the existence of a solution to the SOT problem in Eq. (25).

We practically solve the following relaxed LSB problem, where the terminal constraint are replaced by the soft constraint.

**Definition B.2** (Relaxed LSB problem constrained by the FP equation).

$$\tilde{\mathcal{V}}(\mu_0, \mu_1) := \inf_{(\mathbf{f}, \rho_t) \in \mathcal{S}} \int_0^1 \int_{\mathbb{R}^d} L(t, \mathbf{x}, \mathbf{f}(\mathbf{x}, t)) \, \mathrm{d}\rho_t(\mathbf{x}) \, \mathrm{d}t + \int_{\mathbb{R}^d} G(\mathbf{x}) \, \mathrm{d}\rho_1(\mathbf{x}), \tag{36}$$

$$\text{subject to} \quad \rho_0 = \mu_0, \tag{37}$$

$$G(\mathbf{x}) := \frac{\delta}{\delta\rho_1} \mathbb{D}(\rho_1(\mathbf{x})|\mu_1(\mathbf{x})),$$

where $G$ is the terminal cost introduced by relaxing the constraint $\rho_1 = \mu_1$ and $\frac{\delta}{\delta\rho_1}$ is the variational derivative with respect to $\rho_1$.

We derive the optimality conditions for the LSB problems in Eqs. (34) and (36) using variational method in the following theorem. The derivation procedure is similar to that for the variational formulation of the SB problem by (Chen et al., 2021b).

**Theorem B.3** (Optimality conditions for the LSB problem). *There exists a space-time dependent potential function $\Phi \colon \mathbb{R}^d \times [0, 1] \mapsto \mathbb{R}$, which satisfies:*

$$\mathbf{f}^*(\mathbf{x}, t) = \nabla_{\mathbf{z}} H(t, \mathbf{x}, -\nabla_{\mathbf{x}} \Phi(\mathbf{x}, t)), \tag{38}$$

$$\partial_t \Phi(\mathbf{x}, t) + \sum_{i,j=1}^d D_{i,j}(\mathbf{x}, t; \mathbf{f}^*, \rho_t^*) \left[ \nabla_{\mathbf{x}}^2 \Phi(\mathbf{x}, t) \right]_{i,j} - H(t, \mathbf{x}, \mathbf{f}^*(\mathbf{x}, t)) = 0, \tag{39}$$

$$\Phi(\mathbf{x}, 1) = G(\mathbf{x}),$$

*where $(\mathbf{f}^*, \rho_t^*)$ is the minimizer of the relaxed LSB problem in Eq. (36).*

*The equation 38 and 39 are the optimality conditions for both the relaxed LSB problem in Eq. (36) and the LSB problem where $G(\mathbf{x}) = 0$ in Eq. (34).*

*Proof.* We derive the optimality conditions for the relaxed LSB problem as shown in Eq. (36) by reformulating it as a saddle point problem for $(p_t, \mathbf{m}_t) := (p_t, p_t \mathbf{f}_t)$. Let $\mathcal{L}$ be the the Lagrangian with the time-space-dependent Lagrange multiplier $\Phi(\mathbf{x}, t)$.

$$\mathcal{L}(p, \mathbf{m}, \Phi) := \int_0^1 \int_{\mathbb{R}^d} L\left(t, \mathbf{x}, \frac{\mathbf{m}_t}{p_t}\right) p_t(\mathbf{x}) \, \mathrm{d}\mathbf{x}\mathrm{d}t + \int_{\mathbb{R}^d} G(\mathbf{x}) p_1(\mathbf{x}) \, \mathrm{d}\mathbf{x}$$

$$- \int_0^1 \int_{\mathbb{R}^d} \Phi(\mathbf{x}, t) \left( \underbrace{\partial_t p_t}_{(a)} + \underbrace{\mathrm{div}(\mathbf{m}_t)}_{(b)} - \underbrace{\sum_{i,j=1}^d \frac{\partial^2}{\partial x_i \partial x_j} \left[ D_{i,j}(\mathbf{x}, t) p_t(\mathbf{x}) \right]}_{(c)} \right) \mathrm{d}\mathbf{x} \, \mathrm{d}t.$$

The term (a) is transformed by performing a partial integral over $t$:

$$\int_0^1 \int_{\mathbb{R}^d} \Phi(\mathbf{x}, t) \partial_t p_t(\mathbf{x}) \, \mathrm{d}\mathbf{x} \, \mathrm{d}t = \int_{\mathbb{R}^d} \Phi(\mathbf{x}, 1) p_1(\mathbf{x}) \, \mathrm{d}\mathbf{x} - \int_{\mathbb{R}^d} \Phi(\mathbf{x}, 0) p_0(\mathbf{x}) \, \mathrm{d}\mathbf{x}$$

$$- \int_0^1 \int_{\mathbb{R}^d} \partial_t \Phi(\mathbf{x}, t) p_t(\mathbf{x}) \, \mathrm{d}\mathbf{x} \, \mathrm{d}t.$$

The term (b) is simplified as follows.

$$\int_{\mathbb{R}^d} \Phi(\mathbf{x}, t) \, \mathrm{div}(\mathbf{m}_t) \, \mathrm{d}\mathbf{x} = \int_{\mathbb{R}^d} \mathrm{div}(\Phi \mathbf{m}_t) - \mathbf{m}_t(\mathbf{x})^\top \nabla_{\mathbf{x}} \Phi(\mathbf{x}, t) \, \mathrm{d}\mathbf{x}$$

$$= - \int_{\mathbb{R}^d} \mathbf{m}_t(\mathbf{x})^\top \nabla_{\mathbf{x}} \Phi(\mathbf{x}, t) \, \mathrm{d}\mathbf{x}.$$

The term (c) is transformed by performing a partial integral over $\mathbf{x}$:

$$\int_{\mathbb{R}^d} \sum_{i,j=1}^d \frac{\partial^2 D_{i,j}(\mathbf{x},t)p_t(\mathbf{x})}{\partial x_i \partial x_j} \Phi(\mathbf{x},t) \, \mathrm{d}\mathbf{x} = -\sum_{i,j=1}^d \int_{\mathbb{R}^d} \frac{\partial \left(D_{i,j}(\mathbf{x},t)p_t(\mathbf{x})\right)}{\partial x_j} \frac{\partial \Phi(\mathbf{x},t)}{\partial x_i} \, \mathrm{d}\mathbf{x}$$

$$= \sum_{i,j=1}^d \int_{\mathbb{R}^d} D_{i,j}(\mathbf{x},t)p_t(\mathbf{x}) \frac{\partial^2 \Phi(\mathbf{x},t)}{\partial x_i \partial x_j} \, \mathrm{d}\mathbf{x}$$

$$= \int_{\mathbb{R}^d} \left( \sum_{i,j=1}^d D_{i,j}(\mathbf{x},t) \left[\nabla_{\mathbf{x}}^2 \Phi(\mathbf{x},t)\right]_{i,j} \right) p_t(\mathbf{x}) \, \mathrm{d}\mathbf{x}.$$

Then, we can rewrite the LSB problem as

$$\inf_{p,\mathbf{m}} \sup_{\Phi} \mathcal{L}(p,\mathbf{m},\Phi), \tag{40}$$

$$\mathcal{L}(p,\mathbf{m},\Phi) = \int_0^1 \int_{\mathbb{R}^d} \left( L\left(t,\mathbf{x},\frac{\mathbf{m}_t}{p_t}\right) + \partial_t \Phi(\mathbf{x},t) + \sum_{i,j=1}^d D_{i,j} \left[\nabla_{\mathbf{x}}^2 \Phi(\mathbf{x},t)\right]_{i,j} \right) p_t(\mathbf{x}) \, \mathrm{d}\mathbf{x} \, \mathrm{d}t$$

$$+ \int_{\mathbb{R}^d} G(\mathbf{x})p_1(\mathbf{x}) \, \mathrm{d}\mathbf{x} + \int_0^1 \int_{\mathbb{R}^d} \mathbf{m}_t(\mathbf{x})^\top \nabla_{\mathbf{x}} \Phi(\mathbf{x},t) \, \mathrm{d}\mathbf{x} \, \mathrm{d}t$$

$$- \int_{\mathbb{R}^d} \Phi(\mathbf{x},1)p_1(\mathbf{x}) \, \mathrm{d}\mathbf{x} + \int_{\mathbb{R}^d} \Phi(\mathbf{x},0)p_0(\mathbf{x}) \, \mathrm{d}\mathbf{x}.$$

The saddle point $(p^*, \mathbf{m}^*, \Phi^*)$ of the problem in Eq. (40) satisfies the following conditions:

$$\partial_\Phi \mathcal{L}|_{(p^*,\mathbf{m}^*,\Phi^*)} = 0 \quad \Leftrightarrow \quad \partial_t p_t^* + \sum_{i=1}^d \frac{\partial}{\partial x_i} m_i^*(\mathbf{x},t) - \sum_{i,j=1}^d \frac{\partial^2}{\partial x_i \partial x_j} \left[D_{i,j}(\mathbf{x},t)p_t^*\right] = 0,$$

$$\partial_p \mathcal{L}|_{(p^*,\mathbf{m}^*,\Phi^*)} = 0 \quad \Leftrightarrow \quad \partial_t \Phi^*(\mathbf{x},t) + \sum_{i,j=1}^d D_{i,j}(\mathbf{x},t) \left[\nabla_{\mathbf{x}}^2 \Phi(\mathbf{x},t)\right]_{i,j} - H(t,\mathbf{x},\mathbf{z}_*) = 0,$$

$$\partial_\mathbf{m} \mathcal{L}|_{(p^*,\mathbf{m}^*,\Phi^*)} = 0 \quad \Leftrightarrow \quad \nabla_\mathbf{f} L(t,\mathbf{x},\mathbf{f}_*) = -\nabla_\mathbf{x} \Phi^*,$$

$$\partial_{p_1} \mathcal{L}|_{(p^*,\mathbf{m}^*,\Phi^*)} = 0 \quad \Leftrightarrow \quad \Phi^*(\mathbf{x},1) = G(\mathbf{x}),$$

where $\mathbf{z}_* := \nabla_\mathbf{u} L(t,\mathbf{x},\mathbf{u})|_{\mathbf{u}=\mathbf{f}^*}$ and the Hamiltonian is defined by $H(t,\mathbf{x},\mathbf{z}) := \sup_\mathbf{u} \{\langle \mathbf{z}, \mathbf{u} \rangle - L(t,\mathbf{x},\mathbf{u})\}$. The Lagrangian satisfies $L(t,\mathbf{x},\mathbf{u}) = \sup_\mathbf{z} \{\langle \mathbf{u}, \mathbf{z} \rangle - H(t,\mathbf{x},\mathbf{z})\}$.

Therefore, the optimal drift function is given by

$$\mathbf{f}^*(\mathbf{x},t) = \frac{\mathbf{m}^*}{p^*} = \nabla_\mathbf{z} H(t,\mathbf{x},\mathbf{z}_*) = \nabla_\mathbf{z} H(t,\mathbf{x},-\nabla_\mathbf{x}\Phi^*(\mathbf{x},t)). \tag{41}$$

The potential function $\Phi$ is the solution of the HJB equation:

$$\partial_t \Phi(\mathbf{x},t) + \sum_{i,j=1}^d D_{i,j}(\mathbf{x},t) \left[\nabla_{\mathbf{x}}^2 \Phi(\mathbf{x},t)\right]_{i,j} - H(t,\mathbf{x},\mathbf{f}^*(\mathbf{x},t)) = 0. \tag{42}$$

Equation 41 and 42 are also optimal conditions even for the LSB problem in Eq. (34) where the terminal constraint is strictly satisfied, *i.e.* $G(\mathbf{x}) = 0$. □

From Theorem A.12, we can show that there exists a semimartingale $\{\mathbf{X}_t\}_{0 \leq t \leq 1}$ for which the following holds:

$$\mathbf{X}_t = \mathbf{X}_0 + \int_0^t \mathbf{f}^*(\mathbf{X}_s,s) \, \mathrm{d}s + \int_0^t \mathbf{g}^*(\mathbf{X}_s,s) \, \mathrm{d}\mathbf{W}_t, \ \mathbf{X}_t \sim \rho_t^* \quad (0 \leq t \leq 1),$$

where $(\mathbf{f}^*, \rho_t^*)$ is the minimizer of the LSB problems in Eq. (34) or Eq. (36) and $\mathbf{g}^*$ is the diffusion function determined from the minimizer $(\mathbf{f}^*, \rho_t^*)$. Therefore, the LSB problems constrained by the FP equation (Eqs. (34) and (36)) and the original LSB problem defined by Eq. (6) are equivalent.

Note that we named Eq. (6) the LSB problem, as the SB problem is well-known in the machine learning field as the problem of finding the most likely stochastic process between sampled time points.

## B.2 Derivation of NLSB and its connection to OT-Flow

The loss function of the NLSB shown in Eq. (7) is the objective function of the relaxed LSB problem (Eq. (36)) with additional PDE loss applied to satisfy the optimality condition, HJB equation. Theorem B.3 justifies the minimization of HJB-PDE loss and a parameterization that strictly satisfies Eq. (38) during training when the terminal constraint is not satisfied, *i.e.* $G(\mathbf{x}) \neq 0$. For computational efficiency, we practically define the HJB-PDE loss $\mathcal{R}_h$ in a weak form as shown in Eq. (9) and evaluate $\mathcal{R}_h$ on the path of the simulated SDE.

NLSB and OT-Flow (Onken et al., 2021) have strong theoretical connections. NLSB is the practical solution to the LSB problem (Eq. (6)) relaxed by the distribution discrepancy measure $\mathbb{D}$ using neural SDE, while OT-Flow is the solution to the Brenier-Benamou formulation of the OT problem (Eq. (12)) relaxed by the KL-divergence using neural ODE. The optimality conditions for the LSB problem shown in Theorem B.3 are analogues of the optimality conditions for the Brenier-Benamou problem shown in Theorem A.3. In NLSB and OT-Flow, the potential function is modeled using NN and optimized by both action cost (Eqs. (1) and (8)) and HJB-PDE loss (Eqs. (1) and (9)) based on the respective optimality conditions. Theoretically, the LSB problem with the Lagrangian $L(t, \mathbf{x}, \mathbf{u}) = \frac{1}{2}\|\mathbf{u}\|^2$ and the diffusion function $\mathbf{g} = \sqrt{2}\epsilon$ is reduced to the classical SB problem (Eq. (21)) and the solution to the Brenier-Benamou problem is recovered from the zero-noise limit of the solution to the classical SB problem (Theorem A.8).

## C Model Architecture

The model structure of the potential function proposed in OT-Flow (Onken et al., 2021) is shown below.

$$\Phi(\mathbf{s}) = \mathbf{w}^\top N\left(\mathbf{s}; \{\mathbf{K}_i, \mathbf{b}_i\}_{0 \leq i \leq M}\right) + \frac{1}{2}\mathbf{s}^\top(\mathbf{A}^\top \mathbf{A})\mathbf{s} + \mathbf{b}^\top \mathbf{s} + c,$$
$$N\left(\mathbf{s}; \{\mathbf{K}_i, \mathbf{b}_i\}_{0 \leq i \leq M}\right) = \mathbf{u}_M,$$
$$\mathbf{u}_i = \mathbf{u}_{i-1} + h\sigma(\mathbf{K}_i\mathbf{u}_{i-1} + \mathbf{b}_i) \quad (1 \leq i \leq M), \ \mathbf{u}_0 = \sigma(\mathbf{K}_0\mathbf{s} + \mathbf{b}_0),$$

where $\mathbf{s} = (\mathbf{x}, t) \in \mathbb{R}^{d+1}$ is a input vector, $\mathbf{w}, \mathbf{K}_0 \in \mathbb{R}^{m \times (d+1)}, \mathbf{K}_i \ (1 \leq i \leq M) \in \mathbb{R}^{m \times m}, \mathbf{b}_i \ (0 \leq i \leq M) \in \mathbb{R}^{m \times m}, \mathbf{A}, \mathbf{b}$, and $c$ are learnable parameters, $m$ is the number of dimensions of the hidden representation vector, $h$ is a fixed step size, and the activation function is defined by $\sigma(\mathbf{x}) = \log(\exp(\mathbf{x}) + \exp(-\mathbf{x}))$.

The gradient of the potential function is described.

$$\nabla_\mathbf{s}\Phi(\mathbf{s}) = \nabla_\mathbf{s}N\left(\mathbf{s}; \{\mathbf{K}_i, \mathbf{b}_i\}_{0 \leq i \leq M}\right)\mathbf{w} + (\mathbf{A}^\top \mathbf{A})\mathbf{s} + \mathbf{b},$$
$$\nabla_\mathbf{s}N\left(\mathbf{s}; \{\mathbf{K}_i, \mathbf{b}_i\}_{0 \leq i \leq M}\right) = \mathbf{K}_0^\top \text{diag}(\sigma'(\mathbf{K}_0\mathbf{s} + \mathbf{b}_0))\mathbf{z}_1,$$
$$\mathbf{z}_i = \mathbf{z}_{i+1} + h\mathbf{K}_i^\top \text{diag}(\sigma'(\mathbf{K}_i\mathbf{u}_{i-1} + \mathbf{b}_i))\mathbf{z}_{i+1} \quad (1 \leq i \leq M), \ \mathbf{z}_{M+1} = \mathbf{1}.$$

We can write $\sigma'$ down as $\tanh$ since $\sigma$ is defined as above.

Finally, the diagonal components of the potential function's Hessian is shown below.

$$\nabla_\mathbf{s}^2\Phi(\mathbf{s}) = \nabla_\mathbf{s}(\mathbf{K}_0^\top \text{diag}(\sigma'(\mathbf{K}_0\mathbf{s} + \mathbf{b}_0))\mathbf{z}_1)$$
$$+ h\sum_{i=1}^M \nabla_\mathbf{s}\mathbf{u}_{i-1}\nabla_\mathbf{s}(\mathbf{K}_i^\top \text{diag}(\sigma'(\mathbf{K}_i\mathbf{u}_{i-1} + \mathbf{b}_i))\mathbf{z}_{i+1})\nabla_\mathbf{s}\mathbf{u}_{i-1}^\top$$
$$= \mathbf{K}_0^\top \text{diag}(\sigma''(\mathbf{K}_0\mathbf{s} + \mathbf{b}_0) \odot \mathbf{z}_1)\mathbf{K}_0$$
$$+ \nabla_\mathbf{s}\mathbf{u}_{i-1}\mathbf{K}_i^\top \text{diag}(\sigma''(\mathbf{K}_i\mathbf{u}_{i-1} + \mathbf{b}_i) \odot \mathbf{z}_{i+1})\mathbf{K}_i\nabla_\mathbf{s}\mathbf{u}_{i-1}^\top,$$

$$\left[\nabla_\mathbf{s}^2\Phi(\mathbf{s})\right]_{i,i} = \left[(\sigma''(\mathbf{K}_0\mathbf{s} + \mathbf{K}_0) \odot \mathbf{z}_1)^\top(\mathbf{K}_0 \odot \mathbf{K}_0)\right]_i$$
$$+ h\sum_{i=1}^M \left[(\sigma''(\mathbf{K}_i\mathbf{u}_{i-1} + \mathbf{b}_i) \odot \mathbf{z}_{i+1})^\top(\mathbf{K}_i\nabla_\mathbf{s}\mathbf{u}_{i-1}^\top \odot \mathbf{K}_i\nabla_\mathbf{s}\mathbf{u}_{i-1}^\top)\right]_i,$$

where $\odot$ is the element-wise product, $\nabla_{\mathbf{s}}\mathbf{u}_i$ can be obtained by using the following update equation:

$$\nabla_{\mathbf{s}}\mathbf{u}_i^\top \leftarrow \nabla_{\mathbf{s}}\mathbf{u}_{i-1}^\top + \mathrm{diag}(h\sigma'(\mathbf{K}_i\mathbf{u}_{i-1} + \mathbf{b}_i))\mathbf{K}_i\nabla_{\mathbf{s}}\mathbf{u}_{i-1}^\top.$$

The diagonal component of the potential function Hessian can be computed at a computation cost of $O(m^2 d)$. The complexity $O(m^2 d)$ indicates that there is a trade-off between the expressive power of the DNN and the computational cost of the Hessian.

OT-Flow modeled the potential function $\Phi_\theta$ that satisfies $\mathbf{f}_\theta = -\nabla_{\mathbf{x}}\Phi_\theta$ instead of directly modeling the velocity function $\mathbf{f}$ of the neural ODE. The continuous transformation from $\mathbf{x}(0)$ to $\mathbf{x}(1)$ is described by

$$\mathbf{x}(1) = \mathbf{x}(0) + \int_0^1 \mathbf{f}_\theta(\mathbf{x}(t), t)\,\mathrm{d}t = \mathbf{x}(0) + \int_0^1 -\nabla_{\mathbf{x}}\Phi_\theta(\mathbf{x}(t), t)\,\mathrm{d}t.$$

OT-Flow is trained by likelihood maximization in the well-known CNF framework. The likelihood computation is performed as follows.

$$p_1(\mathbf{x}(1)) = p_0(\mathbf{x}(0)) - \int_0^1 \mathrm{Tr}\left(\nabla_{\mathbf{x}}\mathbf{f}_\theta(\mathbf{x}(t), t)\right)\,\mathrm{d}t,$$

$$= p_0(\mathbf{x}(0)) - \int_0^1 \mathrm{Tr}\left(-\nabla_{\mathbf{x}}^2\Phi_\theta(\mathbf{x}(t), t)\right)\,\mathrm{d}t,$$

where $p_1$ is the density of the target distribution and $p_0$ is the density of the prior distribution, which is usually a Gaussian or Laplace distribution. The fast and exact computation of the diagonal component of the potential function's Hessian is useful for the computation of Appendix C.

In contrast, we propose for the first time to use this technique to speed up the computation of the HJB-PDE loss $\mathcal{R}_h$ in the NLSB under the assumption that the diffusion model's output is a diagonal matrix. The computed loss $\mathcal{R}_h$ is given by

$$\mathcal{R}_h(\theta, \phi; t_0, t_1) = \int_{t_0}^{t_1} \int_{\mathbb{R}^d} \left| \partial_t \Phi_\theta(\mathbf{x}, t) + \sum_{i=1}^d D_{i,i}(\mathbf{x}, t; \phi)\left[\nabla_{\mathbf{x}}^2\Phi_\theta\right]_{i,i} - H_\theta^*(\mathbf{x}, t) \right|\,\mathrm{d}\rho_t(\mathbf{x})\mathrm{d}t,$$

$$H_\theta^*(\mathbf{x}, t) = \langle -\nabla_{\mathbf{x}}\Phi_\theta(\mathbf{x}, t), \mathbf{f}_\theta(\mathbf{x}, t) \rangle - L(t, \mathbf{x}, \mathbf{f}_\theta(\mathbf{x}, t)).$$

## D   BENEFITS OF GENERALIZATION TO LAGRANGIAN

We first present the advantages of generalization by the Lagrangian. Next, we define the general form of the Lagrangian considered in this thesis and explain its generality theoretically. We also provide several examples of Lagrangians to demonstrate their applicability as models for a wide range of real-world systems.

The benefit of generalization by the Lagrangian is that the HJB equation and Hamilton's equation can be described in a unified manner using Lagrangian, independent of the coordinate system of the space on SDE. In other words, the loss function can always be easily derived once the Lagrangian is designed.

The general form of the Lagrangian considered in this paper is given by

$$L(t, \mathbf{x}, \mathbf{u}; \mathbf{R}, \mathbf{c}, \mathbf{v}, \mathbf{m}, U) = \frac{1}{2}(\mathbf{u} - \mathbf{v})^\top \mathbf{R}(\mathbf{u} - \mathbf{v}) + \mathbf{c}^\top(\mathbf{u} - \mathbf{m}) - U(\mathbf{x}, t), \tag{43}$$

where $\mathbf{R} \in \mathbb{R}^{d \times d}$ is the positive definite since the Lagrangian is convex with respect to $\mathbf{u}$, $U$ is the potential function defined from the prior knowledge on the target system. The optimal drift function is obtained by $\mathbf{f}_\theta(\mathbf{x}, t) = -2(\mathbf{R} + \mathbf{R}^\top)^{-1}(\nabla_{\mathbf{x}}\Phi_\theta(\mathbf{x}, t) + \mathbf{c}) + \mathbf{v}$.

In Lagrangian mechanics, deterministic Newtonian dynamical systems can be described by the principle of least action using the general Lagrangian (Eq. (43)), independent of the coordinate system. In contrast, since we are dealing with random dynamical system using SDEs, we can describe an even wider range of systems than Newtonian dynamical systems. The general Lagrangian (Eq. (43)) also includes as the special case the Lagrangian cost used in the optimal control (OC)

problems such as the linear-quadratic regulator (LQR). To show the generality of the Lagrangian of Eq. (43), we demonstrate that the Lagrangian in linearly transformed coordinates can also be written in the same form.

We define SDE on the coordinate-transformed state variable $\mathbf{x} \in \mathbb{R}^d$ from the observed variable $\mathbf{y} \in \mathbb{R}^p$.

$$
\begin{aligned}
\underset{\mathbf{f},\mathbf{g}}{\text{minimize}} \quad & \int_{t_0}^{t_1} \int_{\mathbb{R}^d} L(t, \mathbf{x}, \mathbf{f}(\mathbf{x}, t)) \, \mathrm{d}\rho_t(\mathbf{x}; \mathbf{f}, \mathbf{g}) \, \mathrm{d}t, \\
\text{subject to} \quad & \begin{cases} \mathrm{d}\mathbf{X}_t = \mathbf{f}(\mathbf{X}_t, t) \, \mathrm{d}t + \mathbf{g}(\mathbf{X}_t, t) \, \mathrm{d}\mathbf{W}_t, \\ \mathbf{y} = \mathbf{P}\mathbf{x}, \end{cases} \\
& \mathbf{Y}_0 = \mathbf{P}\mathbf{X}_0 \sim \rho_{t_0} = \mu_0, \ \mathbf{Y}_1 = \mathbf{P}\mathbf{X}_1 \sim \rho_{t_1} = \mu_1.
\end{aligned}
\tag{44}
$$

The linear coordinate transformation is given by $\mathbf{y} = \mathbf{P}\mathbf{x}$ represented by the projection matrix $\mathbf{P} \in \mathbb{R}^{p \times d}$. In particular, when we use the Wasserstein-2 distance $\mathcal{W}_2$ as the distribution discrepancy measure $\mathbb{D}$ in Eq. (11), the loss function of NLSB is almost invariant for the linear coordinate transformations.

**Regular matrix**. When the projection matrix $\mathbf{P} \in \mathbb{R}^{d \times d}$ is regular, the general Lagrangian defined on the observed variable space $\mathbf{y} \in \mathbb{R}^d$ is computed on the state variable space $\mathbf{x} \in \mathbb{R}^d$ as follows.

$$
\begin{aligned}
\mathcal{W}_2(\mu_1, \rho_{t_1}) &= \inf_{\pi} \int_{\mathbb{R}^d \times \mathbb{R}^d} \|\mathbf{y}_1 - \mathbf{y}_2\|^2 \, \mathrm{d}\pi(\mathbf{y}_1, \mathbf{y}_2), \\
&= \inf_{\tilde{\pi}} \int_{\mathbb{R}^d \times \mathbb{R}^d} \|\mathbf{P}\mathbf{x}_1 - \mathbf{P}\mathbf{x}_2\|^2 \, \mathrm{d}\tilde{\pi}(\mathbf{x}_1, \mathbf{x}_2), \\
&= \inf_{\tilde{\pi}} \int_{\mathbb{R}^d \times \mathbb{R}^d} (\mathbf{x}_1 - \mathbf{x}_2)^\top \mathbf{R} (\mathbf{x}_1 - \mathbf{x}_2) \, \mathrm{d}\tilde{\pi}(\mathbf{x}_1, \mathbf{x}_2), \\
& \hspace{-5em} L(t, \mathbf{y}, \mathbf{u_y}; \mathbf{R_y}, \mathbf{c_y}, \mathbf{v_y}, \mathbf{m_y}, U_{\mathbf{y}}) \\
&\hspace{-5em}= \frac{1}{2}(\mathbf{u_y} - \mathbf{v_y})^\top \mathbf{R_y}(\mathbf{u_y} - \mathbf{v_y}) + \mathbf{c_y}^\top(\mathbf{u_y} - \mathbf{m_y}) - U_{\mathbf{y}}(\mathbf{y}, t), \\
&\hspace{-5em}= \frac{1}{2}(\mathbf{u_x} - \mathbf{v_x})^\top \mathbf{P}^\top \mathbf{R_y} \mathbf{P}(\mathbf{u_x} - \mathbf{v_x}) + \mathbf{c_y}^\top \mathbf{P}(\mathbf{u_x} - \mathbf{m_x}) - U_{\mathbf{y}}(\mathbf{P}\mathbf{x}, t), \\
&\hspace{-5em}= \frac{1}{2}(\mathbf{u_x} - \mathbf{v_x})^\top \mathbf{R_x}(\mathbf{u_x} - \mathbf{v_x}) + \mathbf{c_x}^\top(\mathbf{u_x} - \mathbf{m_x}) - U_{\mathbf{x}}(\mathbf{x}, t), \\
&\hspace{-5em}= L(t, \mathbf{x}, \mathbf{u_x}; \mathbf{R_x}, \mathbf{c_x}, \mathbf{v_x}, \mathbf{m_x}, U_{\mathbf{x}}),
\end{aligned}
$$

where $\mathbf{R} = \mathbf{P}^\top \mathbf{P}$ and $\mathbf{R_x} = \mathbf{P}^\top \mathbf{R_y} \mathbf{P} \in \mathbb{R}^{d \times d}$ are guaranteed to be positive definite, $\mathbf{P}_\sharp^{-1}$ is the push-forward operator of the linear map represented by $\mathbf{P}^{-1}$, $\mathbf{c_x} = \mathbf{P}^\top \mathbf{c_y} \in \mathbb{R}^d$, and $U_{\mathbf{x}}(\mathbf{x}, t) = U_{\mathbf{y}}(\mathbf{P}\mathbf{y}, t)$ is the potential function. Especially when $\mathbf{P}$ is an orthogonal transformation, *i.e.* $\mathbf{P}^\top \mathbf{P} = \mathbf{I}$, then $\mathbf{R} = \mathbf{I}$ and $\mathbf{R_x} = \mathbf{R_y}$ hold.

The general Lagrangian has sufficient representational capacity to describe Lagrangians in coordinate systems that can be linearly transformed into each other in a unified manner. Furthermore, similar results are obtained for the PCA projection, which is an irregular matrix.

**PCA projection matrix**. We consider an inverse projection from the latent space $\mathbf{x}$ to the data space $\mathbf{y}$ as the projection matrix $\mathbf{P} \in \mathbb{R}^{p \times d}$ ($d \ll p$). The columns of the PCA projection matrix are orthogonal, *i.e.* $\mathbf{P}^\top \mathbf{P} = \mathbf{I}$. The cost functions defined on the observed variable space $\mathbf{y} \in \mathbb{R}^p$ is

efficiently computed on the low-dimensional space $\mathbf{x} \in \mathbb{R}^d$ as follows.

$$
\begin{aligned}
\mathcal{W}_2(\mu_1, \rho_{t_1}) &= \inf_\pi \int_{\mathbb{R}^p \times \mathbb{R}^p} \|\mathbf{y}_1 - \mathbf{y}_2\|^2 \, \mathrm{d}\pi(\mathbf{y}_1, \mathbf{y}_2), \\
&= \inf_{\tilde{\pi}} \int_{\mathbb{R}^d \times \mathbb{R}^d} \|\mathbf{P}\mathbf{x}_1 - \mathbf{P}\mathbf{x}_2\|^2 \, \mathrm{d}\tilde{\pi}(\mathbf{x}_1, \mathbf{x}_2), \\
&= \inf_{\tilde{\pi}} \int_{\mathbb{R}^d \times \mathbb{R}^d} \|\mathbf{x}_1 - \mathbf{x}_2\|^2 \, \mathrm{d}\tilde{\pi}(\mathbf{x}_1, \mathbf{x}_2), \\
&= \mathcal{W}_2(\mathbf{P}_\sharp^\top \mu_1, \mathbf{P}_\sharp^\top \rho_{t_1})
\end{aligned}
$$

$$
\begin{aligned}
L(t, &\mathbf{y}, \mathbf{u_y}; \mathbf{R_y}, \mathbf{c_y}, \mathbf{v_y}, \mathbf{m_y}, U_\mathbf{y}) \\
&= \frac{1}{2}(\mathbf{u_y} - \mathbf{v_y})^\top \mathbf{R_y}(\mathbf{u_y} - \mathbf{v_y}) + \mathbf{c_y}^\top(\mathbf{u_y} - \mathbf{m_y}) - U_\mathbf{y}(\mathbf{y}, t), \\
&= \frac{1}{2}(\mathbf{u_x} - \mathbf{v_x})^\top \mathbf{P}^\top \mathbf{R_y} \mathbf{P}(\mathbf{u_x} - \mathbf{v_x}) + \mathbf{c_y}^\top \mathbf{P}(\mathbf{u_x} - \mathbf{m_x}) - U_\mathbf{y}(\mathbf{P}\mathbf{x}, t), \\
&= \frac{1}{2}(\mathbf{u_x} - \mathbf{v_x})^\top \mathbf{R_x}(\mathbf{u_x} - \mathbf{v_x}) + \mathbf{c_x}^\top(\mathbf{u_x} - \mathbf{m_x}) - U_\mathbf{x}(\mathbf{x}, t), \\
&= L(t, \mathbf{x}, \mathbf{u_x}; \mathbf{R_x}, \mathbf{c_x}, \mathbf{v_x}, \mathbf{m_x}, U_\mathbf{x}),
\end{aligned}
$$

where $\mathbf{R_x} = \mathbf{P}^\top \mathbf{R_y} \mathbf{P} \in \mathbb{R}^{d \times d}$ is guaranteed to be positive definite, $\mathbf{P}_\sharp^\top$ is the push-forward operator of the PCA projection represented by $\mathbf{P}^\top$, $\mathbf{c_x} = \mathbf{P}^\top \mathbf{c_y} \in \mathbb{R}^d$, and $U_\mathbf{x}(\mathbf{x}, t) = U_\mathbf{y}(\mathbf{P}\mathbf{y}, t)$ is the potential function.

In Section 3.3 and Appendix E.5, we provided the four Lagrangian examples of the NLSB and their use cases. The examples demonstrate that the Lagrangian design allows a variety of prior knowledge to be reflected in the sample trajectories.

In this appendix, we have described the LSB problem on the linear coordinate transformed space. In order to improve the modeling of dynamics, it is recommended to investigate the potential benefits of incorporating nonlinearities in the projection from $\mathbf{x}$ to $\mathbf{y}$ or imposing specific geometric structures on the space of $\mathbf{x}$, as considered in (Huguet et al., 2022). These considerations are deemed promising avenues for the advancement of NLSB.

### D.1 POPULATION DYNAMICS SIMULATION BY REVERSE-TIME SDE

We describe a method for estimating the trajectories backwards through time and show experimental results using artificial synthetic data. According to the result from (Anderson, 1982), the reverse-time SDE of the original SDE (Eq. (3)) is given by

$$
\mathrm{d}\mathbf{X}_t = \{\mathbf{f}_\theta(\mathbf{X}_t, t) - \mathbf{u}_{\theta, \phi}(\mathbf{X}_t, t)\} \, \mathrm{d}t + \mathbf{g}_\phi(\mathbf{X}_t, t) \, \mathrm{d}\tilde{\mathbf{W}}_t,
$$
$$
\mathbf{u}_{\theta, \phi}(\mathbf{X}_t, t) = \mathrm{div}\left(\mathbf{g}_\phi(\mathbf{X}_t, t)\mathbf{g}_\phi(\mathbf{X}_t, t)^\top\right) - \mathbf{g}_\phi(\mathbf{X}_t, t)\mathbf{g}_\phi(\mathbf{X}_t, t)^\top \nabla_\mathbf{x} \log p_t(\mathbf{X}_t),
$$

where $\tilde{\mathbf{W}}_t$ is a Wiener process that flows backwards in time.

However, the computation of the modification term $\mathbf{u}$ of the drift function requires an estimation of the score function and high computational cost. Therefore, we newly parameterize the modification term $\mathbf{u}_\xi$ by NN with parameters $\xi$. Then, we train only the term $\mathbf{u}_\xi$ using Sinkhorn divergence loss in Eq. (45) by numerically simulating the reverse-time SDE. For simplicity of implementation, we used the same model architecture for modification term $\mathbf{u}_\xi$ as for the drift $\mathbf{f}_\theta$.

$$
\tilde{\ell}(\xi) = \sum_{t_k \in T \setminus t_{K-1}} \overline{\mathcal{W}}_\epsilon(\mu_k, \rho_{t_k}^\xi) \tag{45}
$$

The experimental results for the reverse-time SDE learning method are shown in Appendix E.2.2

Table 3: Comparison of implementation

|  | velocity/drift | diffusion | ODE/SDE solver |
|---|---|---|---|
| TrajectoryNet | FFJORD (Grathwohl et al., 2018) | - | dopri5 |
| OT-Flow | $-\nabla_{\mathbf{x}}\Phi_{\theta}(t, \mathbf{x})$ | - | Euler |
| Neural SDE | $-\nabla_{\mathbf{x}}\Phi_{\theta}(t, \mathbf{x})$ | FCNN $\mathbf{g}_{\phi}(\mathbf{x}, t)$ | Euler-Maruyama |
| NLSB (Ours) | $\nabla_{\mathbf{z}}H(t, \mathbf{x}, -\nabla_{\mathbf{x}}\Phi_{\theta})$ | FCNN $\mathbf{g}_{\phi}(\mathbf{x}, t)$ | Euler-Maruyama |
| IPF (GP) | sparse GP | Hyperparameter $\mathbf{g}(t)$ | Euler-Maruyama |
| IPF (NN) | $-\nabla_{\mathbf{x}}\Phi_{\theta}(t, \mathbf{x})$ | Hyperparameter $\mathbf{g}(t)$ | Euler-Maruyama |
| SB-FBSDE | $-\nabla_{\mathbf{x}}\Phi_{\theta}(t, \mathbf{x})$ | Hyperparameter $\mathbf{g}(t)$ | Euler-Maruyama |

# E  EXPERIMENTAL DETAILS AND ADDITIONAL RESULTS

## E.1  IMPLEMENTATION

We compared our methods against standard neural SDE, TrajectoryNet (Tong et al., 2020), OT-Flow (Onken et al., 2021), IPF with GP (Vargas et al., 2021) and NN (De Bortoli et al., 2021), SB-FBSDE (Chen et al., 2021a). TrajectoryNet and OT-Flow are examples of existing ODE-based methods, neural SDE is an example of learning SDE using only data without prior information by the Lagrangian, IPF and SB-FBSDE are methods to find SDE solutions to the classical SB problem. While IPF and SB-FBSDE solve the SB problem, which is special case of the LSB problem, they are a different algorithm from NLSB. The parameterization and numerical solvers for ODEs and SDEs are summarized in Table 3. For a fair comparison, we used the same potential model $\Phi_{\theta}$ described in Section 4.2 for OT-Flow, neural SDE, NLSB, IPF (NN), and SB-FBSDE. We set the number of ResNet layers $M = 2$, the step size $h = 1.0$, the rank of matrix $\text{rank}(\mathbf{A}) = 10$, and the dimension of the hidden vector $\mathbf{z}$ to 2. In training all models, we used Adam optimizer to optimize all learnable parameters with a learning rate of $0.001$ and the decay rate of $\beta_1 = 0.9$, $\beta_2 = 0.999$. We searched all weight coefficients of regularization terms $\lambda_e, \lambda_h$ in $(0.0, 0.5]$ and selected those with the largest possible coefficients among those with sufficiently small EMD-L2 values on the validation data.

In our implementation, we modified code in the TrajectoryNet[1], OT-FLow[2], IPF[3], and SB-FBSDE[4] repositories, which were released under the MIT license. Our experimental environment consists of an Intel Xeon Plantinum 8360Y (36-core) CPU and a single NVIDIA A100 GPU. Our code is available at `https://github.com/take-koshizuka/nlsb`. TrajectoryNet and OT-Flow were trained using the torchdiffeq library[5], and neural SDE and NLSB with the torchsde library[6]. The settings common to all experiments for each method are described below.

### NEURAL SDE AND NLSB

We trained the drift and diffusion models of the standard neural SDE using only the Sinkhorn divergence $\overline{\mathcal{W}}_{\epsilon}$. We used the Euler-Maruyama method with the constant step size of $0.01$ as an SDE solver. Backpropagation was performed without using the adjoint method.

### TRAJECTORYNET

The velocity model of TrajectoryNet includes three concatsquash layers with hyperbolic tangent activations. A concatsquash layer cs was defined in the released code of FFJORD (Grathwohl et al., 2018) by:

$$\text{cs}(\mathbf{x}, t) = (\mathbf{W}_x \mathbf{x} + \mathbf{b}_x)\sigma(\mathbf{W}_t t + \mathbf{b}_t) + (\mathbf{W}_b t + \mathbf{b}_b t),$$

where $\sigma$ is the sigmoid function, and $\mathbf{W}_x$, $\mathbf{W}_t$, $\mathbf{W}_b$, $\mathbf{b}_x$, $\mathbf{b}_t$, $\mathbf{b}_b$ are all learnable parameters.

---

[1] `https://github.com/KrishnaswamyLab/TrajectoryNet`
[2] `https://github.com/EmoryMLIP/OT-Flow`
[3] `https://github.com/AforAnonyMeta/IPML-2548`
[4] `https://github.com/ghliu/SB-FBSDE`
[5] `https://github.com/rtqichen/torchdiffeq`
[6] `https://github.com/google-research/torchsde`

The base models of TrajectoryNet was trained using only the negative log-likelihood loss in standard CNF scheme. +OT represents a model trained with the OT-based regularization $\tilde{\mathcal{R}}_e$ defined by Eq. (1). We set the interval-dependent coefficients $\tilde{\lambda}_e$ for the OT-based regularization as well as Eq. (11). For the ODE solver, the dopri5 solver with both absolute and relative tolerances set to $10^{-5}$ was used.

### OT-FLOW

The base models of OT-Flow was also trained with CNF scheme. We used both $\tilde{\mathcal{R}}_e$ and $\tilde{\mathcal{R}}_h$ with the interval-dependent coefficients $\tilde{\lambda}_e$, $\tilde{\lambda}_h$ in OT-Flow + OT. We used the Euler method as the ODE solver with a constant step size of $0.01$ and both absolute and relative tolerances set to $10^{-5}$.

### IPF

The drift model of IPF (GP) was changed to sparse GP with the exponential kernel from vanilla GP (Vargas et al., 2021) to save computation cost. We selected 100 inducing points using the K-means algorithm in IPF (GP). The SB problem was solved by using IPF algorithm with 15 iterations. We used the Euler-Maruyama method as the SDE solver. The diffusion coefficients of IPF were tuned as hyperparameters.

### SB-FBSDE

We employed alternating training and solved the classical SB problem specifying Brownian motion as the prior stochastic process and did not use collectors.

## E.2 SYNTHETIC POPULATION DYNAMICS: TIME-DEPENDENT ORNSTEIN–UHLENBECK PROCESS

We validated NLSB on artificial synthetic data generated from one-dimensional SDEs, the time-dependent Ornstein–Uhlenbeck process used in (Kidger et al., 2021). In this experiment, the predicted trajectory and uncertainty can be compared with the ground-truth and easily evaluated by visualization. The purpose of this experiment is to confirm that the NLSB works well on a simple linear SDE.

### E.2.1 DETAILS OF EXPERIMENTAL SETUP

We trained all models with a batch size of $512$ for each time point. The tuned weight coefficients are shown in Table 4.

### POTENTIAL MODEL IN OT-FLOW, NEURAL SDE, NLSB, AND IPF (NN)

For the potential model $\Phi_\theta(\mathbf{x}, t)$, we set the number of ResNet layers $M = 2$, the step size $h = 1.0$, the rank of matrix $\text{rank}(\mathbf{A}) = 10$, and the dimension of the hidden vector $\mathbf{z}$ to 2.

### NEURAL SDE AND NLSB

We used a two-layer FCNN of a hidden dimension 16 for the diffusion function $\mathbf{g}_\phi(\mathbf{x}, t)$. The activations functions were LipSwish. In NLSB, we used the Lagrangian for the potential-free system $L(t, \mathbf{x}, \mathbf{u}) = \frac{1}{2}||\mathbf{u}||^2$.

### TRAJECTORYNET

We used the concatsquash layers of a hidden dimension 16.

### IPF

We set the diffusion coefficients as follows.

$$\mathbf{g}(t) = \begin{cases} 0.2 & t \in [0.0, 1.0) \\ 0.6 & t \in [1.0, 2.0) \\ 1.0 & t \in [2.0, 3.0) \\ 1.4 & t \in [3.0, 4.0] \end{cases}.$$

Other experimental settings have not been changed from (Vargas et al., 2021).

Table 4: Weight coefficients for regularization terms in experiments on synthetic data

|  | $[t_0, t_1]$ | $[t_1, t_2]$ | $[t_2, t_3]$ | $[t_3, t_4]$ |
|---|---|---|---|---|
| $\lambda_e, \lambda_h$ for NLSB | $0.3, 0.2$ | $0.1, 0.01$ | $0.01, 0.0001$ | $0.01, 0.0001$ |
| $\tilde{\lambda}_e$ for TrajectoryNet + OT | $0.1$ | $0.1$ | $0.001$ | $0.001$ |
| $\tilde{\lambda}_e, \tilde{\lambda}_h$ for OT-Flow + OT | $0.1, 0.01$ | $0.1, 0.01$ | $0.001, 0.001$ | $0.001, 0.001$ |

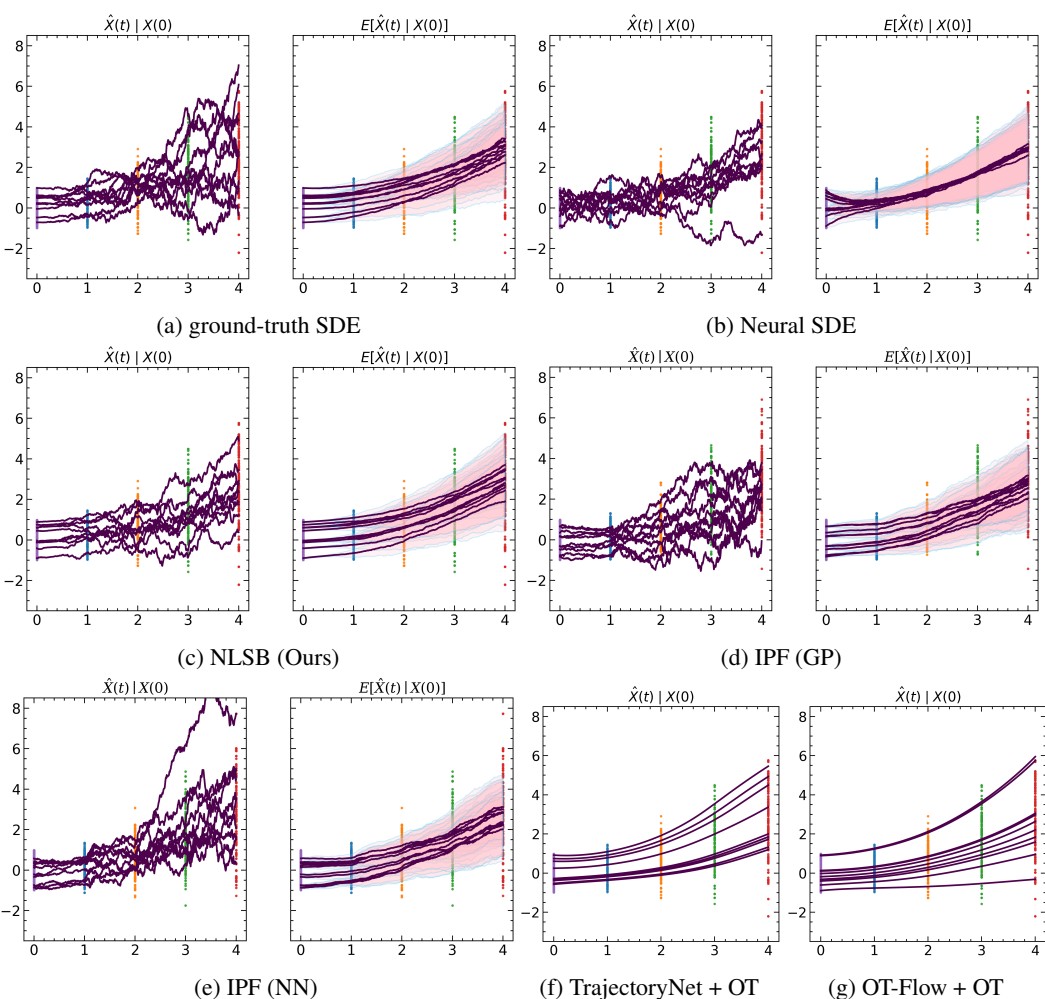

Figure 6: 1D OU process data and predictions.

## E.2.2 RESULTS

Figure 6 shows the visualization of both the original trajectory and the averaged trajectory by the SDE-based methods (neural SDE, NLSB, and IPF) and the only original trajectory by the ODE-based methods (TrajectoryNet and OT-Flow). All trajectories were generated by all-step prediction from the initial samples at the time $t = 0$. The five colored point clouds in the background are the ground-truth data given at each time point. The pink area and the light blue line are the one-sigma empirical confidence intervals and their boundaries for each trajectory, respectively.

Figure 3 indicates that NLSB and IPF outperform neural SDE and is comparable to other ODE-based methods in estimating populations with small variance. In contrast, the SDE-based methods outperform ODE-based methods when estimating populations with a large variance. That indicates

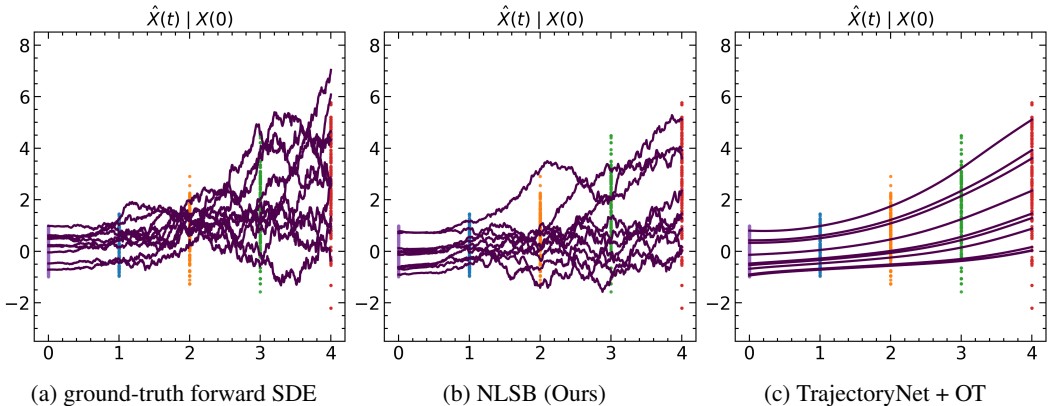

(a) ground-truth forward SDE       (b) NLSB (Ours)       (c) TrajectoryNet + OT

Figure 7: 1D OU process data and predictions backwards through time.

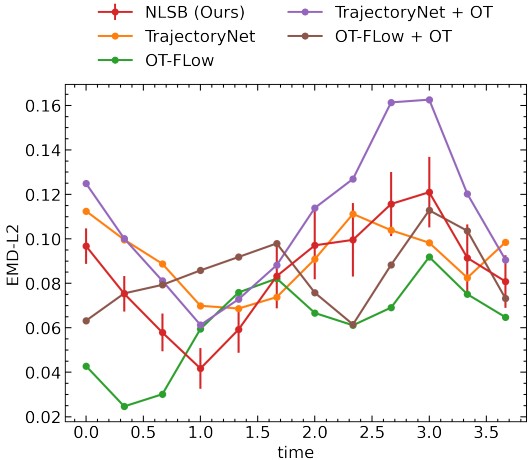

Figure 8: Numerical evaluation for the reverse-time simulation on synthetic OU process data. All MDD values were computed between the ground-truth and the estimated samples within generated trajectories all-step behind from initial samples $\mathbf{x}(t_4)$.

that NLSB and IPF can estimate population-level dynamics even when the population variance is large or small. Furthermore, NLSB and IPF have a smaller CDD value than neural SDE. Figure 6b shows that the average behavior of samples $\mathbb{E}[X(t)|X(0)]$ estimated by neural SDE is different from that of the ground-truth SDE (see Fig. 6a), especially in the interval $[0, 1]$. In contrast, the predictions by NLSB and IPF in Figs. 6c to 6e are much closer to the ground-truth. These results show that the prior knowledge of the potential-free system helps to estimate the sample-level dynamics.

Note that the LSB problem solved by NLSB with the Lagrangian $L = \frac{1}{2}\|\mathbf{u}\|^2$ and the SB problem solved by IPF are almost mathematically equivalent (see Appendices A.3 and B.1). The result that NLSB shows comparable performance to IPF, even though NLSB is trained differently from IPF, indicates that NLSB is a unified framework and can deal with SB problems as a special case.

The quantitative evaluation results of the reverse-time ODE/SDE using MDD are shown in Fig. 8, and the visualization of trajectories is shown in Fig. 7. All trajectories from NLSB and TrajectoryNet were generated by all-step prediction from the initial samples at the time $t = 4$ to $t = 0$. Experimental results show that the proposed method described in Appendix D.1 successfully recovers the population-level dynamics of the reverse-time SDE. The development of appropriate evaluation methods for sample-level dynamics of the reverse-time SDE is included in future work.

Table 5: Weight coefficients for regularization terms in experiments on scRNA-seq data

|  | $[t_0, t_1]$ | $[t_1, t_2]$ | $[t_2, t_3]$ | $[t_3, t_4]$ |
|---|---|---|---|---|
| $\lambda_e, \lambda_h$ for NLSB (E) | 0.1, 0.01 | 0.01, 0.01 | 0.001, 0.0001 | 0.01, 0.001 |
| $\lambda_e, \lambda_h$ for NLSB (V) | 0.01, 0.001 | 0.01, 0.001 | 0.01, 0.001 | 0.01, 0.001 |
| $\lambda_e, \lambda_h$ for NLSB (D) | 0.01, 0.001 | 0.01, 0.001 | 0.01, 0.001 | 0.01, 0.001 |
| $\lambda_e, \lambda_h$ for NLSB (E+D+V) | 0.01, 0.001 | 0.001, 0.001 | 0.001, 0.001 | 0.001, 0.001 |
| $\tilde{\lambda}_e$ for TrajectoryNet + OT | 0.01 | 0.01 | 0.1 | 0.1 |
| $\tilde{\lambda}_e, \tilde{\lambda}_h$ for OT-Flow + OT | 0.01, 0.01 | 0.01, 0.01 | 0.001, 0.01 | 0.001, 0.01 |

### E.3 SINGLE-CELL POPULATION DYNAMICS

We evaluated our method on the time-evolution of single-cell populations obtained from a developing human embryo system. In this experiment, we presented a new quantitative evaluation metric in a practical setting and validated the effectiveness of the NLSB on real data of single-cell population.

#### E.3.1 DETAILS OF DATASET

We evaluated our method on embryoid body scRNA-seq data (Moon et al., 2019), which is also used in (Tong et al., 2020; Vargas et al., 2021; Bunne et al., 2021; 2022). This data shows the differentiation of human embryonic stem cells from embryoid bodies into diverse cell lineages, including mesoderm, endoderm, neuroectoderm, and neural crest, over 27 days. During this period, cells were collected at five different snapshots ($t_0$: day 0 to 3, $t_1$: day 6 to 9, $t_2$: day 12 to 15, $t_3$: day 18 to 21, $t_4$: day 24 to 27). The collected cells were then measured by scRNAseq, filtered at the quality control stage, and mapped to a low-dimensional feature space using a principal component analysis (PCA). For details, see Appendix E.2 in (Tong et al., 2020). We reused the pre-processed data available in the released repository of TrajectoryNet [1] and split the data into 200 samples ($\sim 8.5\%$) of validation data, 350 samples ($\sim 15\%$) of test data, and the rest as train data for each time point. The scRNA-seq data are licensed under Creative Commons Attribution 4.0 International license.

#### E.3.2 DETAILS OF EXPERIMENTAL SETUP

We trained all models with a batch size of 1000 for each time point and used the early stopping method, which monitors the EMD-L2 value on the validation data. The tuned weight coefficients are shown in Table 5. We used the same potential model $\Phi_\theta$ in OT-Flow, neural SDE, NLSB and IPF (NN) with the same hyperparameters described in Appendix E.2.1. In the following, we describe the different settings from the experiment in Section 5.1.

#### NLSB

We used the Lagrangian for the cellular system and compared several combinations of the regularization terms. In Tables 1 and 2, "E" is the energy term, "D" is the density term, and "V" is the velocity term. The density term $U(\mathbf{x}, t)$ is the log-likelihood function of the data estimated by GMM. For the calculation of the density regularization term, the time-dependent density function $U(\mathbf{x}, t)$ was defined by

$$U(\mathbf{x}, t) = c \log p(\mathbf{x}; \Theta_t), \quad \Theta_t = \{\mu_m^{(t)}, \Sigma_m^{(t)}\}_{m=1}^{M_t},$$

where $c$ is a hyperparameter to change the scale, and $\mu_m^{(t)}, \Sigma_m^{(t)}$ are mean and variance parameters of the mixed Gaussian distribution, respectively. When $t \in [t_k, t_{k+1}]$, the parameters $\Theta_t$ were estimated with the data at $t_k$ and $t_{k+1}$ by GMM and the number of mixture components $M_t$ was determined by the value of Bayesian information criterion (BIC). The hyperparameter $c$ was searched among $\{0.1, 1.0, 10.0\}$ and we set $c = 10.0$ for NLSB (D) and $c = 0.1$ for NLSB (E+D+V). For the calculation of the velocity regularization term, we used the same reference velocity as those used in TrajectoryNet (Tong et al., 2020).

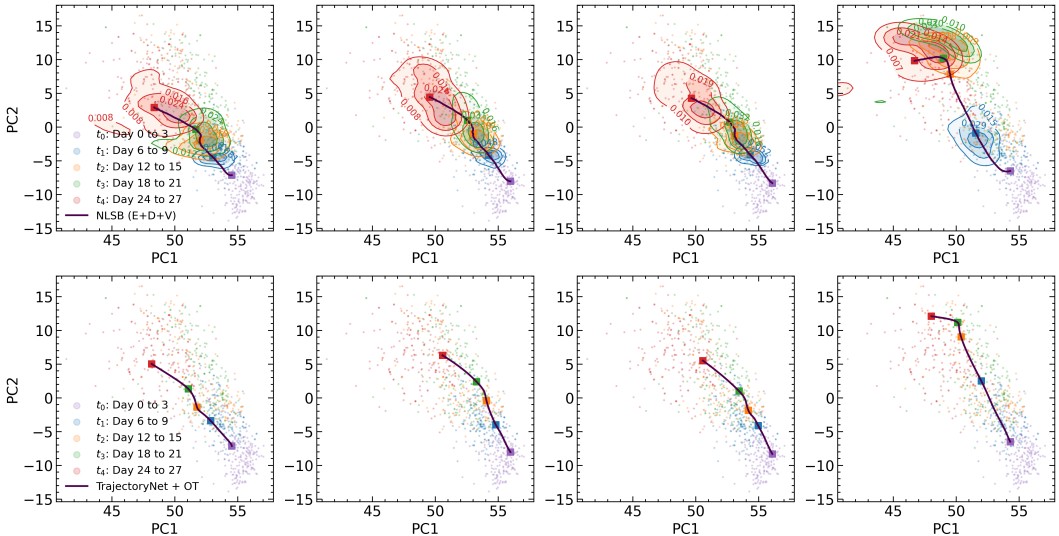

Figure 9: Visualization of the time evolution for a single sample on scRNA-seq data. The upper and lower images are predictions from the same initial sample at $t_0$, with the upper row predicted by NLSB and the lower row predicted by TrajectoryNet + OT. The color gradients depict the magnitude of the probability density. The probability density function is estimated by GMM with five mixture components.

Table 6: The MDD value (EMD-L1) for population-level dynamics on five-dimensional (5D) PCA space at time of observation for scRNA-seq data.

| MDD (EMD-L1) $\downarrow$ | $t_1$ | $t_2$ | $t_3$ | $t_4$ |
|---|---|---|---|---|
| NLSB (E) | $1.13 \pm 0.025$ | $1.36 \pm 0.035$ | $1.34 \pm 0.023$ | $1.30 \pm 0.018$ |
| NLSB (D) | $1.08 \pm 0.021$ | $1.40 \pm 0.043$ | $1.38 \pm 0.030$ | $\mathbf{1.29} \pm 0.024$ |
| NLSB (V) | $1.13 \pm 0.030$ | $1.39 \pm 0.043$ | $1.34 \pm 0.029$ | $1.35 \pm 0.025$ |
| NLSB (E+D+V) | $1.09 \pm 0.023$ | $1.34 \pm 0.037$ | $\mathbf{1.32} \pm 0.024$ | $1.30 \pm 0.025$ |
| Neural SDE | $1.11 \pm 0.028$ | $1.41 \pm 0.041$ | $1.38 \pm 0.033$ | $1.34 \pm 0.025$ |
| OT-Flow | $1.31$ | $1.73$ | $1.68$ | $1.69$ |
| OT-Flow + OT | $1.33$ | $1.65$ | $1.69$ | $1.56$ |
| TrajectoryNet | $1.15$ | $1.60$ | $1.42$ | $1.58$ |
| TrajectoryNet + OT | $1.20$ | $1.60$ | $1.41$ | $1.72$ |
| IPF (GP) | $1.14 \pm 0.024$ | $1.59 \pm 0.052$ | $1.49 \pm 0.037$ | $1.57 \pm 0.042$ |
| IPF (NN) | $1.16 \pm 0.027$ | $1.42 \pm 0.037$ | $1.37 \pm 0.030$ | $1.37 \pm 0.027$ |
| SB-FBSDE | $\mathbf{0.89} \pm 0.016$ | $\mathbf{1.32} \pm 0.025$ | $1.63 \pm 0.030$ | $1.57 \pm 0.015$ |

### TRAJECTORYNET

We used the velocity model with 64 hidden dimensions.

### IPF

We set the diffusion coefficients to $g(t) = 0.5 \ (t \in [0.0, 1.0))$, $1.0 \ (t \in [1.0, 4.0])$. Other experimental settings have not been changed from (Vargas et al., 2021).

### SB-FBSDE

We set the diffusion coefficients to $g(t) = 0.5 \ (t \in [0.0, 3.0))$, $0.1 \ (t \in [3.0, 4.0])$.

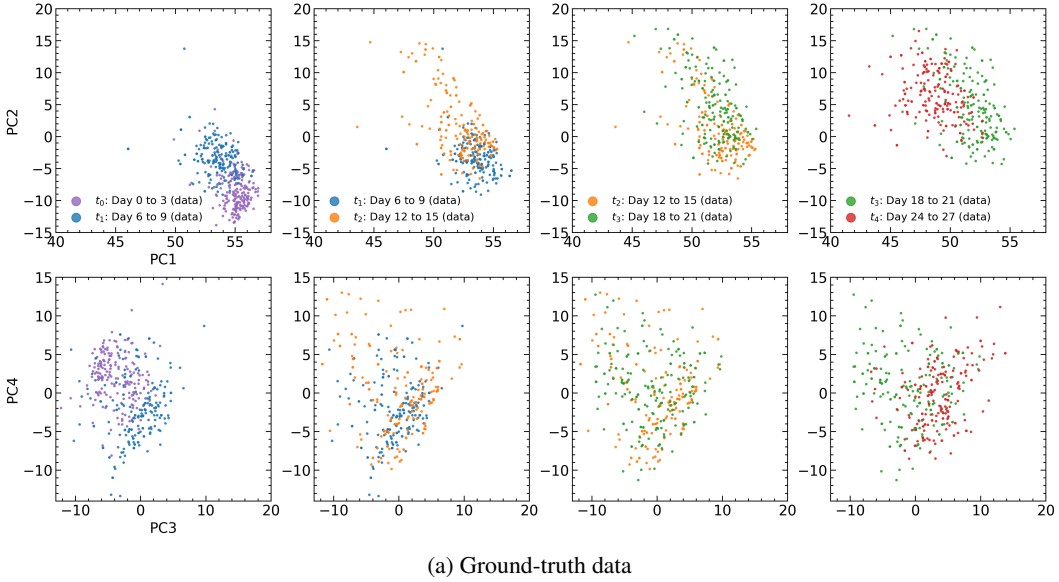

(a) Ground-truth data

### E.3.3 RESULTS

The time evolution of the distribution for a single sample is visualized as a heat map in Fig. 9. The sample population in Fig. 10 and the trajectories are visualized in Fig. 11. All figures are visualizations in the space of the first and second principal components. The x-axis represents the first principal component and the y-axis the second principal component. The experimental results using MDD using EMD with $L^1$ cost is shown in Table 6.

Figure 9 shows that the SDE-based methods, including NLSB, can handle the uncertainty of the trajectories in contrast to ODE-based methods. Figure 10 shows that NLSB outperforms ODE-based methods in predicting the transitions with a high diffusion of samples from $t_1$ to $t_2$ and from $t_3$ to $t_4$, indicating that the explicit modeling of diffusion is effective. Figure 11 shows that the drift estimated by NLSB (E) is linear, the trajectories by NLSB (D) pass on the data manifold, and the trajectories by NLSB (E+D+V) appear to reflect all other regularization effects.

A GIF animation of the NLSB simulation in PCA space is also included in the supplemental materials.

### E.4 SYNTHETIC POPULATION DYNAMICS: TRAJECTORIES REFLECTING THE POTENTIAL FUNCTION

This section presents the experimental results of applying the NLSB to numerical simulations of synthetic population dynamics. Through this experiment, we show that the NLSB can model a wide range of phenomena by designing the Lagrangian based on prior knowledge. In particular, we emphasize the importance of designing potential functions and the flexibility of penalty design for drift functions.

### E.4.1 DATASET

We used two-dimensional uniform distributions $\mathcal{U}_0$ and $\mathcal{U}_1$ for the endpoints at time $t = 0$ and $t = 1$.

$$(X_0, Y_0) \sim \mathcal{U}_0 : -1.25 \leq X_0 \leq -1, \ -1 \leq Y_0 \leq 1,$$
$$(X_1, Y_1) \sim \mathcal{U}_1 : 1 \leq X_1 \leq 1.25, \ -1 \leq Y_1 \leq 1.$$

We generated 2048 and 512 samples from two endpoint distributions as training and validation data, respectively.

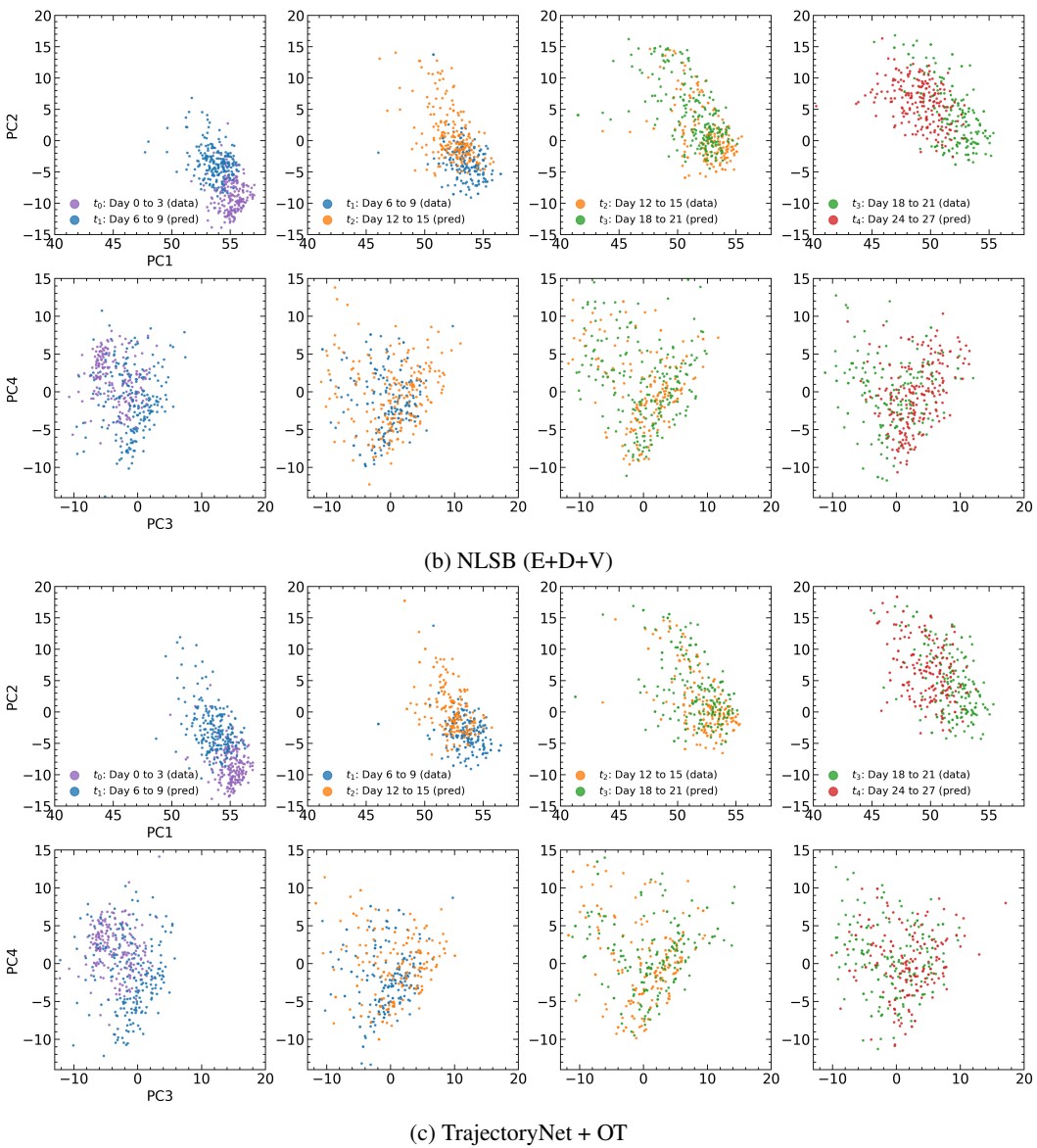

(b) NLSB (E+D+V)

(c) TrajectoryNet + OT

Figure 10: The ground-truth data and one-step ahead sample prediction on scRNA-seq data.

### E.4.2 DETAILS OF EXPERIMENTAL SETUP

We trained NLSB with a batch size of $512$ and used the early stopping method, which monitors the validation loss value. We adopt the Lagrangian for the random dynamical system. All weight coefficients of regularization terms were searched in $\{0.01, 0.001\}$.

We first conducted experiments applying the NLSB with the Lagrangian of the form $L(t, \mathbf{x}, \mathbf{u}) = \frac{1}{2}||\mathbf{u}||^2 - U(\mathbf{x})$ defined by four different potential functions shown below. We implemented the box and slit-shaped obstacle potential functions as differentiable by using the sigmoid functions.

#### BOX-SHAPED OBSTACLE

$$U(x, y) = \begin{cases} -100 & -0.5 \leq x, y \leq 0.5 \\ 0 & \text{otherwise} \end{cases}.$$

#### SLIT-SHAPED OBSTACLE

$$U(x, y) = \begin{cases} -100 & (-0.1 \leq x \leq 0.1) \wedge (y \leq -0.25 \vee 0.25 \leq y) \\ 0 & \text{otherwise} \end{cases}.$$

#### HILL POTENTIAL

$$U(x, y) = -2.5(x^2 + y^2).$$

#### WELL POTENTIAL

$$U(x, y) = -10 \exp(-(x^2 + y^2)).$$

Next, we conducted an experiment using the Lagrangian $L(t, \mathbf{u}, \mathbf{x}) = \frac{1}{2}\mathbf{u}^\top \mathbf{R}\mathbf{u}$. The matrix $\mathbf{R}$ can be used to penalize the magnitude of the drift differently for each dimension. We set $R$ to $\mathrm{diag}([10.0, 0.1])$ and $\mathrm{diag}([0.1, 10.0])$.

### E.4.3 RESULTS

The visualization results are shown in Figs. 12 and 13. Figure 12 shows that NLSB can estimate various trajectories that reflect information about obstacles or regions where samples cannot or are likely to exist represented by the potential function. Figure 13a and 13b show that the drift of the trajectories generated by the NLSB with the Lagrangian $L = \frac{1}{2}\mathbf{u}^\top \mathbf{R}\mathbf{u}$ is larger on the axis with smaller penalties defined by $\mathbf{R}$ and vice versa.

### E.5 OPINION DYNAMICS

We demonstrated the application of NLSB to optimal control on a party model of opinion dynamics (Schweighofer et al., 2020; Gaitonde et al., 2021; Liu et al., 2022).

### E.5.1 DATASET

Opinion dynamics (Schweighofer et al., 2020; Gaitonde et al., 2021) is the time evolution of each agent's opinions interacting with each other. MFGs theory provides a mathematical analytical framework for the opinion dynamics of large agent populations, which are very difficult to handle computationally. The dynamics is modeled using SDEs defined on the opinion representation space of each agent embedded in Euclidean space. In recent years, the phenomenon of strong polarization (Gaitonde et al., 2021), in which agents are divided into groups with opposite opinions, has attracted particular attention. We use the drift $\bar{\mathbf{f}}_{\mathrm{polarize}}$, which causes polarization defined in the party model (Gaitonde et al., 2021), as prior information of the target system.

$$\bar{\mathbf{f}}_{\mathrm{polarize}} := \mathbf{f}_{\mathrm{polarize}}/\|\mathbf{f}_{\mathrm{polarize}}\|^{\frac{1}{2}}, \ \overline{\mathbf{y}} := \mathbf{y}/\|\mathbf{y}\|^{\frac{1}{2}},$$
$$\mathbf{f}_{\mathrm{polarize}}(\mathbf{x}, \rho; \boldsymbol{\xi}) := \mathbb{E}_{\mathbf{y} \sim \rho}\left[a(\mathbf{x}, \mathbf{y}; \boldsymbol{\xi})\overline{\mathbf{y}}\right],$$
$$a(\mathbf{x}, \mathbf{y}; \boldsymbol{\xi}) := \begin{cases} 1 & \text{if } \mathrm{sign}\left(\langle \mathbf{x}, \boldsymbol{\xi} \rangle\right) = \mathrm{sign}\left(\langle \mathbf{y}, \boldsymbol{\xi} \rangle\right) \\ -1 & \text{otherwise} \end{cases},$$

where $\rho$ is the probability measure of the population, $\boldsymbol{\xi}$ is random information from some distribution independent of $\rho$, $a(\mathbf{x}, \mathbf{y}; \boldsymbol{\xi})$ is the agreement function, which represents whether the two opinions $\mathbf{x}$ and $\mathbf{y}$ agree on the information $\boldsymbol{\xi}$.

We used two-dimensional Gaussian distributions $\mathcal{N}_0$ and $\mathcal{N}_1$ for the endpoints at time $t = 0$ and $t = 1$.

$$(X_0, Y_0) \sim \mathcal{N}_0 = \mathcal{N}\left(\mathbf{0}, \begin{bmatrix} 0.5 & 0.0 \\ 0.0 & 0.25 \end{bmatrix}\right),$$

$$(X_1, Y_1) \sim \mathcal{N}_1 = \mathcal{N}\left(\mathbf{0}, \begin{bmatrix} 3.0 & 0.0 \\ 0.0 & 3.0 \end{bmatrix}\right).$$

We generated $2048$ and $512$ samples from two endpoint distributions as training and validation data, respectively.

### E.5.2 DETAILS OF EXPERIMENTAL SETUP

The Lagrangian for the opinion dynamics is defined by

$$L(t, \mathbf{x}, \mathbf{u}) = \frac{1}{2}\|\bar{\mathbf{f}}_{\text{polarize}}(\mathbf{x}, t) + \mathbf{u}(\mathbf{x}, t)\|^2 - U(\mathbf{x}, t).$$

The potential function $U(\mathbf{x}, t)$ represents the averaged interaction that each agent receives from the population. The entropy function $U(\mathbf{x}, t; c) = c \log p(\mathbf{x}, t)$ with a constant coefficient $c$ is a candidate for a useful potential function and helps control changes in population diversity. The optimal drift function is given by $\mathbf{f}_\theta = -\nabla_{\mathbf{x}}\Phi_\theta(\mathbf{x}, t) - \bar{\mathbf{f}}_{\text{polarize}}(\mathbf{x}, t)$. By setting the ideal opinion distribution as the terminal condition, NLSB can be used as a method to find the optimal drift converging to the ideal opinion distribution.

We trained NLSB with a batch size of $512$ and used the early stopping method, which monitors the validation loss value. All weight coefficients of regularization terms were searched in $\{0.01, 0.001\}$. We conducted experiments using the NLSB with the Lagrangian for the opinion dynamics.

### E.5.3 RESULTS

Visualization of polarized opinion dynamics driven by the drift $\bar{\mathbf{f}}_{\text{polarize}}$ and the time variation of directional similarity are shown in Figs. 14a and 15a. The directional similarity (Schweighofer et al., 2020) is the distribution of cosine angles between paired opinions, with the red and blue color gradients representing the degree of disagreement and agreement, respectively. Figures 14b and 15b show the results after applying NLSB. These results show that NLSB can learn a drift function that prevents the polarization caused by $\bar{\mathbf{f}}_{\text{polarize}}$, which also converges to the ideal terminal distribution $\mathcal{N}_1$.

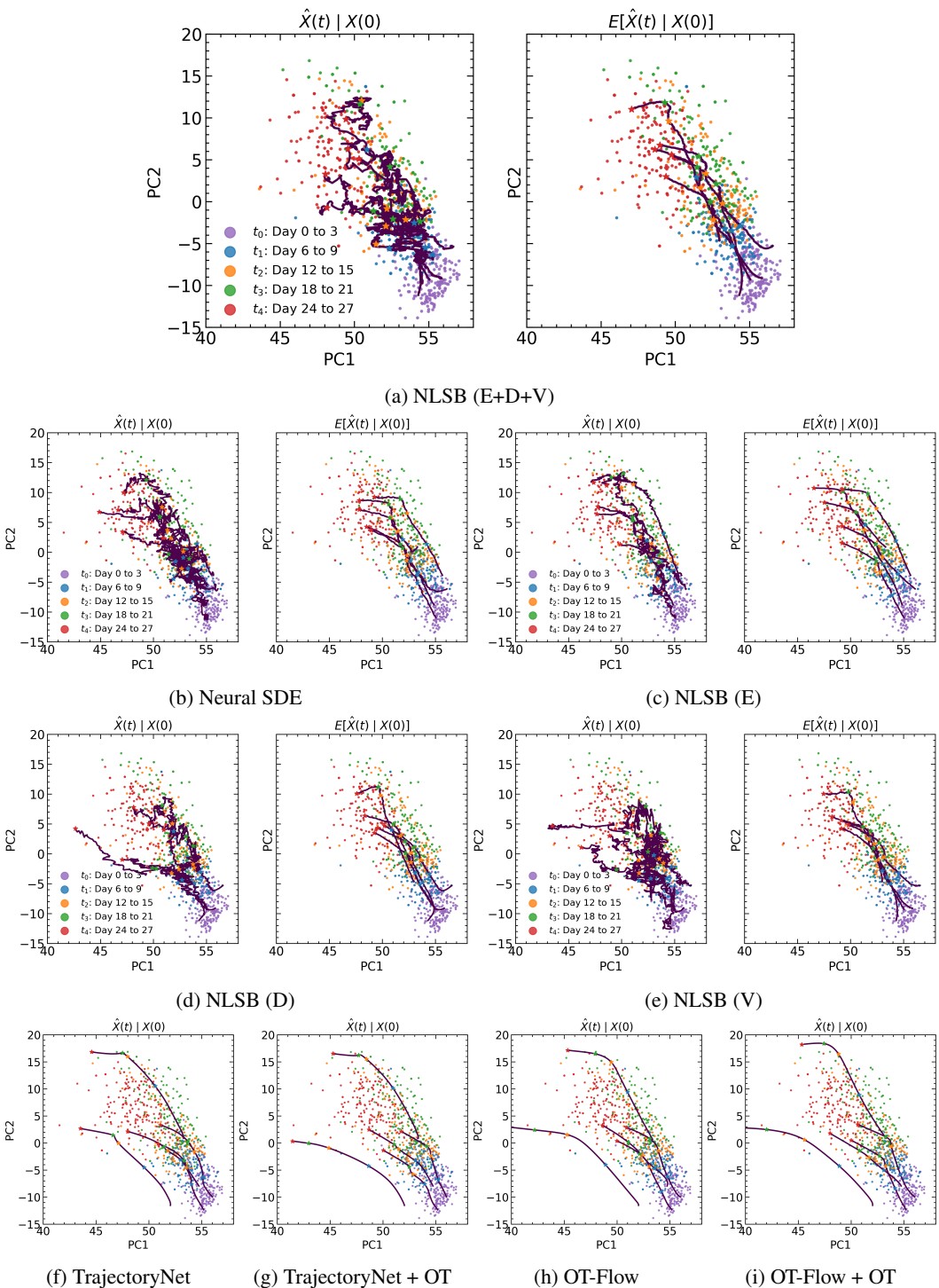

Figure 11: scRNA-seq data and predictions. The x- and y-axes denote the first and second principal components, respectively. The five colored point clouds in the background are the ground-truth data given at each time point. All five trajectories are generated by all-step prediction from the initial samples at $t_0$.

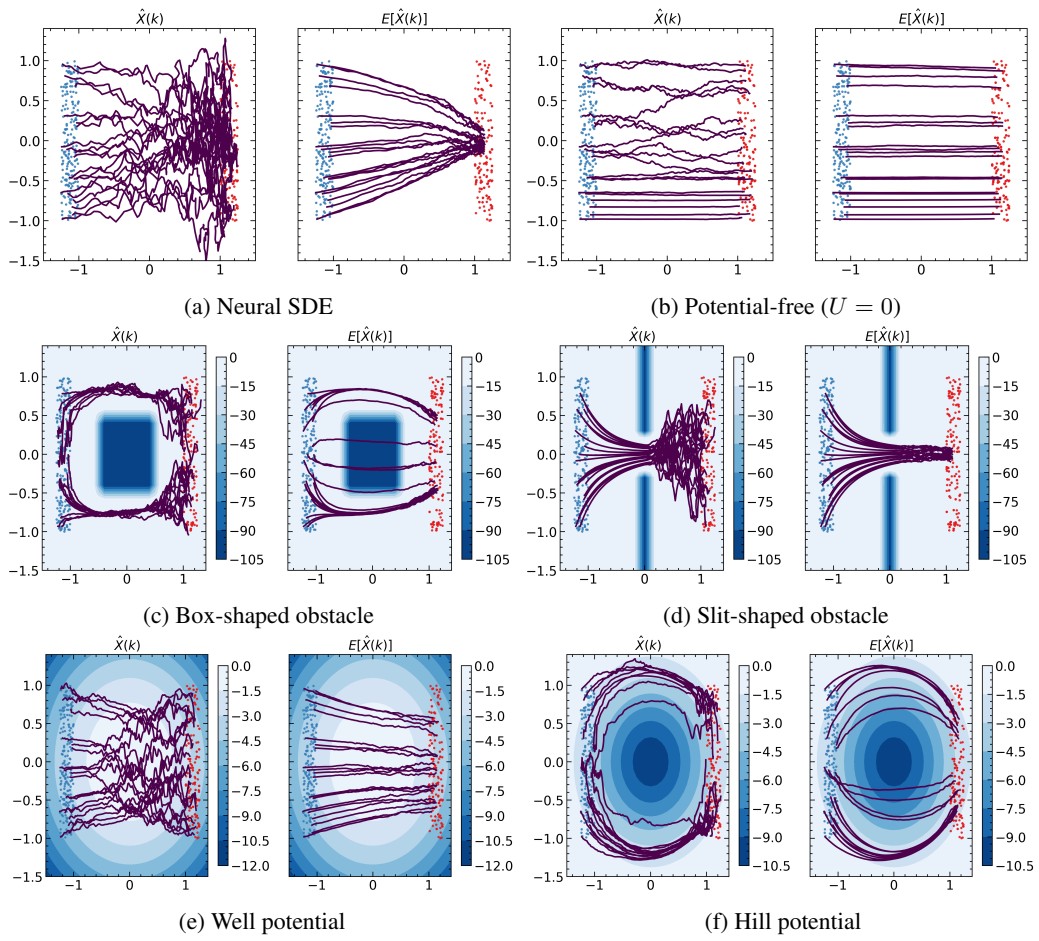

Figure 12: Visualization of trajectories reflecting the potential function. The color gradients depict the magnitude of the potential function.

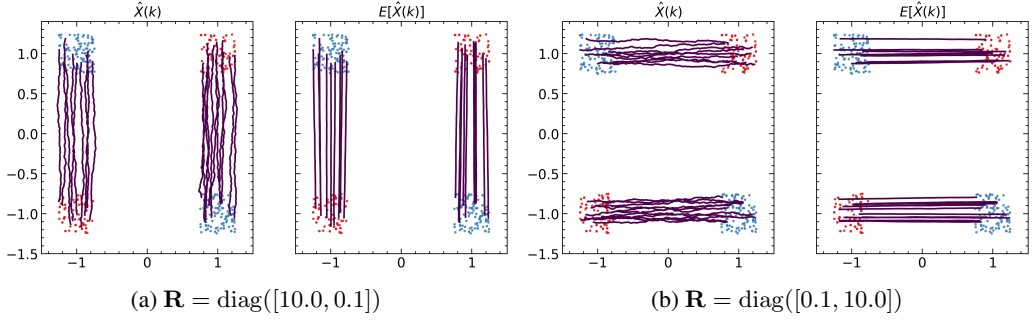

Figure 13: Visualization of trajectories by NLSB using the Lagrangian $L = \frac{1}{2}\mathbf{u}^\top \mathbf{R}\mathbf{u}$. The blue and red point clouds are the source and target distributions, respectively.

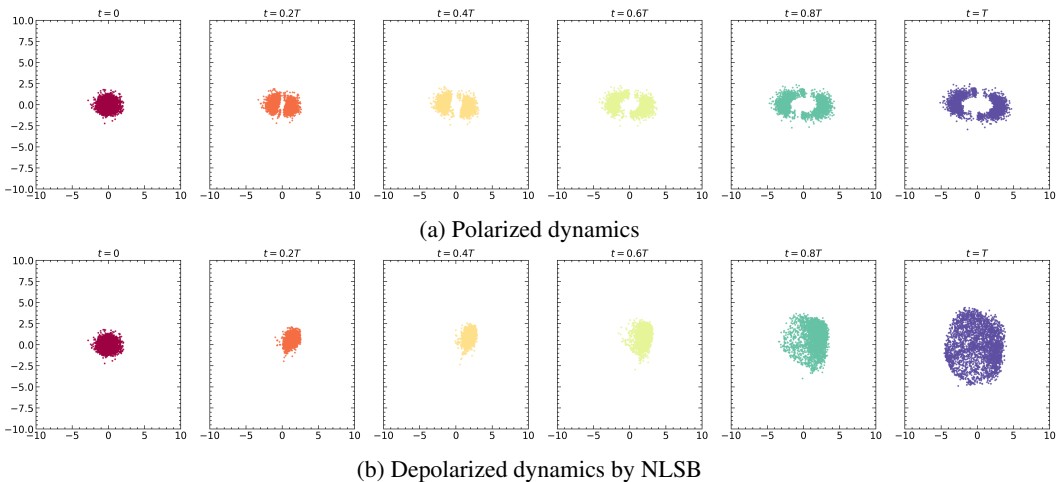

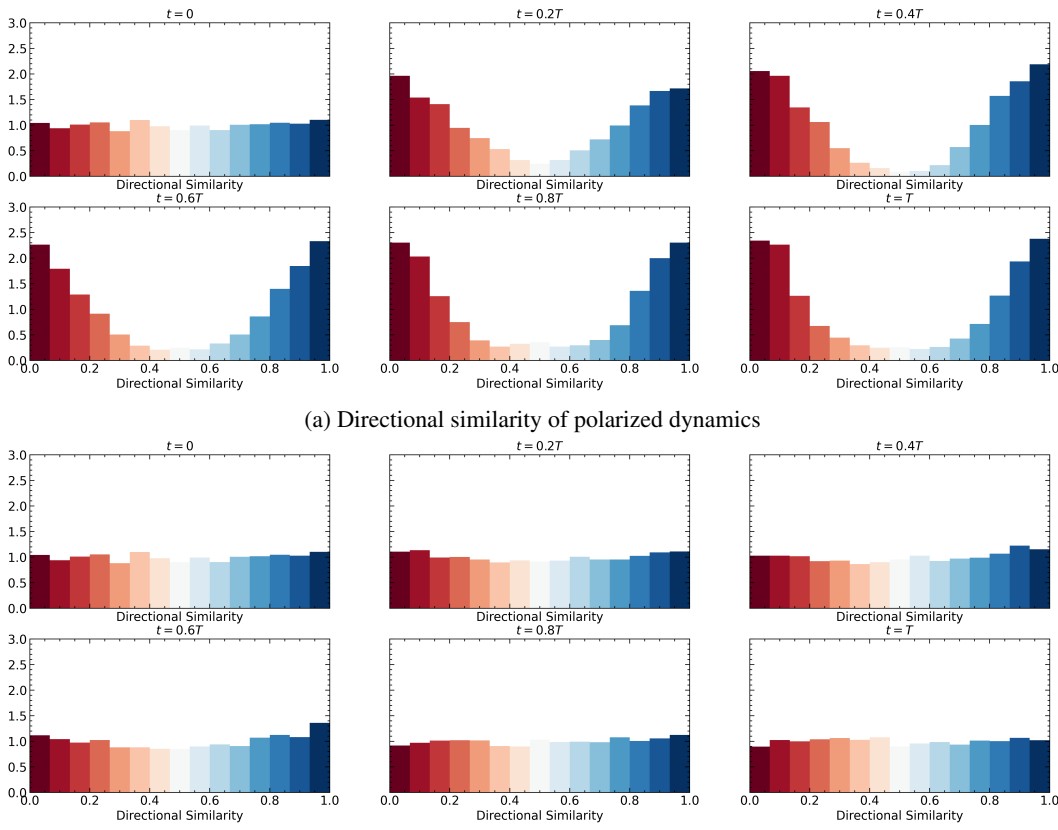

