# OpenReview forum: "Neural Lagrangian Schr\"{o}dinger Bridge: Diffusion Modeling for Population Dynamics"
_ICLR.cc/2023/Conference — ICLR 2023 notable top 25%_

### Official Review · Reviewer_8YwN · 2022-10-19

**Confidence:** 4
**Correctness:** 4
**Technical Novelty And Significance:** 2
**Empirical Novelty And Significance:** 2
**Recommendation:** 8

**Clarity, Quality, Novelty And Reproducibility:**

Please see the Strength and Weaknesses above. I have concerns regarding the novelty of the submission. I also think the presentation can be substantially improved to highlight the contributions.

**Strength And Weaknesses:**

Strength:
The proposed framework is quite flexible. Moreover, it is simple, and it seems to work in experiments. The application of sing-cell population dynamics is interesting.

Weaknesses:
The major issue of the paper for me is that the contributions of the submission are not clear:
- The objective (7) is not really new. The idea of relaxing the endpoint constraint by a softer one ($\mathcal{W}_2$ in (7)) was exploited by the prior work of Lavenant et al. 2021 in a similar context, although without the neural SDE; the authors seem have missed this reference. The idea of incorporating the Hamiltonian term (9) was also present in existing literature (Yang & Karniadakis 2020). As for the Lagrangian term $\mathcal{R}_e$, the authors simply reuse the regularization of Tong et al. 2020 without designing new ones.

- There is also little contribution from the algorithmic side, since the authors essentially combined various known methods to treat each term separately in (7).

- I'm not sure why the authors mentioned various SB related training but did not compare to them in the experiments. For instance, what is IPML (NN) in Table 1? Is it from Vargas et al. 2021 or De Bortoli et al. 2021? Is there any reason of including these methods, instead of, say, Chen et al. 2021a?

- The $\mathcal{W}_2$ in the proposed objective (7) suddenly turns into the entropy-regularized $\mathcal{W}_\epsilon$ in (11). Why not propose $\mathcal{W}_\epsilon$ to begin with?

-----
Reference
Lavenant et al. 2021, Towards a mathematical theory of trajectory inference.

**Summary Of The Paper:**

This paper proposes the Lagrangian Schrodinger bridge (LSB) problem, a framework to recover the population dynamics from temporal data. LSB is a stochastic optimal control problem, which enables the authors to leverage the structure and propose the training objective (7). This method is evaluated on synthetic and single-cell population datasets.

**Summary Of The Review:**

This paper proposed an interesting framework for population dynamics. However, in its present state, it seems like gluing together several pieces (SB, stochastic optimal control, regularization from previous work, etc.) without any core contribution. I'm therefore inclined to rejection.

I suggest the following improvements:

1. Revise the presentation to highlight the contribution, focusing especially on the benefits of the neural SDE component in (6).

2. Expand Section 3.3 to include more examples, even designing new ones, to reflect the generality of the proposed LSB. Otherwise, if the authors merely included one previous example (the cellular system), then the applicability of LSB is not really convincing to me.

3. Since the theoretical contribution is fairly limited, the authors should perform more extensive experiments. In particular, for the single-cell population dynamics, the authors might consider including the dataset of Schiebinger et al. 2019. In addition, following the point above, experiments on more applications, instead of just synthetic and single-cell population dynamics, are recommended.

---

> ### Author Response · Authors · 2022-11-20
> **Response to Reviewer 8YwN (1/3)**
>
> > 1: The idea of relaxing the endpoint constraint by a softer one ( W2  in (7)) was exploited by the prior work of Lavenant et al. 2021 in a similar context, although without the neural SDE; the authors seem have missed this reference.
>
> Thank you for pointing this out.
> * We have cited this literature additionally and added the text as follows.
>
>     First, we relax the constraint at time $t_1$, i.e. $R_e + D(\mu_1, \rho_{t_1})$, by using Wasserstein distance (Bunne et al., 2021; Hashimoto et al., 2016), KL-divergence(Finlay et al., 2020; Onken et al., 2021; Tong et al., 2020), or a combination of these  (Lavenant et al., 2021) for the discrepancy measure $D$.
>
> > 2: The idea of incorporating the Hamiltonian term (9) was also present in existing literature (Yang & Karniadakis 2020).
>
> We should not include the paper on Physics-informed ML, such as Yang & Karniadakis 2020, in the citation because it would confuse the contribution and context of our paper for the following three reasons.
>
> (1) (Darbon et al. 2020) cited in (Yang & Karniadakis 2020) regarding the HJ-equation is about the connection between NN-network architecture and the viscosity solution of the HJ-equation. The goal of our paper is to estimate the population dynamics, while the goal of (Darbon et al. 2020) is quite different.
>
> (2) The LSB problem we formulated is an optimization problem on a coupled system of SDE (Fokker-Planck equation) and HJB equation with fixed endpoint distributions. This problem is mathematically different from simply solving the HJB equation such as (Darbon et al. 2020).
>
> (3) The goal of Physics-informed ML is generally to develop ML-based differential equation solvers with faster and better extrapolation performance, or to solve inverse problems. Many of the datasets used in the papers on Physics-Informed ML were generated by numerical simulations of differential equations. Therefore, the purpose and problem settings are different from our paper.
>
> > 3: As for the Lagrangian term $R_e$, the authors simply reuse the regularization of (Tong et al. 2020) without designing new ones.
>
> Since this comment is related to “point 8:", please see the response there. (the next thread)
>
> > 4: There is also little contribution from the algorithmic side, since the authors essentially combined various known methods to treat each term separately in (7).
>
> Thank you for your comments. Equation (7) is not simply a combination of existing methods for the following three reasons.
>
> (1) The first term in Eq. (7) is the data-fitting term; the use of the Wasserstein metric is not the essence of the NLSB. We just followed existing works (Bunne et al., 2021; Hashimoto et al., 2016). Minimizing the distribution error of the target distribution is common in machine learning, such as in generative modeling tasks, and the introduction of regularization terms is also common.
>
> (2) As we mentioned in the response to “point 2:”, our paper differs in context and mathematical problem setting from the Physics-Informed ML papers, including (Yang & Karniadakis 2020, and Darbon et al. 2020).
>
> (3) The OT-based regularizations proposed in TrajectoryNet and OT-Flow described in Section 2.1 is a loss function for neural ODEs based on theoretical results of the Brenier-Benamou (BB) problem. In contrast, we proposed the loss function for neural SDEs based on the theoretical results of our newly formulated LSB problem. The fact that the losses in Eqs. (8) and (9) appear to be in the general form of Eqs. (1) and (2) is because of non-trivial and desirable theoretical results obtained by the SOT theory and our theoretical contributions. We were the first to introduce the SOT theory into the field of machine learning, formulate the LSB problem, prove the existence of solutions to the LSB problem, and derive optimality conditions by variational methods (Theorem 17). Furthermore, Theorem 17 justifies the minimization of $R_h$ and a parameterization that strictly satisfies Eq. (45) during training when the terminal constraint is not satisfied. The existing mathematical results are explained in Appendix A.1~A.3 and our theoretical contributions are highlighted in Appendix A.4 and A.5. Please check it.
>
> > 5:  The W_2  in the proposed objective (7) suddenly turns into the entropy-regularized Wϵ in (11). Why not propose Wϵ to begin with?
>
> Thank you for your comments. We modified objective (7) by replacing the Wasserstein distance with the general distribution discrepancy $D$.

---

> > ### Author Response · Authors · 2022-11-20
> > **Response to Reviewer 8YwN (2/3)**
> >
> > > 6: I'm not sure why the authors mentioned various SB related training but did not compare to them in the experiments. For instance, what is IPML (NN) in Table 1? Is it from Vargas et al. 2021 or De Bortoli et al. 2021? Is there any reason of including these methods, instead of, say, Chen et al. 2021a?
> >
> > Thank you for your question. IPML(GP) and IPML (NN) were corresponding to the method from (Vargas et al., 2021) and (De Bortoli et al. 2021), respectively. We compared to them because an experiment using the same sc-RNAseq data was done in (Vargas et al., 2021).
> >
> > * We revised the text in Section 5 as follows and renamed IPML as IPF.
> >
> >     We compared our methods against standard neural SDE, OT-Flow (Onken et al., 2021), TrajectoryNet (Tong et al., 2020) and IPF with GP (Vargas et al., 2021) and NN (De Bortoli et al., 2021).  … The drift model of IPF (GP) was changed to sparse GP from vanilla GP (Vargas et al., 2021) to save computation cost. The drift model of IPF (NN) is the same network as the NLSB for a fair comparison.
> >
> > We have not compared our method with the method of (Chen et al. 2021a) for the following reasons.
> >
> > (1) The experiments in (Chen et al. 2021a) are mainly concerned with image generation. Because many settings, such as network architecture, are tuned for image generation, a lot of trial and error is required when the method of (Chen et al. 2021a) is used on sc-RNA data.
> >
> > (2) The choice of the prior-SDE $dX=fdt+gdW$ is non-trivial. In the case of image generation, it is natural to use VP-SDE or VE-SDE for prior-SDE from existing studies (Song et al. 2020). However, this is not clear in the case of simulations of cellular dynamics. At least, tuning the diffusion function $g$ is intrinsically important.
> >
> > (3) If, as in our paper, we parameterize $g$ using NN and train it using backpropagation, this is a new approach. (This is a similar approach to data assimilation methods such as 4D-Var and is a strength of our method.)
> >
> > (Opinion) The fact that $g$ cannot be a function of $X$ may be a serious problem in natural science simulations. This is because, in many natural phenomena, the effect of disturbances clearly depends on $X$. We too are interested in comparing the effectiveness of these methods in terms of prediction performance and computation time. We would like to do this as future research.
> >
> > (Song et al. 2020) –  Score-Based Generative Modeling through Stochastic Differential Equations
> >
> > > 7: Revise the presentation to highlight the contribution, focusing especially on the benefits of the neural SDE component in (6).
> >
> > Thank you for your suggestion. We revised Section 3.1 to highlight the benefits of using the neural SDE. Please confirm it.
> >
> > > 8: Expand Section 3.3 to include more examples, even designing new ones, to reflect the generality of the proposed LSB. Otherwise, if the authors merely included one previous example (the cellular system), then the applicability of LSB is not really convincing to me.
> >
> > Thank you for your comments. We added the new Lagrangian, the quadratic-form Lagrangian, and experimental results using it in Appendix D.5. Please check it.
> >
> > From Lagrangian dynamics, it has been theoretically shown that the quadratic-form Lagrangian can describe all systems of objects following the equations of motion, independent of the coordinate system. The Lagrangian cost used in optimal control problems is also defined in the quadratic-form Lagrangian. In Appendix D.5, we showed that the quadratic-form Lagrangian has sufficient representational capacity to describe Lagrangians in coordinate systems that can be linearly transformed into each other in a unified manner. We also experimentally showed that NLSB using different potential functions can produce diverse trajectories in Figure 10.
> >
> > The benefit of generalization by Lagrangian is that the HJB equation and Hamilton equation can be described in a unified manner using Lagrangian, independent of the coordinate system of the space on SDE. In other words, the loss function can always be easily derived once the Lagrangian is designed, even when the potential function U is additionally considered or the coordinate system is transformed. We also clarified this in Appendix D.5.
> >
> > Furthermore, the energy regularization $1/2 \|v\|^2$ of (Tong et al. 2020) is a regularization for the velocity function of **neural ODEs**, and the term $1/2 \|u\|^2$ in our paper is a regularization for the drift function of **neural SDE**. And our major contribution is to reformulate density regularization and velocity regularization of (Tong et al. 2020) in a unified Lagrangian framework. (TrajectoryNet did not use the Hamilton and HJB equations.)  Our loss function is based on a more generalized theoretical background than existing loss functions as explained in the response to “point 4:”. The reorganized Appendix A provides this theoretical background in full.

---

> > > ### Author Response · Authors · 2022-11-20
> > > **Response to Reviewer 8YwN (3/3)**
> > >
> > > > 9: Since the theoretical contribution is fairly limited, the authors should perform more extensive experiments. In particular, for the single-cell population dynamics, the authors might consider including the dataset of Schiebinger et al. 2019. In addition, following the point above, experiments on more applications, instead of just synthetic and single-cell population dynamics, are recommended.
> > >
> > > Thank you for your suggestion. From points 4: and 8:, it seems to me that our theoretical contribution might be underrated. While we agree that many experiments would better confirm the validation of the method, we think that experimental validation is already sufficient, since only artificial data experiments and one experiment using the public real data were conducted in (TrajectoryNet et al. 2020) and (Bunne et al., 2021). Also, the data used in experiment 5.2 is the dataset of (Schiebinger et al. 2019).

---

> > > > ### Author Response · Authors · 2022-12-01
> > > > **Additional experiments related to point 8: and 9:**
> > > >
> > > > We added the experiment to drive opinion dynamics (Schweighofer et al. 2020) in the field of Mean-field games.
> > > >
> > > > The party model is proposed by (Gaitonde et al. 2021): given random information $\xi$ sampled from some distribution independent of the probability measure $\rho$, each agent updates the opinion following a normalized polarize dynamic $\bar{f}\_{\text {polarize }}=f\_{\text {polarize }} /\left\|f\_{\text {polarize }}\right\|^{\frac{1}{2}}$, where
> > > >
> > > > $$f\_{\text {polarize }}(x, \rho ; \xi):=\mathbb{E}\_{y \sim \rho}[a(x, y ; \xi) \bar{y}], a(x, y ; \xi):=
> > > > \begin{cases}1 & \text { if } \operatorname{sign}(\langle x, \xi\rangle) = \operatorname{sign}(\langle y, \xi\rangle) \\\\ -1 & \text { otherwise }\end{cases},\ \bar{y} = y/||y||^{\frac{1}{2}}$$
> > > >
> > > > The drift $\bar{f}\_{\text{poralize}}$ leads to polarization as shown in Experiment 5.2 of (Liu et al. 2022) and our result: [https://github.com/fhiuwbhnvsu/image_archive/tree/main/Opinion/Poralization](https://github.com/fhiuwbhnvsu/image_archive/tree/main/Opinion/Poralization)
> > > >
> > > > The experimental setup for the data is the same as (Liu et al. 2022). (The paper (Liu et al. 2022) was published on 2022/9/20 and our paper was released earlier.)
> > > >
> > > > We defined the new Lagrangian for the opinion dynamics by $L(t, x, u) = 1/2 ||\bar{f}\_{\text{poralize}}+ u||^2 - F\_{\text{entropy}}$ where  $\bar{f}\_{\text{poralize}}$ and $F\_{\text{entropy}}$ were defined in Eqs. (17) and (18) of (Liu et al. 2022). We set the target distribution at the time $t_1 = 3.0$ to the Gaussian distribution representing depolarized population and trained NLSB with $\lambda\_e = 0.01$, $\lambda\_h = 0.01$. The log-likelihood $\log \rho(x, t)$ was estimated using GMM. Our goal is to obtain dynamics that prevent polarization and converge at the termination time to the desired Gaussian distribution.
> > > >
> > > > Our results are shown in [https://github.com/fhiuwbhnvsu/image_archive/tree/main/Opinion/NLSB](https://github.com/fhiuwbhnvsu/image_archive/tree/main/Opinion/NLSB) The image prediction.png shows the simulation of the population dynamics, and sims.png shows the dynamics of the directional similarity. The color gradient of the red and blue colors represent the degree of disagreement and agreement (Liu et al. 2022), respectively.
> > > >
> > > > These results show that NLSB can learn a drift function that prevents the polarization caused by $\bar{f}\_{\text{poralize}}$. Furthermore, this experiment emphasizes the high applicability of our framework.
> > > >
> > > > ---
> > > >
> > > > (Schweighofer et al. 2020) — “An agent-based model of multidimensional opinion dynamics and opinion alignment.” Chaos: An Interdisciplinary Journal of Nonlinear Science, 30(9):093139, 2020.
> > > >
> > > > (Gaitonde et al. 2021) — “Polarization in geometric opinion dynamics.” In Proceedings of the 22nd ACM Conference on Economics and Computation, pages 499–519,
> > > > 2021.
> > > >
> > > > (Liu et al. 2022) — "Deep Generalized Schrödinger Bridge." *Advances in Neural Information Processing Systems.*

---

> > > > > ### Comment · Reviewer_8YwN · 2022-12-07
> > > > > **Thank you for the rebuttal**
> > > > >
> > > > > I thank the authors for their strong rebuttal and apologize for the late reply. I have raised my score from 5 to 6 as the rebuttal resolved most of my concerns. The remaining ones are discussed below.
> > > > >
> > > > > ---
> > > > > - The reason for not comparing to (Chen et al. 2021a) is not convincing.
> > > > >
> > > > > The authors mentioned that the methods of (Chen et al. 2021a) are mainly concerned with image generation. By the same token, (De Bortoli et al. 2021), a baseline of the submission, should also be excluded. In my opinion, the authors should not pick and choose which method to include. I encourage the authors to also incorporate (Chen et al. 2021a) with moderate tuning so as to substantiate the claim "a lot of trial and error is required when the method of (Chen et al. 2021a) is used on sc-RNA data."
> > > > >
> > > > > ---
> > > > > - Regarding the contribution of stochastic optimal control.
> > > > >
> > > > > I believe it is an overstatement that the authors "were the first to introduce the SOT theory into the field of machine learning, formulate the LSB problem." First, the equivalence between the SOT and its various connections to the variational formulas is well-known. For instance, the Lagrange multiplier proof for Theorem 17 is a standard tool in the field; see https://arxiv.org/abs/2005.10963.
> > > > >
> > > > > Second, at least one paper in ML is concerned with SOT: https://arxiv.org/pdf/2207.02149.pdf. This work proposes a soft-constraint formulation of the SOT problem by relaxing $X_1\sim\rho_1$ via a penalty method. I have not done my best effort to search for related topics, and I encourage the authors to include a more detailed comparison to other relevant contributions in the ML fields.

---

> > > > > > ### Author Response · Authors · 2022-12-07
> > > > > > **Response to Reviewer 8YwN (part. 4)**
> > > > > >
> > > > > > Thank you very much for your reply to my rebuttal and your reconsideration of the score.
> > > > > >
> > > > > > > 1:  The reason for not comparing to (Chen et al. 2021a) is not convincing me
> > > > > >
> > > > > > Thank you for your constructive suggestions. I'm not sure if I can add more experiments before the end of the discussion, but we're struggling to add a comparison with (Chen et al. 2021a).
> > > > > >
> > > > > > > 2: First, the equivalence between the SOT and its various connections to the variational formulas is well-known. For instance, the Lagrange multiplier proof for Theorem 17 is a standard too in the field; see [https://arxiv.org/abs/2005.10963](https://arxiv.org/abs/2005.10963)
> > > > > >
> > > > > > I agree that my proof in Theorem 17 is a common and similar approach to the existing proof method. However, I have not seen any paper that gives proof using variational methods for a SOC problem with a general Lagrangian L constrained by the general Fokker-Planck equation. Thus, it may not be a major contribution, but it is solid.
> > > > > >
> > > > > > > 3: Second, at least one paper in ML is concerned with SOT: [https://arxiv.org/pdf/2207.02149.pdf](https://arxiv.org/pdf/2207.02149.pdf) This work proposes a soft-constraint formulation of the SOT problem by relaxing X1∼ρ1 via a penalty method. I have not done my best effort to search for related topics, and I encourage the authors to include a more detailed comparison to other relevant contributions in the ML fields.
> > > > > >
> > > > > > Although we cannot show it here due to anonymity, our paper is published on the arXiv on Mon, 11 Apr 2022 03:32:17 UTC. On the other hand, the paper [https://arxiv.org/pdf/2207.02149.pdf](https://arxiv.org/pdf/2207.02149.pdf) was published on Mon, 27 Jun 2022 14:01:06 UTC, (This paper seems to be also currently under review at ICLR2023.)

---

> > > > > > > ### Author Response · Authors · 2022-12-11
> > > > > > > **Additional Experiments and further explanation of related studies**
> > > > > > >
> > > > > > > > 1: The reason for not comparing to (Chen et al. 2021a) is not convincing me
> > > > > > >
> > > > > > > We added the experiments of (Chen et al. 2021a) on the RNA-seq data. We used the same prior-SDE as IPF, and the drift function was set to $-\nabla_x \Phi_{\theta}$ using the potential model from OT-Flow for a fair comparison. Implementation is available at [https://github.com/fhiuwbhnvsu/image_archive/tree/main/SB-FBSDE](https://github.com/fhiuwbhnvsu/image_archive/tree/main/SB-FBSDE).
> > > > > > >
> > > > > > > The results are shown in [https://github.com/fhiuwbhnvsu/image_archive/blob/main/Fig5.png](https://github.com/fhiuwbhnvsu/image_archive/blob/main/Fig5.png),
> > > > > > >
> > > > > > > [https://github.com/fhiuwbhnvsu/image_archive/blob/main/Tab.1.png](https://github.com/fhiuwbhnvsu/image_archive/blob/main/Tab.1.png), and [https://github.com/fhiuwbhnvsu/image_archive/blob/main/Tab.2.png](https://github.com/fhiuwbhnvsu/image_archive/blob/main/Tab.2.png).
> > > > > > >
> > > > > > > Table 1 and Table 2 are also shown below.
> > > > > > >
> > > > > > > Tab. 1
> > > > > > > | MDD (EMD-L2) | $t_1$ | $t_2$ | $t_3$ | $t_4$ |
> > > > > > > | --- | --- | --- | --- | --- |
> > > > > > > | NLSB (E) | 0.71$\pm$ 0.020 | 0.86 $\pm$ 0.027 | 0.83 $\pm$ 0.016 | 0.79 $\pm$ 0.012 |
> > > > > > > | NLSB (D) | 0.67 $\pm$ 0.017 | 0.90 $\pm$ 0.029 | 0.87 $\pm$ 0.018 | 0.79 $\pm$ 0.016 |
> > > > > > > | NLSB (V) | 0.70 $\pm$ 0.023 | 0.89 $\pm$ 0.030 | 0.83 $\pm$ 0.022 | 0.81 $\pm$ 0.019 |
> > > > > > > | NLSB (E+D+V) | 0.68 $\pm$ 0.016 | 0.84 $\pm$ 0.030 | 0.81 $\pm$ 0.018 | 0.79 $\pm$ 0.017 |
> > > > > > > | Neural SDE | 0.69 $\pm$ 0.020 | 0.91 $\pm$ 0.029 | 0.85 $\pm$ 0.025 | 0.81$\pm$ 0.017 |
> > > > > > > | OT-Flow | 0.83 | 1.10 | 1.07 | 1.05 |
> > > > > > > | OT-Flow + OT | 0.85 | 1.05 | 1.09 | 1.00 |
> > > > > > > | TrajectoryNet | 0.73 | 1.06 | 0.90 | 1.01 |
> > > > > > > | TrajectoryNet + OT | 0.76 | 1.05 | 0.88 | 1.10 |
> > > > > > > | IPF (GP) | 0.70 $\pm$ 0.015 | 1.04 $\pm$ 0.041 | 0.94 $\pm$ 0.029 | 0.98 $\pm$ 0.033 |
> > > > > > > | IPF (NN) | 0.73 $\pm$ 0.019 | 0.89 $\pm$ 0.030 | 0.84 $\pm$ 0.019 | 0.83 $\pm$ 0.020 |
> > > > > > > | SB-FBSDE  | 0.56 $\pm$ 0.010 | 0.97 $\pm$ 0.091 | 1.01 $\pm$ 0.023 | 1.15 $\pm$ 0.029 |
> > > > > > >
> > > > > > > Tab. 2
> > > > > > > | Mean CDD | $[t_0, t_2]$ | $[t_1, t_3]$ | $[t_2, t_4]$ |
> > > > > > > | --- | --- | --- | --- |
> > > > > > > | NLSB (E) | $0.88$ | $0.72$ | $0.79$ |
> > > > > > > | NLSB (D) | $1.64$ | $0.76$ | $0.77$ |
> > > > > > > | NLSB (V) | $1.15$ | $0.82$ | $0.86$ |
> > > > > > > | NLSB (E+D+V) | $0.96$ | $0.83$ | $0.84$ |
> > > > > > > | Neural SDE | $1.36$ | $0.85$ | $0.87$ |
> > > > > > > | IPF (GP) | $0.97$ | $1.03$ | $1.06$ |
> > > > > > > | SB-FBSDE | $0.83$ | $1.53$ | $1.69$ |
> > > > > > >
> > > > > > >
> > > > > > > SB-FBSDE shows better performances in some specific intervals, but overall our NLSB shows better performances.
> > > > > > >
> > > > > > >
> > > > > > > > 4: I encourage the authors to include a more detailed comparison to other relevant contributions in the ML fields.
> > > > > > > We provide below a description of related papers on ‘the path integral approach’.
> > > > > > >
> > > > > > > * Path Integral Sampler: A Stochastic Control Approach For Sampling[Zhang ‘21] [https://arxiv.org/abs/2111.15141](https://arxiv.org/abs/2111.15141)
> > > > > > >
> > > > > > >     In this paper, they solve the SOC-formulation of the SB problem using neural SDE by using the regularization term $R_e$. However, it does not use optimality conditions such as Hamilton's equation and HJB-equation.
> > > > > > >
> > > > > > > * Path Integral Stochastic Optimal Control for Sampling Transition Paths[Holdijk ‘22] [https://arxiv.org/pdf/2207.02149.pdf](https://arxiv.org/pdf/2207.02149.pdf)
> > > > > > >
> > > > > > >     This paper was published after ours. Their Lagrangian differs from ours in that it includes Brownian motion as input and does not consider potential functions. Furthermore, this paper does not minimize the KL-divergence of the termination condition, but rather the KL-divergence between the ground-truth path measure $\pi_{u^*}$ and the estimated path measure $\pi_{u_{\theta}}$ in Eq. (11).
> > > > > > >
> > > > > > >
> > > > > > > We included these works in our citation.

---

> > > > > > > > ### Comment · Reviewer_8YwN · 2022-12-11
> > > > > > > > **Thank you for the followup**
> > > > > > > >
> > > > > > > > I thank the authors for their efforts in addressing my concerns. I've further raised my score from 6 to 8.
> > > > > > > >
> > > > > > > > I have the following comments on the latest version:
> > > > > > > >
> > > > > > > > 1. Please consider acknowledging the weakness in terms of theory. The derivation in the paper presents no new techniques but the authors did not properly cite/compare their theory to the existing literature such as https://arxiv.org/abs/2005.10963. The revisions greatly improved the empirics, which I deem to be the merit of the paper. I am hence not convinced that the paper makes valuable contributions to both theory and practice as advertised in the paper.
> > > > > > > >
> > > > > > > > 2. The comparison to FBSDE is not conclusive as the improvements are fairly minor, although I fully understand that, given the short time frame, an extensive experiment is beyond reasonable. However, I hope the authors can follow up on the comparison to the FBSDE method of Chen et al. 2021a. It is a rather flexible framework that can serve as an out-of-the-box solver for many applications and hence is an adequate baseline.

---

> > > > > > > > > ### Author Response · Authors · 2022-12-12
> > > > > > > > > **Thank you for your comments**
> > > > > > > > >
> > > > > > > > > We are very grateful to you for raising our score again.
> > > > > > > > >
> > > > > > > > > > 1: Please consider acknowledging the weakness in terms of theory.
> > > > > > > > >
> > > > > > > > > Thank you very much for your suggestion. We acknowledge that the theoretical contribution is not significant and not be emphasized in the paper. We emphasize that our framework is a comprehensive framework for existing methods. We make the following revisions in the camera-ready version.
> > > > > > > > >
> > > > > > > > > - We added the following text in Appendix A.4.
> > > > > > > > >
> > > > > > > > >     The derivation of optimality conditions using variational methods is a procedure similar to the derivation of optimality conditions for variational formulations of the SB problem in (Chen et al. 2021b).
> > > > > > > > >
> > > > > > > > > - We will add new appendices detailing ML-based methods for solving SB problems such as (Vargas et al. 2021), (De Bortoli et al., 2021), and (Chen et al., 2021a).
> > > > > > > > >
> > > > > > > > > > 2: The comparison to FBSDE is not conclusive as the improvements are fairly minor, although I fully understand that, given the short time frame, an extensive experiment is beyond reasonable.
> > > > > > > > >
> > > > > > > > > We promise to include a comparison with the more tuned FBSDE in the camera-ready version.

---

### Official Review · Reviewer_6WTu · 2022-10-23

**Confidence:** 3
**Clarity, Quality, Novelty And Reproducibility:** 1) In page 5, when $L=\frac{1}{2} ||u…
**Correctness:** 3
**Technical Novelty And Significance:** 3
**Empirical Novelty And Significance:** 3
**Recommendation:** 6

**Strength And Weaknesses:**

**Pros:**

1. An interesting extension of Schr\"{o}dinger bridge by including additional regularizers, which ends up with an interesting Lagrangian Schr\"{o}dinger bridge that generalized the quadratic transport cost function.

2. avoids the expensive computation of the divergence suffered by the Hutchinson estimator; the adoption of architecture in OT-Flow is interesting.

**Summary Of The Paper:**

The authors formulated the Lagrangian Schrodinger bridge problem and proposed to solve it approximately by the advection-diffusion process with regularized neural SDE. The expensive trace computation operation was also alleviated by adopting a model architecture motivated by OT-Flow. A few experiments were conducted on population dynamics and showed the efficiency in modeling stochastic behavior.

**Summary Of The Review:**

An interesting extension of SB with speed-ups on divergence estimation.

---

> ### Author Response · Authors · 2022-11-16
> **Response to Reviewer 6WTu**
>
> 6WTu
>
> > 1: In page 5, when $L=\frac{1}{2}\|u\|^2$, $f =[\nabla_u L]^{-1} (- \nabla_x \Phi) \neq - \nabla_x \Phi$, the transition from LSB to SB fails, any comments on that?
>
> LSB is turned into SB as follows:
>
> $y(u) = \nabla_u (\frac{1}{2}\|u\|^2)= u$, then the inverse mapping is $u = [\nabla_u L]^{-1}(y) = y$ and $f = [\nabla_u L]^{-1}( - \nabla_x \Phi) = - \nabla_x \Phi$
>
> * However, the notation of the inverse mapping is indeed confusing, so I rewrote $f$ using the Hamiltonian $H$, i.e., $f = \nabla_z H(t, x, z)|_{z=-\nabla_x \Phi(x, t)}$. Please confirm.
>
> > 2: Any comments on m in section B of the appendix is greatly appreciated. What is the benefit to improve $m$.
>
> Thank you for your question.
>
> * We added the following explanation in Appendix B.
>
>     $m$ is the number of dimensions of the hidden representation vector, so the complexity $O(m^2d)$ indicates that there is a trade-off between the expressive power of the DNN and the computational speed of the Hessian.
>
> (Note) Section 3 of the OT-Flow paper provides a detailed comparison of our Hessian calculation method with other Hessian calculation methods, such as the Hutchinson Hessian estimator. The following is a summary of the comparison in OT-Flow paper.
>
> > Our exact trace computation has similar computational complexity as FFJORD’s and RNODE’s trace estimation. In clock time, the analytic exact trace computation is competitive with the Hutchinson’s estimator using automatic differentiation (AD), while introducing no estimation error.
>
>
> > 3: Could the authors draw more connections between LSB and OT-Flow? If there is indeed a close connection, the more mathematical, the better.
>
> * We reorganized Appendix A. We discussed the optimality conditions for the LSB problem and the relaxed LSB problem in Appendix A.3. Then, we described the derivation of the NLSB and its connection to OT-Flow in Appendix A.4. Please check them.
>
>
> > 4: Is this algorithm applicable to generative tasks such as CIFAR10 simulation? If not, is there a scalability issue?
>
> This is a very interesting question and contains implications for future development. There is no serious scalability problem, so we can apply NLSB to the image generation task. Discovering a suitable Lagrangian for image generation (other than $L=\frac{1}{2}\|u|\|^2$) could lead to improved image generation performance and faster generation speed.
>
> > 5:  $L(p,m,Φ)$ and $L(Φ,p,m)$ is not consistent.
>
> We appreciate your finding our mistake. We fixed it.

---

> > ### Comment · Reviewer_6WTu · 2022-12-08
> > **Thanks for the response**
> >
> > I am satisfied with the response and appreciate the authors' hard work.
> >
> > I would suggest the authors run image generation tasks in future works to maximize the impact of the idea of LAGRANGIAN SCHRODINGER BRIDGE and detail limitations in this area.

---

### Official Review · Reviewer_hMLS · 2022-10-25

**Confidence:** 3
**Correctness:** 3
**Technical Novelty And Significance:** 2
**Empirical Novelty And Significance:** 2
**Recommendation:** 6

**Clarity, Quality, Novelty And Reproducibility:**

The Appendix A to explain Eq. (7,8,9) is provided but is not self-contained. Some more discussions on HJB solutions (based on OT-Fflow) are needed to arrive the final formulation in Eq. (7,8,9).

There are some minor gramatical errors which can be improved.

**Strength And Weaknesses:**

Strength:
- To my knowledge, the approach using the LSB is interesting and novel.


Weaknesses:
- The paper claims that one of the main contributions is the model architecture of potential functions. It seems that the paper adopts such an architecture from OT-Flow.
- The paper does not highlight or motivate the reason for using Lagrangian Schrodinger bridge.
- The empirical results, unfortunately, only give marginal improvements compared to models (i.e. IPML).


**Summary Of The Paper:**

The paper aims to learn population dynamics using neural stochastic differential equations.  The paper formulates the Lagrangian Schrodinger bridge (LSB) of optimal transport. Experiments are conducted on synthetic data sets and single-cell embryoid body scRNA-seq data.


**Summary Of The Review:**

The paper presents an interesting approach to learn neural SDEs for the problems of population dynamic modelling. However, the paper is not well-motivated and the technical contributions seem incremental.

---

> ### Author Response · Authors · 2022-11-16
> **Response to Reviewer hMLS (1/2)**
>
> > 1: The paper claims that one of the main contributions is the model architecture of potential functions. It seems that the paper adopts such an architecture from OT-Flow.
>
> Thank you for your comment. We use the special properties of the model differently, as discussed in Section 4.2.
>
> * To clarify our contribution, we revisited the text as follows.
>
>     (before) While this model architecture trick was originally used for speeding up the computation of the Jacobian term in neural ODE maximum likelihood training, we propose to use it as a technique to speed up the computation of $R_h$.
>
>     (after) While this model architecture trick was originally used for speeding up the computation of the Jacobian term in neural ODE maximum likelihood training, **we propose for the first time to use it as a technique to speed up the computation of $\mathcal{R}_h$ in neural SDE.**
>
> * We added detailed explanations on the use of potential functions of OT-Flow and NLSB respectively in Appendix B. Please check it.
>
> > 2: The paper does not highlight or motivate the reason for using Lagrangian Schrodinger bridge.
>
> We described our motivation for the Lagrangian Schrodinger bridge in the following parts of the paper.
>
> ---
>
> (Section1: Introduction)
>
> To handle the stochastic and complex behavior of individual samples, we propose to model the advection-diffusion processes by using SDEs to describe the time evolution of the sample. Furthermore, on the basis of the principle of least action, we estimate the sample trajectories that minimize action, defined by the time integral of the Lagrangian determined from the prior knowledge.
>
> ---
>
> (Section 3.1: Lagrangian Schrödinger Bridge)
>
> To solve this problem, we make two assumptions about the target system model.
>
> 1. The stochastic behavior of individual samples yields the population diffusion phenomenon.
> 2. Individual samples are encouraged to move according to the principle of least action.
>
> The principle of least action states that when a sample moves from point A to point B, its trajectory is the one that has least action, which is known as a fundamental principle in dynamical systems. We propose to explicitly model diffusion phenomena as well as drift by using SDEs for the time evolution of the sample and to explore the sample paths that minimize action defined by the time integral of the Lagrangian.
>
> ---
>
> However, we thought that the relationship between the LSB problem and the SOT problems including the SB problem was not clear and may have confused you. So I reorganized Appendix A to clarify them. Are there any other suggestions? I would like to hear your opinion.
>
> > 3: The empirical results, unfortunately, only give marginal improvements compared to models (i.e. IPML)
>
> Thank you for your comment. We guess you are talking about the results of experiment 5.1. However, this is not a weakness of our paper for the following three reasons.
>
> (1) Existing ODE-based methods fail in Experiment 5.1 which is rather an example of population dynamics arguing for the need for SDE-based methods (including NLSB and IPML). Furthermore, it visually demonstrates the inapplicability of the standard neural SDE to population dynamics in even simple settings and illustrates the need for a priori knowledge.
>
> (2) In Experiment 5.1, the problems solved by NLSB ($L=\frac{1}{2}\|u\|^2$) and IPML are almost mathematically equivalent. The result that NLSB shows comparable performance to IPML, even though NLSB is trained differently from existing IPML, indicates that NLSB is a unified framework and can deal with SB problems as a special case.
>
> (3) In Figure 6, IPML is able to adequately reproduce the ground-truth, suggesting that IPML and NLSB almost achieve the upper limit of accuracy in Experiment 5.1.
>
> * We revised the ‘Results’ part in Section 5.1 as follows.
>
>   Figure 3 shows that …
>   That indicates that **NLSB and IPF** can estimate population-level dynamics even when the population variance is large or small. Furthermore, **NLSB and IPF** have a smaller CDD value than neural SDE. … In contrast, the prediction by **NLSB and IPF** in Figs. 4c, 6d, and 6e are much closer to the ground-truth. These results show that the prior knowledge of the potential-free system helps to estimate the sample-level dynamics. …
>
> * We added the following description in Appendix D.1.
>
>   Note that the LSB problem solved by NLSB with the Lagrangian $L=\frac{1}{2}\|u\|^2$ and the SB problem solved by IPF are almost mathematically equivalent (see Appendix A.3 to A.5). The result that NLSB shows comparable performance to IPF, even though NLSB is trained differently from IPF, indicates that NLSB is a unified framework and can deal with SB problems as a special case.
> (We have renamed IPML as IPF in response to reviewer 8YwN.)

---

> > ### Author Response · Authors · 2022-11-16
> > **Response to Reviewer hMLS (2/2)**
> >
> > > 4: The Appendix A to explain Eq. (7,8,9) is provided but is not self-contained. Some more discussions on HJB solutions (based on OT-Flow) are needed to arrive the final formulation in Eq. (7,8,9).
> >
> > * We reorganized Appendix A. We discussed the optimality conditions for the LSB problem and the relaxed LSB problem in Appendix A.3. Then, we described the derivation of the NLSB and its connection to OT-Flow in Appendix A.4. Please check them.
> >
> > > 5: There are some minor grammatical errors which can be improved.
> >
> > Thank you very much. We fix them.
> >
> > > 6: the paper is not well-motivated and the technical contributions seem incremental.
> >
> > Our technical contributions are not incremental for the following reason.
> >
> > The OT-based regularizations proposed in TrajectoryNet and OT-Flow described in Section 2.1 is a loss function for neural ODEs based on theoretical results of the Brenier-Benamou (BB) problem. In contrast, we proposed the loss function for neural SDEs based on the theoretical results of our newly formulated LSB problem. The fact that the losses in Eqs. (8) and (9) appear to be in the general form of Eqs. (1) and (2) is due to non-trivial and desirable theoretical results obtained by the SOT theory and our theoretical contributions. We were the first to introduce the SOT theory into the field of machine learning, formulate the LSB problem, prove the existence of solutions to the LSB problem, and derive optimality conditions by variational methods (Theorem 17). Furthermore, Theorem 17 justifies the minimization of  and a parameterization that strictly satisfies Eq. (45) during training when the terminal constraint is not satisfied. The existing mathematical results are explained in Appendix A.1~A.3 and our theoretical contributions are highlighted in Appendix A.4 and A.5. Please check it.

---

> > ### Comment · Reviewer_hMLS · 2022-12-07
> > **Thanks for the response**
> >
> > Thank you for your time and effort improving Appendix which, I think, is now more accessible. The correction on the use of OT-Flow seems fair. However, I think the contribution here is not significant and should not be over-emphasized. In my opinion, the paper may need to rephase the last sentence of Introduction "3. Our model architecture of the potential function enables".
> >
> > Considering the recent update, I have raised the score to 6.

---

> > > ### Author Response · Authors · 2022-12-08
> > > **Response to Reviewer hMLS**
> > >
> > > I greatly appreciate your response and reconsideration of the score.
> > >
> > > > 1: the paper may need to rephrase the last sentence of Introduction "3. Our model architecture of the potential function enables”.
> > >
> > > Thank you for your suggestion. We revised the last sentence of the Introduction as follows.
> > >
> > > * We adopt the model architecture of the potential function from OT-Flow (Onken et al., 2021) to speed up the computation of the regularization term to minimize HJB-PDE loss.

---

### Official Review · Reviewer_WFgo · 2022-11-02

**Confidence:** 5
**Correctness:** 4
**Technical Novelty And Significance:** 2
**Empirical Novelty And Significance:** 2
**Recommendation:** 6

**Clarity, Quality, Novelty And Reproducibility:**

Writing is clear. Formulation of the regularizations and architecture are novel to my knowledge. Architecture and setup is clear and reproducible. I have some questions as to the exact hyperparameter tuning setup, including “We searched all weight coefficients of regularization terms in [0.0, 0.5]”, this can’t possibly be true right? Also is EMD-L2 the same as Wasserstein-2? these are not the same to me as Wasserstein-2 generally implies squared L2 cost.

**Strength And Weaknesses:**

Strengths:

- Clearly explained motivation formalizing previously adhoc regularizations in Tong et al. 2020
- Adds ideas from cellular modeling to SDE and Schrodinger bridge models for improved modeling of a cellular system.
- Good tradeoff of time vs. accuracy in figure 5 across dimensionalities.
- Interesting architecture adaptation for fast gradient computation.

Weaknesses:

- Limited evaluation (2 settings) with relatively minor improvements in performance.
- The network is not as easy to invert as other models which use ODE dynamics (Tong et al. 2020) or train both a forward and backwards model. I see that a reverse time simulation was performed in E.4, but it is unclear to me how well this works in practice (quantitatively). I have had numerical difficulties training this backwards network in the past.

Comments / questions:

There is something strange in these comparisons… what are the differences with NeuralSDE?

Figure 10, are the captions correct? It looks to me that (e) and (f) are switched.

“The weight coefficients $\lambda_e$ and $\lambda_h$ are tuned for each interval $[t_{k-1}, t_k]$ respectively”. Can the authors elaborate here? This seems concerning as there may be potential for data leakage.

It might be interesting to visualize the potential landscape (as is done in Bunne et al. 2022). Biologists are quite interested in what is termed “Waddington’s landscape” for developing systems.

Would it be possible to include an additional evaluation metric (besides the 2-Wasserstein distance)? The 2-Wasserstein distance mostly cares about outliers, that the distribution is “mostly right” which might miss some of the other differences between the predicted and true distributions. Common choices I believe are 1-Wasserstein distance (Tong et al. 2020, Vargas et al. 2021) and Maximum mean discrepancy (Huguet et al. 2022). Since you directly optimize for the 2-Wasserstein distance this may be an overoptimistic metric. If this is not possible it would at least be good to note the limitations of the current evaluation.

**Summary Of The Paper:**

This work tackles the problem of learning dynamics from population observations over time. Existing works have used Continuous normalizing flows (CNFs) or schrodinger bridges (SBs) to tackle this problem. This work (NLSB) combines adhoc regularizations in Tong et al. 2020 and more recent SDE formulations to learn a principled and regularized SDE to model cell populations and incorporate additional regularizations. The method is tested on a synthetic OU SDE and a single-cell RNA-seq dataset measuring embryoid body development over time.

**Summary Of The Review:**

This paper introduces a new architecture and training procedure for neural Lagrangian Schrodinger bridges. This provides a more principled approach to cellular systems than Tong et al., and includes additional prior information not present in the general Schrodinger bridge formulations. This is an incremental but solid contribution to the literature on modelling population dynamics with additional constraints.

---

> ### Author Response · Authors · 2022-11-16
> **Response to Reviewer WFgo (1/2)**
>
> > 1: Limited evaluation (2 settings) with relatively minor improvements in performance.
>
> Thank you for your comment. This is not a weakness of our paper for the following three reasons.
>
> (1) Existing ODE-based methods fail in Experiment 5.1 which is rather an example of population dynamics arguing for the need for SDE-based methods (including NLSB and IPML). Furthermore, it visually demonstrates the inapplicability of the standard neural SDE to population dynamics in even simple settings and illustrates the need for a priori knowledge.
>
> (2) In Experiment 5.1, the problems solved by NLSB ($L=\frac{1}{2}\|u\|^2$) and IPML are almost mathematically equivalent. The result that NLSB shows comparable performance to IPML, even though NLSB is trained differently from existing IPML, indicates that NLSB is a unified framework and can deal with SB problems as a special case.
>
> (3) In Figure 6, IPML is able to adequately reproduce the ground-truth, suggesting that IPML and NLSB almost achieve the upper limit of accuracy in Experiment 5.1.
>
> * We revised the ‘Results’ part in Section 5.1 as follows.
>
>   Figure 3 shows that …
>   That indicates that **NLSB and IPF** can estimate population-level dynamics even when the population variance is large or small. Furthermore, **NLSB and IPF** have a smaller CDD value than neural SDE. … In contrast, the prediction by **NLSB and IPF** in Figs. 4c, 6d, and 6e are much closer to the ground-truth. These results show that the prior knowledge of the potential-free system helps to estimate the sample-level dynamics. …
>
> * We added the following description in Appendix D.1.
>
>   Note that the LSB problem solved by NLSB with the Lagrangian $L=\frac{1}{2}\|u\|^2$ and the SB problem solved by IPF are almost mathematically equivalent (see Appendix A.3 to A.5). The result that NLSB shows comparable performance to IPF, even though NLSB is trained differently from IPF, indicates that NLSB is a unified framework and can deal with SB problems as a special case.
> (We have renamed IPML as IPF in response to reviewer 8YwN.)
>
> > 2: How well a reverse time simulation works in practice (quantitatively)?
>
> * We added the experimental results (Figure 12) and the following description in Appendix D.4. Please confirm it.
>
>     Experimental results using the MDD between reverse-time simulations and ground-truth are shown in Figure 12. Figure 12 shows that the proposed method in Appendix D.4 successfully recovers the population-level dynamics of the reverse-time SDE. The development of appropriate evaluation methods for sample-level dynamics of the reverse-time SDE is included in future work.
>
> > 3: What are the differences between NeuralSDE and NLSB?
>
> Neural SDE represents a method that does not use $R_e$ or $R_h$, but only Sinkhorn loss. In other words, Neural SDE cannot reflect the prior knowledge introduced by the Lagrangian in the dynamics.
>
> * For clarification, we revised the texts in Section 5 as follows.
>
>     (before) The drift and diffusion models of neural SDE are the same as NLSB.
>
>     (after) We trained the drift and diffusion models of the standard neural SDE using only the Sinkhorn divergence.
>
> > 4: Figure 10, are the captions correct? It looks to me that (e) and (f) are switched.
>
> Thank you for pointing out our mistakes. We fixed it.
>
> > 5: “The weight coefficients λ_e and λ_h are tuned for each interval [t_{k-1}, t_k] respectively”. Can the authors elaborate here?
>
> We prepared training, validation, and test data for all experiments, and used the validation data for tuning. As we describe in Appendix D.1 and Appendix D.2,  “We searched all weight coefficients of regularization terms in [0.0, 0.5] and selected those with the largest possible coefficients among those with sufficiently small EMD-L2 values on the validation data.” We determined the weight coefficients for the interval $[t_{k-1}, t_k]$ from the value of MDD (EMD-L2) at time $t_k$. For the calculation of the EMD-L2 value at the time $t_k$, we simulated the dynamics from $t_{k-1}$ to $t_k$ using the validation data at time $t_{k-1}$ and evaluated the EMD-L2 value between the simulated samples and the validation data at time $t_k$.
>
> > 6: It might be interesting to visualize the potential landscape (as is done in Bunne et al. 2022). Biologists are quite interested in what is termed “Waddington’s landscape” for developing systems.
>
> Thank you for your suggestion. I’m not sure if we can do it before the end of the revision period, but we will try to visualize the potential landscape.
>
> > 7: Would it be possible to include an additional evaluation metric (besides the 2-Wasserstein distance)?
>
> Thank you for your comments.
> * We added the results using EMD with L1 cost in Appendix D.2. Please check it.

---

> > ### Author Response · Authors · 2022-11-20
> > **Response to Reviewer WFgo (2/2)**
> >
> > > 8: Also is EMD-L2 the same as Wasserstein-2?
> >
> > Thank you for your question. EMD-L2 is defined by EMD-L2$= \min_{\pi_{i, j}} \sum_{i, j} || x_i - y_j ||^2 \pi_{i, j}$, which is the squared approximation value of the Wasserstein-2 distance using finite samples $x_i, x_j$. Wasserstein-2 distance is defined by $\mathcal{W}(\mu, \nu) := \left(\inf_{\pi} \int_{X \times X} ||x - y||^2 \pi(x, y) dxdy \right)^{\frac{1}{2}}$.

---

> > > ### Author Response · Authors · 2022-12-01
> > > **Report Code issues and fixes**
> > >
> > > We checked the code and found that I had incorrectly evaluated the value of EMD-L2 with $\min_{\pi} \sum_{i, j} \sqrt{||x_i-y_j||^2} \pi_{i,j}$. So we corrected the EMD-L2 evaluation with $\sqrt{\min_{\pi} \sum_{i,j} ||x_i-y_j||^2 \pi_{i,j}}$.
> > >
> > > All results did not change significantly, but please check https://github.com/fhiuwbhnvsu/image_archive.
> > >
> > > Table 1
> > > | MDD (EMD-L2) | $t_1$ | $t_2$ | $t_3$ | $t_4$ |
> > > | :--- | :---: | :---: | :---: | :---: |
> > > | NLSB (E) | 0.71$\pm$ 0.020 | 0.86 $\pm$ 0.027 | 0.83 $\pm$ 0.016 | 0.79 $\pm$ 0.012 |
> > > | NLSB (D) | 0.67 $\pm$ 0.017 | 0.90 $\pm$ 0.029 | 0.87 $\pm$ 0.018 | 0.79 $\pm$ 0.016 |
> > > | NLSB (V) | 0.70 $\pm$ 0.023 | 0.89 $\pm$ 0.030 | 0.83 $\pm$ 0.022 | 0.81 $\pm$ 0.019 |
> > > | NLSB (E+D+V) | 0.68 $\pm$ 0.016 | 0.84 $\pm$ 0.030 | 0.81 $\pm$ 0.018 | 0.79 $\pm$ 0.017 |
> > > | Neural SDE | 0.69 $\pm$ 0.020 | 0.91 $\pm$ 0.029 | 0.85 $\pm$ 0.025 | 0.81$\pm$ 0.017 |
> > > | OT-Flow | 0.83 | 1.10 | 1.07 | 1.05 |
> > > | OT-Flow + OT | 0.85 | 1.05 | 1.09 | 1.00 |
> > > | TrajectoryNet | 0.73 | 1.06 | 0.90 | 1.01 |
> > > | TrajectoryNet + OT | 0.76 | 1.05 | 0.88 | 1.10 |
> > > | IPF (GP) | 0.70 $\pm$ 0.015 | 1.04 $\pm$ 0.041 | 0.94 $\pm$ 0.029 | 0.98 $\pm$ 0.033 |
> > > | IPF (NN) | 0.73 $\pm$ 0.019 | 0.89 $\pm$ 0.030 | 0.84 $\pm$ 0.019 | 0.83 $\pm$ 0.020 |
> > >
> > > Table 2
> > > |  Mean CDD  | $[t_0, t_2]$ | $[t_1, t_3]$ | $[t_2, t_4]$ |
> > > |--------------|-------------|--------------|--------------|
> > > | NLSB (E)     | $0.88$      | $0.72$       | $0.79$       |
> > > | NLSB (D)     | $1.64$      | $0.76$       | $0.77$       |
> > > | NLSB (V)     | $1.15$      | $0.82$       | $0.86$       |
> > > | NLSB (E+D+V) | $0.96$      | $0.83$       | $0.84$       |
> > > | Neural SDE   | $1.36$      | $0.85$       | $0.87$       |
> > > | IPF (GP)     | $0.97$      | $1.03$       | $1.06$       |

---

> > > > ### Comment · Reviewer_WFgo · 2022-12-12
> > > > **Thank you for your response and thoughtful work**
> > > >
> > > > I thank the authors for their response and for answering most of my questions. Overall my opinion after the revision is that the empirical advantages are now slightly more clear with the added metrics. However, there are still some questions as to how much of the improvement is due to hyper parameter tuning. I still don't fully see the connection between LSB and SB, it seems that LSB is more closely connected to SOT. Overall, I still lean slightly towards acceptance, but there are still open questions regarding theory. Some more thoughts below.
> > > >
> > > > I think there is a bit of confusion for the authors between the EMD-L2 and the 2-Wasserstein-L2. Personally I prefer if the EMD-L2 denotes a 1-Wasserstein distance with L2 ground distance, as the earth mover's distance (EMD) is generally defined as the 1-Wasserstein distance. It is my understanding that your original EMD-L2 code is actually the 1-Wasserstein distances with L2 ground distance. This new evaluation is the 2-Wasserstein with L2 ground distance.  I believe your original numbers for Tables 1 and 2 are more appropriate, but it would be great to include the L1 and 2-Wasserstein numbers in the appendix.
> > > >
> > > > For clarity:
> > > > $$\text{EMD-L2}(a, b) := \min_{\pi} ||x - y||_2 d \pi(x, y)$$
> > > > $$\text{2-Wasserstein-L2}(a, b) := \sqrt{\min_{\pi} ||x - y||_2^2 d \pi(x, y)}$$
> > > >
> > > > > The result that NLSB shows comparable performance to IPML, even though NLSB is trained differently from existing IPML, indicates that NLSB is a unified framework and can deal with SB problems as a special case.
> > > >
> > > > I urge the authors to be cautious here. SB is *not* a special case of NLSB as far as I can tell. The standard Brownian motion SB problem might be a special case of LSB, but certainly not the general formulation. Although even after reading through the new theory theory presented in Appendix A this is not clear to me. It seems that LSB is shown to generalize stochastic optimal transport, (New Thm. 17), but it is unclear to me how / if this shows that SB is a special case of LSB. I guess the question is why is this Lagrangian-Schr\"dinger Bridge problem and not Lagrangian Stochastic Optimal Transport?
> > > >
> > > > I think the author's may have missed one of my comments:
> > > > > I have some questions as to the exact hyperparameter tuning setup, including “We searched all weight coefficients of regularization terms in [0.0, 0.5]”
> > > >
> > > > It would be great if this could be clarified further. Clearly not all real numbers in [0.0,0.5] were searched, maybe a line search of some kind?
> > > >
> > > >
> > > > Some small suggestions:
> > > > * It might be helpful to separate out the parameterized functions from the LSB definition into a NLSB definition, as these parameters are again dropped in the appendix
> > > > * The new Theory section has a lot of conflicting notation from various works that it would nice to unify.
> > > > * I saw some work that might be of interest to the authors at NeurIPS https://arxiv.org/abs/2206.14928. No need to compare to this work, but may be interesting to discuss. I believe NLSB has some significant benefits when compared to this work.

---

> > > > > ### Author Response · Authors · 2022-12-12
> > > > > **Response to Reviewer WFgo**
> > > > >
> > > > > > 1: I think there is a bit of confusion for the authors between the EMD-L2 and the 2-Wasserstein-L2.
> > > > >
> > > > > We followed the EMD-distance definition of the following common optimal transport library. https://pythonot.github.io/all.html#ot.emd2
> > > > >
> > > > > We have modified the code to calculate as follows:
> > > > > ```python
> > > > > M = torch.pow(torch.cdist(ref_data, pred_data, p=p),p)
> > > > > a, b = ot.unif(pred_data.size()[0]), ot.unif(ref_data.size()[0])
> > > > > loss = ot.emd2(a, b, M.cpu().detach().numpy(),  numItermax=1000000)
> > > > > if p == 2:
> > > > >    loss = np.sqrt(loss)
> > > > > return loss
> > > > > ```
> > > > > We show both EMD-L2 and EMD-L1 results in our paper.
> > > > >
> > > > > > 2:   The standard Brownian motion SB problem might be a special case of LSB, but certainly not the general formulation.
> > > > >
> > > > > Indeed, NLSB with L=1/2|||u||^2 has a more generalized diffusion function and so does not exactly match SB. However, this difference is actually trivial because ou-sde is one-dimensional and very simple.
> > > > >
> > > > > > 3:  if this shows that SB is a special case of LSB. I guess the question is why is this Lagrangian-Schr"dinger Bridge problem and not Lagrangian Stochastic Optimal Transport?
> > > > >
> > > > > In recent years, many ML-based methods for solving SB problems have been proposed, and in this context, we named ours the LSB problem, since generalization using Lagrangian is our main proposal. However, if it is confusing, we will correct it; we do not think Lagrangian Stochastic Optimal Transport is a good naming since the SOT problem is formulated using the Lagrangian.
> > > > >
> > > > > > 4: It would be great if this could be clarified further. Clearly not all real numbers in [0.0,0.5] were searched, maybe a line search of some kind?
> > > > >
> > > > > These parameters are the weight coefficients for regularization, and if they are too large, training will not be successful. Existing studies such as TrajectoryNet and OT-FLow have also tuned these values around $[0.0, 0.5]$.
> > > > >
> > > > > And thank you for your all suggestions. Due to lack of time, I can't show the revised texts, but I promise to reflect your suggestions in camera-ready version.

---

> > > > > > ### Comment · Reviewer_WFgo · 2022-12-12
> > > > > > **Clarification**
> > > > > >
> > > > > > > 1 We show both EMD-L2 and EMD-L1 results in our paper.
> > > > > >
> > > > > > Thank you, however to me this looks like the EMD-L1 and the 2-Wasserstein L2 to me for $p=1$ and $p=2$ respectively. EMD-L2 would be
> > > > > >
> > > > > > ```
> > > > > > M = torch.cdist(ref_data, pred_data, p=2)
> > > > > > a, b = ot.unif(pred_data.size()[0]), ot.unif(ref_data.size()[0])
> > > > > > loss = ot.emd2(a, b, M.cpu().detach().numpy(),  numItermax=1000000)
> > > > > > return loss
> > > > > > ```
> > > > > >
> > > > > > To be clear, I think these metrics are fine, but the current metric called EMD-L2 shouldn't be called that as it is not an EMD.
> > > > > >
> > > > > > >>2: The standard Brownian motion SB problem might be a special case of LSB, but certainly not the general formulation.
> > > > > >
> > > > > > >Indeed, NLSB with L=1/2|||u||^2 has a more generalized diffusion function and does not exactly match SB. However, this difference is actually trivial because ou-sde is one-dimensional and very simple.
> > > > > >
> > > > > > I don't understand this remark sorry. Particularly what the second sentence is getting at.
> > > > > >
> > > > > > >> 3: if this shows that SB is a special case of LSB. I guess the question is why is this Lagrangian-Schr"dinger Bridge problem and not Lagrangian Stochastic Optimal Transport?
> > > > > > > LSB is just a name. If it causes confusion, we will correct it; we do not think Lagrangian Stochastic Optimal Transport is a good naming since the SOT problem is formulated using the Lagrangian.
> > > > > >
> > > > > > Ahhh yes, I see why LSOT would be confusing. I'm okay with the current name, it's just not clear to me the connection from SOT --> SB. Thoughts on Neural Stochastic Optimal Transport?

---

> > > > > > > ### Author Response · Authors · 2022-12-12
> > > > > > > **Response**
> > > > > > >
> > > > > > > > I don't understand this remark sorry. Particularly what the second sentence is getting at.
> > > > > > >
> > > > > > > Sorry, I misread your question. The LSB problem is not a generalization of the GENERAL SB problem. When we refer to the SB problem in this paper, we are referring to the classical SB problem in the case of standard Brownian motion.
> > > > > > >
> > > > > > > We will clarify this point more in the text.
> > > > > > >
> > > > > > > > To be clear, I think these metrics are fine, but the current metric called EMD-L2 shouldn't be called that as it is not an EMD.
> > > > > > >
> > > > > > > I understand your point. Since this is just a matter of scale, the results of the quantitative evaluation will not essentially change, but I will correct it.
> > > > > > >
> > > > > > > > Ahhh yes, I see why LSOT would be confusing. I'm okay with the current name, it's just not clear to me the connection from SOT --> SB.
> > > > > > >
> > > > > > > When $L=1/2\|u\|^2$ and $g = \sqrt{2\epsilon}$, the SOT problem is reduced to the classical SB problem in the case of standard Brownian motion. The relationship between the SOT problem and the GENERAL SB problem is theoretically unclear.
> > > > > > >
> > > > > > > > Thoughts on Neural Stochastic Optimal Transport?
> > > > > > >
> > > > > > > This is actually what we thought, but we think our paper should be read by people who are looking for a method to solve SB problems. Therefore, we wanted to include the word SB in the name.

---

> > > > > > > > ### Comment · Reviewer_WFgo · 2022-12-12
> > > > > > > > **Agreed Thanks**
> > > > > > > >
> > > > > > > > Thanks for the clarification, yes I believe this would be clearer as there are multiple senses of "classical".
> > > > > > > >
> > > > > > > > Agreed, the results are valid and will not change, thank you for clarifying this in the text.

---

> > > > > > > > > ### Author Response · Authors · 2022-12-13
> > > > > > > > > **Thank you for your suggestions**
> > > > > > > > >
> > > > > > > > > We also explain the reasons for naming the LSB in more detail in the Appendix to avoid misunderstandings. Thank you for your valuable feedback.

---

> > > > > ### Author Response · Authors · 2022-12-13
> > > > > **Thank you for your valuable suggestions**
> > > > >
> > > > > > It might be helpful to separate out the parameterized functions from the LSB definition into a NLSB definition, as these parameters are again dropped in the appendix
> > > > >
> > > > > Thank you for your suggestion. We rewrite the symbols $\mathbf{f}\_{\theta}$, $\mathbf{g}\_{\phi}$, $\rho\_t(x; \theta, \phi)$ as $\mathbf{f}$, $\mathbf{g}$, $\rho\_t(x; \mathbf{f}, \mathbf{g})$ in the definition of the LSB problem (Eq. (6)).
> > > > >
> > > > > > The new Theory section has a lot of conflicting notation from various works that it would nice to unify.
> > > > >
> > > > > We unify all notations for drift and velocity functions with $\mathbf{f}$ and all notations for the diffusion functions with $\mathbf{g}$.
> > > > >
> > > > > > I saw some work that might be of interest to the authors at NeurIPS https://arxiv.org/abs/2206.14928. No need to compare to this work, but may be interesting to discuss. I believe NLSB has some significant benefits when compared to this work.
> > > > >
> > > > > Thank you for introducing me to a very interesting and relevant study. We define SDE on the raw data space or PCA space. The direction of specifying some geometric structures on the space of SDE, as in this paper, is a very promising direction for further future development of the NLSB. In Appendix D.5, we discuss only linear coordinate transformations, but if we do non-linear coordinate transformations, as in this paper, the meaning of cost changes completely. It is not easy, but if Lagrangian cost and SDE space can be properly selected together as a pair, we are likely to be able to make more accurate estimates. We will cite this paper and add this discussion to Appendix D.5.
> > > > >
> > > > > ---
> > > > >
> > > > > To avoid future confusion regarding the naming of LSB, We made the following two revisions.
> > > > >
> > > > > - We revised the text in Appendix D.1 as follows.
> > > > >
> > > > >     Note that the LSB problem solved by NLSB with the Lagrangian L = 1/2||u||^2 and **the classical SB problem with standard Brownian motion as a prior** solved by IPF are almost mathematically equivalent (see Appendices A.3 to A.5).
> > > > >
> > > > >
> > > > > - We added the following text in Appendix A.4
> > > > >
> > > > >     Modeling the evolution of a stochastic process between sampled time points is classically called the Schr¨odinger bridge problem. In tribute to this problem set, we call our problem setting this way.

---

### Decision · Program_Chairs · 2023-01-20

**Decision:**

Accept: notable-top-25%

**Justification For Why Not Higher Score:**

I think the paper underwent several modifications, which do not make me feel confident to support an oral, but I think the content is very timely and will be of interest to the diffusion crowd.

**Justification For Why Not Lower Score:**

Not reject because the rebuttal convinced a very knowledgeable reviewer to switch from a low to very high (8) score. I prefer having this as a spotlight because I think the paper is great and addresses a hot topic.

**Metareview: Summary, Strengths And Weaknesses:**

The authors propose to model a diffusion between two known marginal distributions. They formulate the Lagrangian Schrödinger bridge (LSB) problem, a generalization of the Schrödinger bridge problem using a Lagrangian to quantify displacement costs (see Sec.3.3). They solve this process using samples through regularized neural SDEs.

The paper was initially criticized for lacking on the experimental side, and for failing in some aspects w.r.t. the accuracy of its bibliographic references. Most of these concerns have been solved following a lengthy rebuttal process. I am therefore happy to suggest acceptance, with the important caveat that the authors will need to put all of the observations in their final version, including comparisons w.r.t. other baselines that have been requested.

**Note From Pc:**

if the above contains the word "oral" or "spotlight" please see: "oral" presentation means -> notable-top-5% and "spotlight" means -> notable-top-25%. As stated in our emails, we are disassociating presentation type from AC recommendations